# PROVABLE DOMAIN GENERALIZATION VIA INFORMATION THEORY GUIDED DISTRIBUTION MATCHING

## ABSTRACT

Domain generalization (DG) aims to learn predictors that perform well on unseen data distributions by leveraging multiple related training environments. To this end, DG is commonly formulated as an average or worst-case optimization problem, which however either lacks robustness or is overly conservative. In this work, we propose a novel probabilistic framework for DG by minimizing the gap between training and test-domain population risks. Our formulation is built upon comprehensive information-theoretic analysis and enables direct optimization without stringent assumptions. Specifically, we establish information-theoretic upper bounds for both source and target-domain generalization errors, revealing the key quantities that control the capability of learning algorithms to generalize on unseen domains. Based on the theoretical findings, we propose Inter-domain Distribution Matching (IDM) for high-probability DG by simultaneously aligning inter-domain gradients and representations, and Per-sample Distribution Matching (PDM) for high-dimensional and complex data distribution alignment. Extensive experimental results validate the efficacy of our methods, showing superior performance over various baseline methods.

## 1 INTRODUCTION

In real-world scenarios, distribution shifts are inevitable due to variations in the data collection procedures, resulting in machine learning systems overfitting to environment-specific correlations that may negatively impact performance when facing out-of-distribution (OOD) data (Geirhos et al., 2018; Hendrycks & Dietterich, 2019; Azulay & Weiss, 2019; Hendrycks et al., 2021). The DG problem is then proposed in the literature to address this challenge. By assuming that the training data constitutes multiple training domains that share some invariant underlying correlations, DG algorithms then attempt to learn this invariance so that domain-specific variations do not affect the model's performance. To this end, various DG approaches have been proposed, including invariant representation learning (Sun & Saenko, 2016; Li et al., 2018b), adversarial learning (Ganin et al., 2016; Li et al., 2018c), causal inference (Arjovsky et al., 2019; Chevalley et al., 2022), gradient manipulation (Koyama & Yamaguchi, 2020; Shi et al., 2021; Rame et al., 2022), and robust optimization (Sagawa et al., 2019; Eastwood et al., 2022) techniques.

DG is commonly formulated as an average-case (Blanchard et al., 2021; Zhang et al., 2021) or worst-case (Arjovsky et al., 2019; Sagawa et al., 2019) optimization problem, which however either lacks robustness against OOD data (Arjovsky et al., 2019; Nagarajan et al., 2020) or leads to overly conservative solutions (Eastwood et al., 2022). In this paper, we formulate DG from a novel probabilistic perspective, by measuring the ability to minimize the gap between training and test-domain population risks with high probability. Our formulation leverages the mild identical distribution assumption of the environments and enables direct optimization. Through comprehensive information-theoretic generalization analysis, we provide key insights into high-probability DG by showing that the input-output mutual information of the learning algorithm and the extent of distribution shift together control the gap between training and test-domain population risks.

Motivated by these theoretical findings, we propose Inter-domain Distribution Matching (IDM) for high-probability DG by aligning marginal distributions of the gradients and the representations across different training domains, which are proven to promote source and target-domain generalization respectively. Furthermore, we demonstrate that traditional distribution alignment techniques

based on moment matching are either ineffective or insufficient for high-dimensional and complex probability distributions. To circumvent these issues, we propose Per-sample Distribution Matching (PDM) for distribution alignment by aligning individual sorted data points. IDM jointly working with PDM achieves superior performance on the Colored MNIST dataset (Arjovsky et al., 2019) and the DomainBed benchmark (Gulrajani & Lopez-Paz, 2020).

The primary contributions of this paper can be summarized as follows:

- **A probabilistic formulation**: We measure the capability of DG algorithms to minimize the gap between training and test-domain population risks with high probability. Our formulation leverages mild assumptions of the environments and enables direct optimization (Section 2).
- **An information-theoretic perspective**: We derive novel information-theoretic upper bounds for both source and target-domain generalization errors, providing explanations for the success of DG algorithms based on gradient or representation matching in the literature (Section 3).
- **A novel DG algorithm**: We propose IDM for high-probability DG by simultaneously aligning inter-domain distributions of the gradients and the representations. We further propose PDM for high-dimensional distribution matching by aligning individual sorted data points (Section 4).
- We validate the effectiveness of the proposed IDM method on Colored MNIST and DomainBed, achieving superior performance over various baseline methods (Section 6).

## 2 PROBLEM SETTING

We denote random variables by capitalized letters ($X$), their realizations by lower-case letters ($x$), and their spaces by calligraphic letters ($\mathcal{X}$). Let $\mathcal{Z} = \mathcal{X} \times \mathcal{Y}$ be the instance space of interest, where $\mathcal{X}$ and $\mathcal{Y}$ are the input space and the label space respectively. Let $\mathcal{W}$ be the hypotheses space, each $w \in \mathcal{W}$ characterizes a predictor $f_w$ mapping from $\mathcal{X}$ to $\mathcal{Y}$, comprised of an encoder $f_\phi$: $\mathcal{X} \mapsto \mathcal{R}$ and a classifier $f_\psi$: $\mathcal{R} \mapsto \mathcal{Y}$ with the assist of an intermediate representation space $\mathcal{R}$.

We assume an existing distribution $\nu$ over all possible environments $\mathcal{D}$. The source domains $D_{tr} = \{D_i\}_{i=1}^m$ and target domains $D_{te} = \{D_k\}_{k=1}^{m'}$ are both randomly sampled from $\nu$, with each domain $d$ corresponding to a specific data-generating distribution $\mu_d$. Let $S_{tr} = \{S_i\}_{i=1}^m$ denote the training dataset, with each subset $S_i = \{Z_j^i\}_{j=1}^n$ containing i.i.d data sampled from $\mu_{D_i}$. The task is to design algorithm $\mathcal{A}: \mathcal{D}^m \mapsto \mathcal{W}$, taking $D_{tr}$ as the input (with proxy $S_{tr}$) and providing possibly randomized hypothesis $W = \mathcal{A}(D_{tr})$. Given the loss function $\ell : \mathcal{Y} \times \mathcal{Y} \mapsto \mathbb{R}^+$, the ability of some hypothesis $w \in \mathcal{W}$ to generalize in average is evaluated by the expected population risk:

$$L(w) = \mathbb{E}_{D \sim \nu}[L_D(w)] = \mathbb{E}_{D \sim \nu}[\mathbb{E}_{Z \sim \mu_D}[\ell(f_w(X), Y)]] = \mathbb{E}_{Z \sim \mu}[\ell(f_w(X), Y)].$$

Since $\nu$ is unknown in practice, only the source and target-domain population risks are tractable:

$$L_{tr}(w) = \frac{1}{m} \sum_{i=1}^m L_{D_i}(w) \qquad \text{and} \qquad L_{te}(w) = \frac{1}{m'} \sum_{k=1}^{m'} L_{D_k}(w).$$

**Main Assumptions.** We list the assumptions considered in our theoretical analysis as follows:

**Assumption 1.** *(Independent) The target domains $D_{te}$ are independent of source domains $D_{tr}$.*

**Assumption 2.** *(Bounded) The loss function $\ell(\cdot, \cdot)$ is bounded in $[0, M]$.*

**Assumption 3.** *(Subgaussian) $\ell(f_w(X), Y)$ is $\sigma$-subgaussian w.r.t $Z \sim \mu$ for any $w \in \mathcal{W}$.*

**Assumption 4.** *(Metric) The loss function $\ell(\cdot, \cdot)$ is symmetric and satisfies the triangle inequality, i.e. for any $y_1, y_2, y_3 \in \mathcal{Y}$, $\ell(y_1, y_2) = \ell(y_2, y_1)$ and $\ell(y_1, y_2) \leq \ell(y_1, y_3) + \ell(y_3, y_2)$.*

**Assumption 5.** *(Lipschitz) The loss function $\ell(f_w(X), Y)$ is $\beta$-Lipschitz w.r.t the metric $c$ on $\mathcal{Z}$ for any $w \in \mathcal{W}$, i.e. for any $z_1, z_2 \in \mathcal{Z}$, $|\ell(f_w(x_1), y_1) + \ell(f_w(x_2), y_2)| \leq \beta c(z_1, z_2)$.*

Subgaussianity (Assumption 3) is one of the most common assumptions for information-theoretic generalization analysis (Xu & Raginsky, 2017; Negrea et al., 2019; Neu et al., 2021; Wang & Mao, 2021). Notably, Assumption 2 is a strengthened version of Assumption 3, since any $[0, M]$-bounded random variable is always $M/2$-subgaussian. Lipschitzness (Assumption 5) is a crucial prerequisite for stability analysis and has also been utilized in deriving Wasserstein distance generalization bounds (Hardt et al., 2016; Bassily et al., 2020; Lei et al., 2021; Rodríguez Gálvez et al., 2021; Yang et al., 2021b;a). Assumption 4 is fulfilled when distance functions, such as mean absolute error

(MAE) and 0-1 loss, are used as loss functions. This assumption has also been examined in previous studies (Mansour et al., 2009; Shen et al., 2018; Wang & Mao, 2022).

**High-Probability DG** The classical empirical risk minimization (ERM) technique, which minimizes the average-case risk: $\min_w L(w)$, is found ineffective in achieving invariance across different environments (Arjovsky et al., 2019; Nagarajan et al., 2020). To overcome this limitation, recent works (Krueger et al., 2021; Ahuja et al., 2021; Shi et al., 2021; Rame et al., 2022; Lin et al., 2022; Zhou et al., 2022) have cast DG as a worst-case optimization problem: $\min_w \max_d L_d(w)$. However, this approach is generally impractical without strong assumptions made in the literature (Christiansen et al., 2021; Eastwood et al., 2022), e.g. linearity of the underlying causal mechanism (Arjovsky et al., 2019; Krueger et al., 2021; Ahuja et al., 2021), or strictly separable spurious and invariant features (Zhou et al., 2022). On the contrary, we propose the following high-probability objective by leveraging the mild Assumption 1, which is trivially satisfied in practice.

**Problem 1.** *(High-Probability DG)* $\min_{\mathcal{A}} \mathbb{E}[L_{tr}(W)], \ \ s.t. \ \ \Pr\{|L_{te}(W) - L_{tr}(W)| \geq \epsilon\} \leq \delta.$

Problem 1 is directly motivated by intuition that the training-domain population risk $L_{tr}(W)$ should be predictive of the test-domain risk $L_{te}(W)$, and the optimal algorithm $\mathcal{A}$ should be chosen in consideration of minimizing the gap between the two. Here, the probability is taken over the sampling process of the two groups of domains ($D_{tr}$ and $D_{te}$) and the training process of the hypothesis ($W$). Note that the source (or target) domains are not necessarily independent of each other.

## 3 INFORMATION-THEORETIC ANALYSIS

The primary goal of DG is to tackle the distribution shift problem, where the data-generating distribution $\mu_d$ varies depending on the corresponding environment $d$, influenced by the data collection process. This inconsistency can be quantified by the mutual information $I(Z; D)$ between data pair $Z$ and environment identifier $D$, which can be further decomposed into (Federici et al., 2021):

$$I(Z; D) \text{ (distribution shift)} = I(X; D) \text{ (covariate shift)} + I(Y; D|X) \text{ (concept shift)}. \quad (1)$$

While $\mathcal{D}$ is binary to differentiate training and test samples in (Federici et al., 2021), we extend this concept to any discrete or continuous space, provided that each $d \in \mathcal{D}$ corresponds to a distinct data distribution $\mu_d = P_{Z|D=d}$. The right-hand side (RHS) characterizes the changes in the marginal input distribution $P_X$ (covariate shift) as well as the predictive distribution $P_{Y|X}$ (concept shift). As we will show later, these two quantities capture the main challenges of the DG problem.

We start by demonstrating that the achievable level of average-case risk $L(w)$ is constrained by the degree of concept shift. Specifically, we have the following theorem:

**Theorem 1.** *For any predictor $Q_{Y|X}$, we have* $\mathrm{KL}(P_{Y|X,D} \parallel Q_{Y|X}) \geq I(Y; D|X)$.

When $\ell$ represents the cross-entropy loss, the population risk of predictor $Q$ on domain $d$ can be represented as the KL divergence between $P_{Y|X,D=d}$ and $Q_{Y|X}$, provided that $H(Y|X, D) = 0$ (i.e. the label can be entirely inferred from $X$ and $D$). This implies that any model fitting well in training domains will suffer from strictly positive risks in test domains once concept shift is induced. This observation verifies the trade-off between optimization and generalization as we characterized in Problem 1, and highlights the inherent difficulty of the DG problem.

We further show that Problem 1 directly serves as an optimization objective by connecting source and target-domain population risks via the average-case risk $L(W)$. To be specific, since the predictor $W$ is trained on the source domains $D_{tr}$, it is reasonable to assume that $W$ achieves lower population risks on $D_{tr}$ than on average, i.e. $L_{tr}(W) \leq L(W)$. Moreover, since the sampling process of test domains is independent of the hypothesis, the test-domain population risk $L_{te}(W)$ is actually an unbiased estimate of $L(W)$. Combining these two observations, it is natural to assume that $L_{tr}(W) \leq L(W) \approx L_{te}(W)$, implying that the average-case risk $L(W)$ acts as a natural bridge between the two. For any constant $\lambda \in (0, 1)$, we have the following decomposition:

$$\Pr\{|L_{tr}(W) - L_{te}(W)| \geq \epsilon\} \leq \Pr\{|L_{tr}(W) - L(W)| \geq \lambda\epsilon\} + \Pr\{|L_{te}(W) - L(W)| \geq (1-\lambda)\epsilon\}$$

While the first event on the RHS heavily correlates with the training domains $D_{tr}$, the second event is instead data-independent. This observation inspires us to explore both data-dependent and data-independent generalization bounds for source and target-domain population risks respectively, which serve as the basis for our algorithmic design presented in Section 4.

**Generalization Bounds for Source-Domain Population Risk**

Our results are motivated by recent advancements in characterizing the generalization behavior of learning algorithms within the information-theoretic learning framework (Bu et al., 2020; Harutyunyan et al., 2021). Specialized to our problem, we quantify the changes in the hypothesis once the training domains are observed through the input-output mutual information $I(W; D_i)$:

**Theorem 2.** *Let $W = \mathcal{A}(D_{tr})$. If Assumption 2 holds, then $|\mathbb{E}_{W,D_{tr}}[L_{tr}(W)] - \mathbb{E}_W[L(W)]| \leq$ $\frac{1}{m} \sum_{i=1}^{m} \sqrt{\frac{M^2}{2} I(W, D_i)}$, and $\Pr\{|L_{tr}(W) - L(W)| \geq \epsilon\} \leq \frac{M^2}{\epsilon^2}(\frac{1}{m} \sum_{i=1}^{m} I(W, D_i) + \log 3)$.*

Intuitively, extracting correlations between $X$ and $Y$ that are invariant across training domains does not lead to changes in the hypothesis when these domains are observed. The input-output mutual information $I(W; D_i)$ approaches zero when the correlations that a model learns from a specific training domain $D_i$ are also present in other training environments. This does not imply that the model learns nothing from $D_{tr}$: by assuming the independence of these domains, the summation of $I(W, D_i)$ can be relaxed to $I(W; D_{tr})$, which measures the actual amount of information that the model learned from all training domains. By minimizing each $I(W; D_i)$ and $L_{tr}(W)$ simultaneously, learning algorithms are encouraged to discard domain-specific correlations while preserving invariant ones and thus achieve high generalization performance.

We further present an alternative approach by assuming Lipschitzness instead of Subgaussianity, which usually leads to tighter bounds beyond information-theoretic measures:

**Theorem 3.** *Let $W = \mathcal{A}(D_{tr})$. If $\ell(f_w(X), Y)$ is $\beta'$-Lipschitz w.r.t $w$, then $|\mathbb{E}_{W,D_{tr}}[L_{tr}(W)] - \mathbb{E}_W[L(W)]| \leq \frac{\beta'}{m} \sum_{i=1}^{m} \mathbb{E}_{D_i}[\mathbb{W}(P_{W|D_i=d}, P_W)]$.*

Here $\mathbb{W}$ denotes the Wasserstein distance with metric $c$ defined in Assumption 5, serving as a class of distance measures between probability density functions (PDF) since each PDF necessarily integrates to 1. Besides its elegant symmetry compared to KL divergence, the Wasserstein upper bound is considered a tighter improvement over KL divergence or mutual information bounds. To see this, we assume that $c$ is discrete, which leads to the following reductions:

$$\mathbb{E}[\mathbb{W}(P_{W|D_i=d}, P_W)] = \mathbb{E}[\text{TV}(P_{W|D_i=d}, P_W)] \leq \mathbb{E}\sqrt{\tfrac{1}{2}\text{KL}(P_{W|D_i=d} \| P_W)} \leq \sqrt{\tfrac{1}{2} I(W; D_i)} \quad (2)$$

where TV is the total variation. These reductions confirm the superiority of Theorem 3 over Theorem 2 through a stronger Lipschitz assumption. Meanwhile, the RHS of Theorem 2 also upper bounds these alternative measures of domain differences i.e. total variation and Wasserstein distance, and thus minimizing $I(W; D_i)$ simultaneously penalizes these alternative measures. This observation encourages us to directly penalize the input-output mutual information, which is also shown to be easier and more stable for optimization (Nguyen et al., 2021; Wang & Mao, 2022).

**Generalization Bounds for Target-Domain Population Risk**

Since the training process is independent of the sampling process of test domains, we could consider the predictor as some constant hypothesis $w \in \mathcal{W}$. Then it is straightforward to verify that $\mathbb{E}_{D_{te}}[L_{te}(w)] = L(w)$ due to their identical marginal distribution. Moreover, by combining Assumption 3, we obtain the following high-probability bound:

**Theorem 4.** *If Assumption 3 holds, then $\forall w \in \mathcal{W}$, $\Pr\{|L_{te}(w) - L(w)| \geq \epsilon\} \leq \frac{2\sigma^2}{\epsilon^2} I(Z; D)$.*

The result above can be interpreted from two perspectives. Firstly, evaluating the predictor $w$ on randomly sampled test environments reflects its ability to generalize on average, since $L_{te}(w)$ is an unbiased estimate of $L(w)$. Secondly, knowledge about $L(w)$ can be used to predict the ability of $w$ to generalize on unseen domains, which complements Theorem 2 in solving Problem 1. The probability of generalization is mainly controlled by the distribution shift $I(Z; D)$, which can be further decomposed into the covariate shift and the concept shift. We then demonstrate that bounding the covariate shift $I(X; D)$ solely is sufficient to solve Problem 1 with Assumption 4:

**Theorem 5.** *If Assumption 4 holds and $\ell(f_w(X), f_{w'}(X))$ is $\sigma$-subgaussian w.r.t $X$ for any $w, w' \in \mathcal{W}$, then $\forall w \in \mathcal{W}$, $\Pr\{L_{te}(w) - L(w) \geq \epsilon + \min_{w^* \in \mathcal{W}}[L_{te}(w^*) + L(w^*)]\} \leq \frac{2\sigma^2}{\epsilon^2} I(X; D)$.*

This indicates that test-domain generalization is mainly controlled by the amount of covariate shift. When the hypothesis space $\mathcal{W}$ is large enough, the minimizer $w^* \in \mathcal{W}$ is expected to attain low-level population risks in both test environments $L_{te}(w^*)$ and average-case $L(w^*)$. However, it is

important to note that $w^*$ cannot reach zero average-case risk when concept shift is induced (i.e. $I(Y; D|X) > 0$) as indicated by Theorem 1. This implies an inevitable and strictly positive lower bound for the attainable risk level in test domains. Similarly, we further refine these test-domain generalization bounds by incorporating the more stringent Assumption 5 in Appendix D.

## 4   INTER-DOMAIN DISTRIBUTION MATCHING

In this section, we propose Inter-domain Distribution Matching (IDM) to achieve high-probability DG, which is highly motivated by our theoretical analysis in Section 3. Recall that the average-case risk $L(W)$ serves as a natural bridge to connect $L_{tr}(W)$ and $L_{te}(W)$, the regularization in Problem 1 directly indicates an objective for optimization by combining the high-probability concentration bounds in Theorem 2 and 4. Specifically, for any $\lambda \in (0, 1)$, we have:

$$\Pr\{|L_{te}(W) - L_{tr}(W)| \geq \epsilon\} \leq \frac{M^2}{\lambda^2 \epsilon^2} \left( \frac{1}{m} \sum_{i=1}^m I(W, D_i) + \log 3 \right) + \frac{2\sigma^2}{(1-\lambda)^2 \epsilon^2} I(Z; D). \quad (3)$$

This observation motivates us to minimize the input-output mutual information $I(W; D_i)$ and the distribution shift $I(Z; D)$ simultaneously to achieve high-probability DG, which further guides us to aligning inter-domain conditional distributions of gradients and representations respectively.

**Gradient Space Distribution Matching**
We first demonstrate that the minimization of $I(W; D_i)$ in equation (3) for each $i \in [1, m]$ can be achieved by matching the conditional distributions of inter-domain gradients. To see this, we assume that $W$ is optimized by some noisy and iterative learning algorithms, e.g. Stochastic Gradient Descent (SGD). Then the rule of updating $W$ at step $t$ through ERM can be formulated as:

$$W_t = W_{t-1} - \eta_t \sum_{i=1}^m g(W_{t-1}, B_t^i), \quad \text{where} \quad g(w, B_t^i) = \frac{1}{m|B_t^i|} \sum_{z \in B_t^i} \nabla_w \ell(f_w(x), y),$$

providing $W_0$ as the initial guess. Here, $\eta_t$ is the learning rate, and $B_t^i$ is the batch of data points randomly drawn from training environment $D_i$ and used to compute the direction for gradient descent. Suppose that algorithm $\mathcal{A}$ finishes in $T$ steps, we have the following upper bound for $I(W_T; D_i)$:

**Theorem 6.** *Let* $G_t = -\eta_t \sum_{i=1}^m g(W_{t-1}, B_t^i)$*, then* $I(W_T; D_i) \leq \sum_{t=1}^T I(G_t; D_i | W_{t-1})$*.*

Although our analysis is derived from the bare SGD algorithm, the same conclusion also applies to advanced techniques such as momentum and weight decay. Theorem 6 suggests that minimizing $I(G_t; D_i | W_{t-1})$ in each update step $t$ penalizes the input-output mutual information $I(W_T; D_i)$ and thus leads to training-domain generalization. This insight can also be verified by the Markov chain relationship $D_i \to \{G_t\}_{t=1}^T \to W_T$, which implies $I(W_T; D_i) \leq I(\{G_t\}_{t=1}^T; D_i)$ by the data-processing inequality. Notably, the mutual information $I(G_t; D_i | W_{t-1})$ can be rewritten as the KL divergence between marginal and conditional distributions of $G_t$ after observing $D_i$, which directly motivates matching the distribution of inter-domain gradients. Intuitively, gradient alignment enforces the model to learn common correlations shared across training domains, thus preventing overfitting to spurious features and promoting invariance (Shi et al., 2021; Rame et al., 2022).

**Representation Space Distribution Matching**
We now turn to test-domain generalization, which involves minimizing the distribution shift. Note that both $I(Z; D)$ and $I(X; D)$ are intrinsic properties of the data collection process, and thus cannot be penalized from the perspective of learning algorithms. Fortunately, the encoder $\phi$ can be considered as part of the data preprocessing procedure, enabling learning algorithms to minimize the representation space distribution shift. Similar to Theorem 4, we have that for any classifier $\psi$:

$$\Pr\{|L_{te}(\psi) - L(\psi)| \geq \epsilon\} \leq \frac{2\sigma^2}{\epsilon^2} I(R, Y; D), \qquad \text{if Assumption 3 holds.}$$

Let $P_{R,Y}$ be the joint distribution by pushing forward $P_Z$ via $R = f_\phi(X)$, where $\phi$ is some fixed encoder. We then have the following decomposition for the representation space distribution shift:

$$I(R, Y; D) \text{ (distribution shift)} = I(R; D) \text{ (covariate shift)} + I(Y; D|R) \text{ (concept shift)}.$$

This motivates us to simultaneously minimize the covariate shift and concept shift in the representation space. However, the concept shift is still intractable as shown by the following theorem:

**Theorem 7.** *For any $R$ satisfying $H(Y|R, D) \leq H(Y|X, D)$, we have $I(Y; D|R) \geq I(Y; D|X)$.*

That is, the representation space concept shift is lower-bounded by the original sample space concept shift when the representation is sufficient for prediction, characterizing the intrinsic trade-off between optimization and test-domain generalization. Consider that $I(Y; D|R)$ is hard to estimate when the number of classes is large, and recall that minimizing the covariate shift $I(R; D)$ solely is sufficient to solve Problem 1 (Theorem 5), we propose to penalize the encoder $\phi$ on minimizing $I(R_i; D_i)$ for each training domain $D_i \in D_{tr}$ to achieve test-domain generalization. Notably, the mutual information $I(R_i; D_i)$ is equivalent to the KL divergence $\mathrm{KL}(P_{R_i|D_i} \parallel P_{R_i})$, which directly motivates matching the conditional distributions of inter-domain representations.

**Per-sample Distribution Matching**

In this section, we propose Per-sample Distribution Matching (PDM) for inter-domain distribution alignment. Typically, learning algorithms have no knowledge about the underlying distribution of either the representation or the gradient, and the only available way is to align them across batched data points. The key limitation of such an approach is that when the number of samples (i.e. batch size) is limited, it is even impossible to distinguish different high-dimensional distributions:

**Theorem 8.** *(Informal) Let $n$ and $b$ be the dimension and the number of samples respectively. If $n > b + 1$, then there exist infinite environments whose conditional probabilities of an arbitrarily given group of samples are indistinguishable. If $n > 2b + 1$, then there exist infinite environments whose conditional probabilities cannot distinguish two arbitrarily given groups of samples.*

We refer the readers to Appendix C for a formal statement of Theorem 8. In real-world scenarios, the dimensionality of the feature or the gradient easily exceeds that of the batch size, making algorithms that aim to align the entire distribution (e.g. CORAL (Sun & Saenko, 2016) and MMD (Li et al., 2018b)) generally ineffective since distribution alignment is basically impossible given such few data points. This observation is also verified by Rame et al. (2022) that aligning the entire covariance matrix achieves no better performance than aligning the diagonal elements only. Furthermore, prior distribution alignment techniques mainly focus on aligning the directions (Parascandolo et al., 2020; Shahtalebi et al., 2021; Shi et al., 2021) or low-order moments (Sun & Saenko, 2016; Koyama & Yamaguchi, 2020; Rame et al., 2022), which are insufficient for complex probability distributions. For example, while the standard Gaussian distribution $N(0, 1)$ and the uniform distribution $U(-\sqrt{3}, \sqrt{3})$ share the same expectation and variance, they are fundamentally different. To address these issues, we propose PDM for distribution matching in a per-dimension manner, by minimizing an upper bound of the KL divergence between probability density estimators.

Let $\{x_i^1\}_{i=1}^b$ and $\{x_i^2\}_{i=1}^b$ be two groups of 1-dimensional data points drawn from probability distributions $P$ and $Q$ respectively. Let $p_i$ denote the PDF of the Gaussian distribution with expectation $x_i^1$ and variance $\sigma^2$, then the probability density estimator $\bar{P}$ of $P$ can be written as $\bar{p}(x) = \frac{1}{b} \sum_i p_i(x)$ (respectively for $q_i$, $\bar{Q}$ and $\bar{q}$). The following theorem suggests a computable upper bound for the KL divergence or the Wasserstein distance between probability density estimators:

**Theorem 9.** *Let $f$ be a bijection: $[1, b] \leftrightarrow [1, b]$ and $P_i$ $(Q_i)$ be the probability measure defined by PDF $p_i$ $(q_i)$, then $\mathrm{KL}(\bar{P} \parallel \bar{Q}) \leq \frac{1}{b} \sum_{i=1}^b \mathrm{KL}(P_i \parallel Q_{f(i)})$, and $\mathbb{W}(\bar{P}, \bar{Q}) \leq \frac{1}{b} \sum_{i=1}^b \mathbb{W}(P_i, Q_{f(i)})$.*

Hence, distribution matching can be achieved by minimizing the KL divergence or Wasserstein distances between point Gaussian densities, which is equivalent to aligning individual data points. The following theorem suggests an optimal bijection for choosing the order of alignment:

**Theorem 10.** *$f(j) = j$ is the minimizer of both $\sum_{i=1}^b \mathrm{KL}(P_i \parallel Q_{f(i)})$ and $\sum_{i=1}^b \mathbb{W}(P_i, Q_{f(i)})$ when $\{x_i^1\}_{i=1}^b$ and $\{x_i^2\}_{i=1}^b$ are sorted in the same order.*

To summarize, the main procedure of PDM is to divide the data points into separate dimensions, sort the data points in ascending (or descending) order in each dimension, and then match the sorted data points across different training domains. PDM improves over previous distribution matching techniques by simultaneously capturing multiple orders of moments, avoiding ineffective high-dimensional distribution matching, and enabling straightforward implementation and efficient computation. We provide pseudo-codes for both PDM and IDM in Appendix E for better comprehension.

It is noteworthy that this per-dimension sorting and matching scheme exhibits similarities with the computation of sliced Wasserstein distance (Kolouri et al., 2019; Deshpande et al., 2019; Dai & Seljak, 2021). This further validates the effectiveness of PDM by demonstrating its applicability to probability distributions beyond density estimators. Nevertheless, it should be noted that sliced

Wasserstein distance is not directly applicable to Problem 1, as the generalization error cannot be unequivocally bounded by Wasserstein distance metrics without the stringent Lipschitz condition. In contrast, our analysis based on KL divergence only necessitates the mild Subgaussian assumption.

**Algorithm Design**

Combining the methods discussed above, we finally propose the IDM algorithm for high-probability DG by simultaneously aligning inter-domain distributions of the gradients and the representations. Recall that Problem 1 incorporates an additional regularization based on ERM, we adopt the following Lagrange multipliers to optimize the IDM objective:

$$\mathcal{L}_{\text{IDM}} = \mathcal{L}_{\text{E}} + \lambda_1 \mathcal{L}_{\text{G}} + \lambda_2 \mathcal{L}_{\text{R}} = \frac{1}{m} \sum_{i=1}^{m} \left[ L'_{D_i}(W) + \lambda_1 \text{PDM}(G_i) + \lambda_2 \text{PDM}(R_i) \right]. \quad (4)$$

Here $\mathcal{L}_{\text{E}}$ is the risk of ERM, $\mathcal{L}_{\text{G}}$ and $\mathcal{L}_{\text{R}}$ denote the penalty of inter-domain distribution alignment for the gradient and the representation respectively, implemented with the proposed PDM method. To cooperate representation alignment which regards the classifier $\psi$ as the true predictor and also for memory and time concerns, we only apply gradient alignment for the classifier $\psi$ as in (Rame et al., 2022). Furthermore, $\lambda_1$ and $\lambda_2$ should be adaptively chosen according to the amount of covariate and concept shifts respectively: Firstly, $I(R; D)$ is upper bounded by $I(X; D)$ by the Markov chain $D \to X \to R$, so the representations are naturally aligned when $I(X; D) = 0$. Secondly, gradient alignment is not required when $I(Y; D|X) = 0$, since the entire distribution shift can then be minimized by aligning the representations solely. Therefore, the Lagrange multipliers $\lambda_1$ and $\lambda_2$ should scale with the amount of the covariate and concept shifts respectively.

## 5 RELATED WORKS

In the literature, various approaches for DG have been proposed by incorporating external domain information to achieve OOD generalization. Most recent works achieve invariance by employing additional regularization criteria based on ERM. These methods differ in the choice of the statistics used to match across training domains and can be categorized by the corresponding objective of 1) gradient, 2) representation, and 3) predictor, as follows:

**Invariant Gradients.** Gradient alignment enforces batch data points from different domains to cooperate and has been employed in OOD generalization by finding minima in the loss landscape that are shared across training domains. Specifically, IGA (Koyama & Yamaguchi, 2020) proposed to align the empirical expectations, Fish (Shi et al., 2021) suggested aligning the directions of gradients, AND-mask (Parascandolo et al., 2020) and SAND-mask (Shahtalebi et al., 2021) only update weights when the gradients share the same direction, and Fishr (Rame et al., 2022) proposes matching the gradient variance. The key limitation of these gradient-based methods is their coarse alignment of either the directions or low-order moments, resulting in substantial information loss in more granular statistics. Our work is the first to connect gradient alignment and training-domain generalization, providing insights into how gradient matching enhances generalization.

**Invariant Representations.** Extracting domain-invariant features has been extensively studied to solve both DG and domain adaptation (DA) problems. DANN (Ganin et al., 2016) and CDANN (Li et al., 2018c) align inter-domain representations via adversarial methods, MMD (Li et al., 2018b) uses kernel methods for distribution alignment, and CORAL (Sun & Saenko, 2016) achieves invariance by matching low-order moments of the representations. Still, these methods are insufficient for complex probability distributions (Zhao et al., 2019), ineffective for high-dimensional distributions (Theorem 8), and incapable to address the concept shift (Theorem 7). Our theoretical results provide an information-theoretic perspective for test-domain generalization by showing how representation alignment minimizes the variance of target-domain risks, and thus minimizes the gap between training and test-domain population risks with high probability.

**Invariant Predictors.** A recent line of works proposes to explore the connection between invariance and causality. IRM (Arjovsky et al., 2019) and subsequent works (Zhou et al., 2022; Lin et al., 2022) learn an invariant classifier that is simultaneously optimal for all training domains. However, later works have shown that IRM may fail on non-linear data and lead to sub-optimal predictors (Kamath et al., 2021; Ahuja et al., 2021). Parallel works include: V-REx (Krueger et al., 2021) which minimizes the difference between training-domain risks, GroupDRO (Sagawa et al., 2019) which minimizes the worst-domain training risk, and QRM (Eastwood et al., 2022) which optimizes the quantile of the risk distribution across training domains. We will show that IDM also promotes domain-invariant predictors and ensures optimality across different training domains.

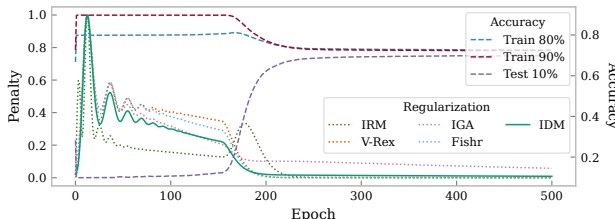

Figure 1: Learning dynamics of IDM.

Table 1: The Colored MNIST task.

| Method | Train Acc | Test Acc | Gray Acc |
|--------|-----------|----------|----------|
| ERM | $86.4 \pm 0.2$ | $14.0 \pm 0.7$ | $71.0 \pm 0.7$ |
| IRM | $71.0 \pm 0.5$ | $65.6 \pm 1.8$ | $66.1 \pm 0.2$ |
| V-REx | $71.7 \pm 1.5$ | $67.2 \pm 1.5$ | $68.6 \pm 2.2$ |
| IGA | $68.9 \pm 3.0$ | $67.7 \pm 2.9$ | $67.5 \pm 2.7$ |
| Fishr | $69.6 \pm 0.9$ | $71.2 \pm 1.1$ | $70.2 \pm 0.7$ |
| IDM | $70.2 \pm 1.4$ | $70.6 \pm 0.9$ | $70.5 \pm 0.7$ |

## 6 EXPERIMENTAL RESULTS

In this section, we evaluate the proposed IDM algorithm on Colored MNIST (Arjovsky et al., 2019) and DomainBed (Gulrajani & Lopez-Paz, 2020) to demonstrate its capability of generalizing against different types of distribution shifts[1]. Detailed experimental settings and further empirical results including ablation studies are reported in Appendix E and F.

The **Colored MNIST** task proposed by Arjovsky et al. (2019) is carefully designed to create high correlations between image colors and the true labels, leading to spurious features that possess superior predictive power ($90\%$ and $80\%$ accuracy) over the actual digits ($75\%$). However, this correlation is reversed in the test domain ($10\%$), causing any learning algorithm that solely minimizes training-domain errors to overfit the color information and fail during test time. As such, Colored MNIST is an ideal task to evaluate the capability of learning algorithms to achieve invariance across source domains and generalize well on target environments.

We follow the experimental settings in (Arjovsky et al., 2019) and adopt a two-stage training technique, where the penalty strength $\lambda$ is set low initially and set higher afterward. We visualize the learning dynamics of relevant DG penalties, including IRM, V-Rex, IGA, and Fishr, using the IDM objective for optimization in Figure 1. The penalty values are normalized for better clarity. This visualization confirms Theorem 2 that IDM promotes source-domain generalization by minimizing the gap between training risks, thus ensuring the optimality of the predictor across different training domains. Moreover, the learning dynamics verify the superiority of PDM by showing that penalizing the IDM objective solely is sufficient to minimize other types of invariance penalties, including IRM/V-REx which ensures the optimality of the classifier across training domains, and IGA/Fishr which aligns the expectation/variance of inter-domain gradient distributions.

Table 1 presents the mean accuracy and standard deviation of different DG algorithms on Colored MNIST across 10 independent runs. Notably, the Oracle model (ERM trained with gray-scale images) achieves $71.0\%$ accuracy on both source and target domains. Following the hyper-parameter tuning technique as Arjovsky et al. (2019), we select the best model by $\max_w \min(L_{tr}(w), L_{te}(w))$. IDM achieves the best trade-off between training and test-domain accuracies ($70.2\%$), and near-optimal gray-scale accuracy ($70.5\%$) compared to the Oracle predictor ($71.0\%$).

The **DomainBed Benchmark** (Gulrajani & Lopez-Paz, 2020) comprises of synthetic datasets (Colored MNIST (Arjovsky et al., 2019) and Rotated MNIST (Ghifary et al., 2015)) and also various real-world datasets (VLCS (Fang et al., 2013), PACS (Li et al., 2017), OfficeHome (Venkateswara et al., 2017), TerraIncognita (Beery et al., 2018), and DomainNet (Peng et al., 2019)) for assessing the performance of both DA and DG algorithms. To ensure a fair comparison, DomainBed limits the number of attempts for hyper-parameter tuning to 20, and the results are averaged over 3 independent trials. Therefore, DomainBed serves as a rigorous and comprehensive benchmark to evaluate different DG strategies. We compare the performance of our method with 20 baselines in total for a thorough evaluation. Table 2 summarizes the results using test-domain model selection, i.e. the validation set follows the same distribution as test domains. Although training-domain validation is more common, we argue that this would result in sub-optimal selection results as indicated by Theorem 1, and discuss the viability of test-domain selection in Appendix F.3.

As can be seen, IDM achieves top-1 accuracy ($72.0\%$) on CMNIST which is competitive with the Oracle ($75.0\%$). This verifies the superiority of the proposed PDM method by outperforming all

---

[1]The source code is provided along with the supplementary materials.

Table 2: The DomainBed benchmark. We format **best**, second best and worse than ERM results.

| Algorithm | Accuracy (↑) | | | | | | | | Ranking (↓) | | |
|---|---|---|---|---|---|---|---|---|---|---|---|
| | CMNIST | RMNIST | VLCS | PACS | OffHome | TerraInc | DomNet | Avg | Mean | Median | Worst |
| ERM | $57.8 \pm 0.2$ | $97.8 \pm 0.1$ | $77.6 \pm 0.3$ | $86.7 \pm 0.3$ | $66.4 \pm 0.5$ | $53.0 \pm 0.3$ | $41.3 \pm 0.1$ | 68.7 | 12.3 | 11 | 20 |
| IRM | $67.7 \pm 1.2$ | $97.5 \pm 0.2$ | $76.9 \pm 0.6$ | $84.5 \pm 1.1$ | $63.0 \pm 2.7$ | $50.5 \pm 0.7$ | $28.0 \pm 5.1$ | 66.9 | 18.3 | 20 | 22 |
| GroupDRO | $61.1 \pm 0.9$ | $97.9 \pm 0.1$ | $77.4 \pm 0.5$ | $87.1 \pm 0.1$ | $66.2 \pm 0.6$ | $52.4 \pm 0.1$ | $33.4 \pm 0.3$ | 67.9 | 11.7 | 10 | 19 |
| Mixup | $58.4 \pm 0.2$ | $98.0 \pm 0.1$ | $78.1 \pm 0.3$ | $86.8 \pm 0.3$ | $68.0 \pm 0.2$ | $\mathbf{54.4} \pm 0.3$ | $39.6 \pm 0.1$ | 69.0 | 7.3 | 6 | 15 |
| MLDG | $58.2 \pm 0.4$ | $97.8 \pm 0.1$ | $77.5 \pm 0.1$ | $86.8 \pm 0.4$ | $66.6 \pm 0.3$ | $52.0 \pm 0.1$ | $41.6 \pm 0.1$ | 68.7 | 12.6 | 13 | 18 |
| CORAL | $58.6 \pm 0.5$ | $98.0 \pm 0.0$ | $77.7 \pm 0.2$ | $87.1 \pm 0.5$ | $\mathbf{68.4} \pm 0.2$ | $52.8 \pm 0.2$ | $41.8 \pm 0.1$ | 69.2 | 6.4 | 5 | 14 |
| MMD | $63.3 \pm 1.3$ | $98.0 \pm 0.1$ | $77.9 \pm 0.1$ | $87.2 \pm 0.1$ | $66.2 \pm 0.3$ | $52.0 \pm 0.4$ | $23.5 \pm 9.4$ | 66.9 | 10.0 | 10 | 22 |
| DANN | $57.0 \pm 1.0$ | $97.9 \pm 0.1$ | $\underline{79.7} \pm 0.5$ | $85.2 \pm 0.2$ | $65.3 \pm 0.8$ | $50.6 \pm 0.4$ | $38.3 \pm 0.1$ | 67.7 | 15.0 | 18 | 22 |
| CDANN | $59.5 \pm 2.0$ | $97.9 \pm 0.0$ | $\mathbf{79.9} \pm 0.2$ | $85.8 \pm 0.8$ | $65.3 \pm 0.5$ | $50.8 \pm 0.6$ | $38.5 \pm 0.2$ | 68.2 | 12.4 | 14 | 18 |
| MTL | $57.6 \pm 0.3$ | $97.9 \pm 0.1$ | $77.7 \pm 0.5$ | $86.7 \pm 0.2$ | $66.5 \pm 0.4$ | $52.2 \pm 0.4$ | $40.8 \pm 0.1$ | 68.5 | 11.7 | 10 | 21 |
| SagNet | $58.2 \pm 0.3$ | $97.9 \pm 0.0$ | $77.6 \pm 0.1$ | $86.4 \pm 0.4$ | $67.5 \pm 0.2$ | $52.5 \pm 0.4$ | $40.8 \pm 0.2$ | 68.7 | 11.3 | 9 | 17 |
| ARM | $63.2 \pm 0.7$ | $\mathbf{98.1} \pm 0.1$ | $77.8 \pm 0.3$ | $85.8 \pm 0.2$ | $64.8 \pm 0.4$ | $51.2 \pm 0.5$ | $36.0 \pm 0.2$ | 68.1 | 13.0 | 16 | 21 |
| VREx | $67.0 \pm 1.3$ | $97.9 \pm 0.1$ | $78.1 \pm 0.2$ | $87.2 \pm 0.6$ | $65.7 \pm 0.3$ | $51.4 \pm 0.5$ | $30.1 \pm 3.7$ | 68.2 | 10.6 | 8 | 20 |
| RSC | $58.5 \pm 0.5$ | $97.6 \pm 0.1$ | $77.8 \pm 0.6$ | $86.2 \pm 0.5$ | $66.5 \pm 0.6$ | $52.1 \pm 0.2$ | $38.9 \pm 0.6$ | 68.2 | 13.4 | 13 | 19 |
| AND-mask | $58.6 \pm 0.4$ | $97.5 \pm 0.0$ | $76.4 \pm 0.4$ | $86.4 \pm 0.4$ | $66.1 \pm 0.2$ | $49.8 \pm 0.4$ | $37.9 \pm 0.6$ | 67.5 | 17.0 | 16 | 22 |
| SAND-mask | $62.3 \pm 1.0$ | $97.4 \pm 0.1$ | $76.2 \pm 0.5$ | $85.9 \pm 0.4$ | $65.9 \pm 0.5$ | $50.2 \pm 0.1$ | $32.2 \pm 0.6$ | 67.2 | 17.9 | 19 | 22 |
| Fish | $61.8 \pm 0.8$ | $97.9 \pm 0.1$ | $77.8 \pm 0.6$ | $85.8 \pm 0.6$ | $66.0 \pm 2.9$ | $50.8 \pm 0.4$ | $\mathbf{43.4} \pm 0.3$ | 69.1 | 11.3 | 11 | 18 |
| Fishr | $\underline{68.8} \pm 1.4$ | $97.8 \pm 0.1$ | $78.2 \pm 0.2$ | $86.9 \pm 0.2$ | $68.2 \pm 0.2$ | $\underline{53.6} \pm 0.4$ | $41.8 \pm 0.2$ | 70.8 | 5.4 | **3** | 16 |
| SelfReg | $58.0 \pm 0.7$ | $\mathbf{98.1} \pm 0.7$ | $78.2 \pm 0.1$ | $\mathbf{87.7} \pm 0.1$ | $68.1 \pm 0.3$ | $52.8 \pm 0.9$ | $\underline{43.1} \pm 0.1$ | 69.4 | 5.0 | **3** | 19 |
| CausIRL$_{CORAL}$ | $58.4 \pm 0.3$ | $98.0 \pm 0.2$ | $78.2 \pm 0.1$ | $\underline{87.6} \pm 0.1$ | $67.7 \pm 0.2$ | $53.4 \pm 0.4$ | $42.1 \pm 0.1$ | 69.4 | 5.0 | **3** | 15 |
| CausIRL$_{MMD}$ | $63.7 \pm 0.8$ | $97.9 \pm 0.1$ | $78.1 \pm 0.1$ | $86.6 \pm 0.7$ | $65.2 \pm 0.6$ | $52.2 \pm 0.3$ | $40.6 \pm 0.2$ | 69.2 | 10.4 | 10 | 20 |
| IDM | $\mathbf{72.0} \pm 1.0$ | $98.0 \pm 0.1$ | $78.1 \pm 0.4$ | $\underline{87.6} \pm 0.3$ | $\underline{68.3} \pm 0.2$ | $52.8 \pm 0.5$ | $41.8 \pm 0.2$ | **71.2** | **3.3** | **3** | **6** |

previous distribution alignment techniques by aligning the directions (AND-mask, SAND-mask, Fish) or low-order moments (Fishr). On the contrary, algorithms that only align the representations (CORAL, MMD, DANN, CDANN) are incapable of addressing the concept shift, thus performing poorly on CMNIST. Moreover, IDM achieves the highest accuracy among all distribution matching algorithms on RMNIST / PACS, competitive performances to the best algorithm on RMNIST (98.0% v.s. 98.1%), PACS (87.6% v.s. 87.7%), OfficeHome (68.3% v.s. 68.4%), the highest average accuracy (71.2% v.s. 70.8%) and rankings (mean, median and worst ranks on 7 datasets) among all baseline methods. IDM also enables efficient implementation, such that the computational overhead compared to ERM is only $5\%$ on the largest DomainNet dataset, and negligible for other smaller datasets. Notably, IDM is the only algorithm that consistently achieves top rankings (6 v.s. 14), while any other method failed to outperform most of the competitors on at least 1 dataset.

While the results are promising, we notice that IDM is not very effective on TerraIncognita. There are several possible reasons for such a phenomenon: Firstly, the number of hyper-parameters in IDM exceeds most competing methods, which is critical to model selection since the number of tuning attempts is limited in DomainBed. Recall that the value of $\lambda_1$ and $\lambda_2$ should adapt to the amount of covariate and concept shifts respectively: While CMNIST manually induces high concept shift, covariate shift is instead dominant in other datasets, raising extra challenges for hyper-parameter tuning. Secondly, representation space distribution alignment may not always help since $L_{te}(w) \leq L(w)$ is possible by the randomized nature of target domains. These factors together result in sub-optimal hyper-parameter selection results.

## 7 CONCLUSION AND DISCUSSION

In this paper, we conduct information-theoretic generalization analysis upon a probabilistic formulation for DG and provide novel upper bounds for both source and target-domain generalization errors. Our theoretical results successfully explain the elegant performance of distribution-matching DG methods and inspire us to design the IDM algorithm by simultaneously aligning inter-domain gradients and representations. Combining with the proposed PDM method by aligning individual sorted data points, we achieve superior performance on the DomainBed benchmark.

While our analysis does not necessitate the independence condition between source domains or target domains, such a condition is also naturally satisfied in most learning scenarios and leads to tighter generalization bounds. Specifically, $\sum_{i=1}^{m} I(W; D_i) \leq I(W; D_{tr})$ is satisfied in Theorem 2 when the training domains are i.i.d sampled, elsewise we only have $I(W; D_i) \leq I(W; D_{tr})$ for any $i \in [1, m]$. Moreover, Theorem 4 and 5 can be further tightened by a factor of $\frac{1}{m'}$ when test domains are i.i.d, such that one can expect better generalization performance by increasing the number of test domains. We refer the readers to Appendix B and C for the proof of these results.

**Reproducibility Statement**. To ensure reproducibility, we include complete proofs of our theoretical results in Appendix B and C, detailed explanations of our experimental settings in Appendix E, and source codes in the supplementary materials.

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

# Appendix

## Table of Contents

## A  PREREQUISITE DEFINITIONS AND LEMMAS

**Definition 1.** *(Subgaussian) A random variable $X$ is $\sigma$-subgaussian if for any $\rho \in \mathbb{R}$, $\mathbb{E}[\exp(\rho(X - \mathbb{E}[X]))] \leq \exp(\rho^2 \sigma^2 / 2)$.*

**Definition 2.** *(Kullback-Leibler Divergence) Let $P$ and $Q$ be probability measures on the same space $\mathcal{X}$, the KL divergence from $P$ to $Q$ is defined as $\mathrm{KL}(P \parallel Q) \triangleq \int_{\mathcal{X}} p(x) \log(p(x)/q(x)) \, \mathrm{d}x$.*

**Definition 3.** *(Mutual Information) Let $(X, Y)$ be a pair of random variables with values over the space $\mathcal{X} \times \mathcal{Y}$. Let their joint distribution be $P_{X,Y}$ and the marginal distributions be $P_X$ and $P_Y$ respectively, the mutual information between $X$ and $Y$ is defined as $I(X; Y) = \mathrm{KL}(P_{X,Y} \parallel P_X P_Y)$.*

**Definition 4.** *(Wasserstein Distance) Let $\mathrm{c}(\cdot, \cdot)$ be a metric and let $P$ and $Q$ be probability measures on $\mathcal{X}$. Denote $\Gamma(P, Q)$ as the set of all couplings of $P$ and $Q$ (i.e. the set of all joint distributions on $\mathcal{X} \times \mathcal{X}$ with two marginals being $P$ and $Q$), then the Wasserstein distance of order $p$ between $P$ and $Q$ is defined as $\mathbb{W}_p(P, Q) \triangleq \left( \inf_{\gamma \in \Gamma(P,Q)} \int_{\mathcal{X} \times \mathcal{X}} c(x, x')^p \, \mathrm{d}\gamma(x, x') \right)^{1/p}$.*

Unless otherwise noted, we use $\log$ to denote the logarithmic function with base $e$, and use $\mathbb{W}(\cdot, \cdot)$ to denote the Wasserstein distance of order 1.

**Definition 5.** *(Total Variation) The total variation between two probability measures $P$ and $Q$ is* $\mathrm{TV}(P, Q) \triangleq \sup_E |P(E) - Q(E)|$*, where the supremum is over all measurable set $E$.*

**Lemma 1.** *(Lemma 1 in (Harutyunyan et al., 2021)) Let $(X, Y)$ be a pair of random variables with joint distribution $P_{X,Y}$ and let $\bar{Y}$ be an independent copy of $Y$. If $f(x, y)$ is a measurable function such that $E_{X,Y}[f(X, Y)]$ exists and $f(X, \bar{Y})$ is $\sigma$-subgaussian, then*

$$\left| \mathbb{E}_{X,Y}[f(X, Y)] - \mathbb{E}_{X,\bar{Y}}[f(X, \bar{Y})] \right| \leq \sqrt{2\sigma^2 I(X; Y)}.$$

*Furthermore, if $f(x, Y)$ is $\sigma$-subgaussian for each $x$ and the expectation below exists, then*

$$\mathbb{E}_{X,Y}\left[ \left( f(X, Y) - \mathbb{E}_{\bar{Y}}[f(X, \bar{Y})] \right)^2 \right] \leq 4\sigma^2 (I(X; Y) + \log 3),$$

*and for any $\epsilon > 0$, we have*

$$\Pr\left\{ \left| f(X, Y) - \mathbb{E}_{\bar{Y}}[f(X, \bar{Y})] \right| \geq \epsilon \right\} \leq \frac{4\sigma^2 (I(X; Y) + \log 3)}{\epsilon^2}.$$

**Lemma 2.** *(Lemma 2 in (Harutyunyan et al., 2021)) Let $X$ be $\sigma$-subgaussian and $\mathbb{E}[X] = 0$, then for any $\lambda \in [0, 1/4\sigma^2)$:*

$$\mathbb{E}_X\left[ e^{\lambda X^2} \right] \leq 1 + 8\lambda\sigma^2.$$

**Lemma 3.** *(Donsker-Varadhan formula) Let $P$ and $Q$ be probability measures defined on the same measurable space, where $P$ is absolutely continuous with respect to $Q$. Then*

$$\mathrm{KL}(P \parallel Q) = \sup_X \left\{ \mathbb{E}_P[X] - \log \mathbb{E}_Q[e^X] \right\},$$

*where $X$ is any random variable such that $e^X$ is $Q$-integrable and $\mathbb{E}_P[X]$ exists.*

**Lemma 4.** *Let $P$, and $Q$ be probability measures defined on the same measurable space. Let $X \sim P$ and $X' \sim Q$. If $f(X)$ is $\sigma$-subgaussian w.r.t $X$ and the following expectations exists, then*

$$\left| \mathbb{E}_{X'}[f(X')] - \mathbb{E}_X[f(X)] \right| \leq \sqrt{2\sigma^2 \mathrm{KL}(Q \parallel P)},$$

$$\mathbb{E}_{X'}\left[ \left( f(X') - \mathbb{E}_X[f(X)] \right)^2 \right] \leq 4\sigma^2 (\mathrm{KL}(Q \parallel P) + \log 3).$$

*Furthermore, by combining the results above and Markov's inequality, we have that for any $\epsilon > 0$:*

$$\Pr\{ |f(X') - \mathbb{E}_X[f(X)]| \geq \epsilon \} \leq \frac{4\sigma^2}{\epsilon^2} (\mathrm{KL}(Q \parallel P) + \log 3).$$

*Proof.* Let $\lambda \in \mathbb{R}$ be any non-zero constant, then by the subgaussian property of $f(X)$:

$$\log \mathbb{E}_X\left[ e^{\lambda(f(X) - \mathbb{E}_X[f(X)])} \right] \leq \frac{\lambda^2 \sigma^2}{2},$$

$$\log \mathbb{E}_X\left[ e^{\lambda f(X)} \right] - \lambda \mathbb{E}_X[f(X)] \leq \frac{\lambda^2 \sigma^2}{2}.$$

By applying Lemma 3 with $X = \lambda f(X)$ we have

$$\mathrm{KL}(Q \parallel P) \geq \sup_\lambda \left\{ \mathbb{E}_{X'}[\lambda f(X')] - \log \mathbb{E}_X\left[ e^{\lambda f(X)} \right] \right\}$$

$$\geq \sup_\lambda \left\{ \mathbb{E}_{X'}[\lambda f(X')] - \lambda \mathbb{E}_X[f(X)] - \frac{\lambda^2 \sigma^2}{2} \right\}$$

$$= \frac{1}{2\sigma^2} (\mathbb{E}_{X'}[f(X')] - \mathbb{E}_X[f(X)])^2,$$

where the supremum is taken by setting $\lambda = \frac{1}{\sigma^2} (\mathbb{E}_{X'}[f(X')] - \mathbb{E}_X[f(X)])$. This completes the proof of the first inequality.

To prove the second inequality, let $g(x) = (f(x) - \mathbb{E}_X[f(X)])^2$ and $\lambda \in [0, 1/4\sigma^2)$. Apply Lemma 3 again with $X = \lambda g(X)$, we have

$$
\begin{aligned}
\mathrm{KL}(Q \parallel P) &\geq \sup_\lambda \left\{ \mathbb{E}_{X'}[\lambda g(X')] - \log \mathbb{E}_X \left[ e^{\lambda g(X)} \right] \right\} \\
&= \sup_\lambda \left\{ \mathbb{E}_{X'} \left[ \lambda (f(X') - \mathbb{E}_X[f(X)])^2 \right] - \log \mathbb{E}_X \left[ e^{\lambda (f(X) - \mathbb{E}_X[f(X)])^2} \right] \right\} \\
&\geq \sup_\lambda \left\{ \mathbb{E}_{X'} \left[ \lambda (f(X') - \mathbb{E}_X[f(X)])^2 \right] - \log(1 + 8\lambda\sigma^2) \right\} \\
&\geq \frac{1}{4\sigma^2} \mathbb{E}_{X'} \left[ (f(X') - \mathbb{E}_X[f(X)])^2 \right] - \log 3,
\end{aligned}
$$

where the second inequality follows by applying Lemma 2 and the last inequality follows by taking $\lambda \to \frac{1}{4\sigma^2}$. This finishes the proof of the second inequality.

Furthermore, by applying Markov's inequality, we can get:

$$
\begin{aligned}
\Pr\{|f(X') - \mathbb{E}_X[f(X)]| \geq \epsilon\} &= \Pr\left\{ (f(X') - \mathbb{E}_X[f(X)])^2 \geq \epsilon^2 \right\} \\
&\leq \frac{1}{\epsilon^2} \mathbb{E}_{X'} \left[ (f(X') - \mathbb{E}_X[f(X)])^2 \right] \\
&\leq \frac{4\sigma^2}{\epsilon^2} (\mathrm{KL}(Q \parallel P) + \log 3),
\end{aligned}
$$

which completes the proof. $\qquad\square$

**Lemma 5.** *(Kantorovich-Rubinstein Duality) Let $P$ and $Q$ be probability measures defined on the same measurable space $\mathcal{X}$, then*

$$
\mathbb{W}(P, Q) = \sup_{f \in Lip_1} \left\{ \int_{\mathcal{X}} f \, \mathrm{d}P - \int_{\mathcal{X}} f \, \mathrm{d}Q \right\},
$$

*where $Lip_1$ denotes the set of 1-Lipschitz functions in the metric $c$, i.e. $|f(x) - f(x')| \leq c(x, x')$ for any $f \in Lip_1$ and $x, x' \in \mathcal{X}$.*

**Lemma 6.** *(Pinsker's Inequality) Let $P$ and $Q$ be probability measures defined on the same space, then $\mathrm{TV}(P, Q) \leq \sqrt{\frac{1}{2}\mathrm{KL}(Q \parallel P)}$.*

**Lemma 7.** *For any constant $\lambda \in (0, 1)$, we have*

$$
\begin{aligned}
\Pr\{|L_{tr}(W) - L_{te}(W)| \geq \epsilon\} &\leq \Pr\{|L_{tr}(W) - L(W)| \geq \lambda\epsilon\} \\
&\quad + \Pr\{|L_{te}(W) - L(W)| \geq (1-\lambda)\epsilon\}.
\end{aligned}
$$

*Proof.* Notice that $|L_{tr}(W) - L(W)| \leq \lambda\epsilon$ and $|L_{te}(W) - L(W)| \leq (1-\lambda)\epsilon$ together implies $|L_{tr}(W) - L_{te}(W)| \leq \epsilon$, we then have

$$
\Pr\{|L_{tr}(W) - L_{te}(W)| \leq \epsilon\} \geq \Pr\{|L_{tr}(W) - L(W)| \leq \lambda\epsilon \bigcap |L_{te}(W) - L(W)| \leq (1-\lambda)\epsilon\}.
$$

This implies that

$$
\Pr\{|L_{tr}(W) - L_{te}(W)| \geq \epsilon\} \leq \Pr\{|L_{tr}(W) - L(W)| \geq \lambda\epsilon \bigcup |L_{te}(W) - L(W)| \geq (1-\lambda)\epsilon\}.
$$

By applying Boole's inequality, we then have

$$
\begin{aligned}
\Pr\{|L_{tr}(W) - L_{te}(W)| \geq \epsilon\} &\leq \Pr\{|L_{tr}(W) - L(W)| \geq \lambda\epsilon\} \\
&\quad + \Pr\{|L_{te}(W) - L(W)| \geq (1-\lambda)\epsilon\}.
\end{aligned}
$$

$\qquad\square$

## B  Omitted Proofs in Section 3

### B.1  Proof of Theorem 1

**Theorem 1** (Restate). *Let $Q_{R|X}$ and $Q_{Y|R}$ denote the encoder and classifier characterized by $f_\phi$ and $f_\psi$ respectively, and let $Q_{Y|X}$ denote the whole model by $Q_{Y|X} = \int_{\mathcal{R}} Q_{Y|R} \, dQ_{R|X}$. Then for any domain $d \in \mathcal{D}$:*

$$\mathrm{KL}(P_{Y|X,D} \,\|\, Q_{Y|X}) \geq I(Y; D|X),$$
$$\mathrm{KL}(P_{Y|X,D=d} \,\|\, Q_{Y|X}) \leq I(X; Y|R, D = d) + \mathrm{KL}(P_{Y|R,D=d} \,\|\, Q_{Y|R}).$$

*Proof.* For better clarity, we abbreviate $P_X(X)$ as $P_X$ in the following expectations for any random variable $X$.

$$
\begin{aligned}
\mathrm{KL}(P_{Y|X,D} \,\|\, Q_{Y|X}) &= \mathbb{E}_{D,X,Y}\left[\log \frac{P_{Y|X,D}}{Q_{Y|X}}\right] \\
&= \mathbb{E}_{D,X,Y}\left[\log \frac{P_{Y|X,D}}{P_{Y|X}} \cdot \frac{P_{Y|X}}{Q_{Y|X}}\right] \\
&= \mathbb{E}_{D,X,Y}\left[\log \frac{P_{Y,D|X}}{P_{Y|X} P_{D|X}}\right] + \mathbb{E}_{X,Y}\left[\log \frac{P_{Y|X}}{Q_{Y|X}}\right] \\
&= I(Y; D|X) + \mathrm{KL}(P_{Y|X} \,\|\, Q_{Y|X}) \geq I(Y; D|X).
\end{aligned}
$$

The last inequality is by the positiveness of the KL divergence. It holds with equality if and only if $Q_{Y|X} = P_{Y|X}$.

To prove the second inequality, we apply Jensen's inequality on the concave logarithmic function:

$$
\begin{aligned}
\mathrm{KL}(P_{Y|X,D=d} \,\|\, Q_{Y|X}) &= \mathbb{E}_{X,Y|D=d}\left[\log \frac{P_{Y|X,D=d}}{Q_{Y|X}}\right] \\
&= \mathbb{E}_{X,Y|D=d}\left[\log \frac{P_{Y|X,D=d}}{\mathbb{E}_{R|X=x}[Q_{Y|R}]}\right] \\
&\leq \mathbb{E}_{X,Y|D=d}\mathbb{E}_{R|X=x}\left[\log \frac{P_{Y|X,D=d}}{Q_{Y|R}}\right] \\
&= \mathbb{E}_{X,Y,R|D=d}\left[\log \frac{P_{Y|X,D=d}}{P_{Y|R,D=d}} \cdot \frac{P_{Y|R,D=d}}{Q_{Y|R}}\right] \\
&= \mathbb{E}_{X,Y,R|D=d}\left[\log \frac{P_{Y,X|R,D=d}}{P_{Y|R,D=d} P_{X|R,D=d}}\right] + \mathbb{E}_{Y,R|D=d}\left[\log \frac{P_{Y|R,D=d}}{Q_{Y|R}}\right] \\
&= I(X; Y|R, D = d) + \mathrm{KL}(P_{Y|R,D=d} \,\|\, Q_{Y|R}).
\end{aligned}
$$

The only inequality holds with equality when $\mathrm{Var}[Q_{R|X=x}] = 0$ for any $x \in \mathcal{X}$, i.e. $f_\phi$ is deterministic. This completes the proof. $\qquad\square$

### B.2  Proof of Theorem 2

**Theorem 2** (Restate). *Let $W$ be the output of learning algorithm $\mathcal{A}$ with training domains $D_{tr}$. If Assumption 2 holds, then*

$$|\mathbb{E}_{W,D_{tr}}[L_{tr}(W)] - \mathbb{E}_W[L(W)]| \leq \frac{1}{m}\sum_{i=1}^{m}\sqrt{\frac{M^2}{2}I(W, D_i)},$$

$$\Pr\{|L_{tr}(W) - L(W)| \geq \epsilon\} \leq \frac{M^2}{\epsilon^2}\left(\frac{1}{m}\sum_{i=1}^{m}I(W, D_i) + \log 3\right).$$

*Proof.* For any $D \in D_{tr}$, let $\bar{D}$ be an independent copy of the marginal distribution of $D$. Then by setting $X = W$, $Y = D$ and $f(w, d) = L_d(w)$ in Lemma 1, we have

$$|\mathbb{E}_{W,D}[L_D(W)] - \mathbb{E}_W[L(W)]| = |\mathbb{E}_{W,D}[L_D(W)] - \mathbb{E}_{W,\bar{D}}[L_{\bar{D}}(W)]|$$

$$\leq \sqrt{\frac{M^2}{2}I(W,D)}.$$

$$\mathbb{E}_{W,D}[(L_D(W) - L(W))^2] \leq M^2(I(W,D) + \log 3).$$

By summing up each individual training domain, we can get

$$
\left| \mathbb{E}_{W,D_{tr}}[L_{tr}(W)] - \mathbb{E}_W[L(W)] \right| = \left| \frac{1}{m}\mathbb{E}_{W,D_{tr}}\left[ \sum_{i=1}^m L_{D_i}(W) \right] - \mathbb{E}_W[L(W)] \right|
$$

$$
\leq \frac{1}{m}\sum_{i=1}^m \left| \mathbb{E}_{W,D_i}[L_{D_i}(W)] - \mathbb{E}_W[L(W)] \right|
$$

$$
\leq \frac{1}{m}\sum_{i=1}^m \sqrt{\frac{M^2}{2}I(W,D_i)}
$$

$$
\leq \sqrt{\frac{1}{m}\sum_{i=1}^m \frac{M^2}{2}I(W,D_i)}.
$$

where the last inequality is by applying Jensen's inequality on the concave square root function. Similarly, we have

$$
\mathbb{E}_{W,D_{tr}}[(L_{tr}(W) - L(W))^2] = \mathbb{E}_{W,D_{tr}}\left[ \left( \frac{1}{m}\sum_{i=1}^m L_{D_i}(W) - L(W) \right)^2 \right]
$$

$$
\leq \frac{1}{m}\sum_{i=1}^m \mathbb{E}_{W,D_i}\left[ (L_{D_i}(W) - L(W))^2 \right]
$$

$$
\leq \frac{1}{m}\sum_{i=1}^m M^2(I(W,D_i) + \log 3)
$$

$$
\leq \frac{M^2}{m}\left( \sum_{i=1}^m I(W,D_i) + m\log 3 \right),
$$

which further implies

$$
\Pr\{|L_{tr}(W) - L(W)| \geq \epsilon\} \leq \frac{M^2}{m\epsilon^2}\left( \sum_{i=1}^m I(W,D_i) + m\log 3 \right)
$$

$$
= \frac{M^2}{\epsilon^2}\left( \frac{1}{m}\sum_{i=1}^m I(W,D_i) + \log 3 \right)
$$

by Lemma 1.

Additionally, by assuming that the training domains are independent, we have

$$
\begin{aligned}
I(W;D_{tr}) &= I(W;\{D_i\}_{i=1}^m) = I(W;D_1) + I(W;\{D_i\}_{i=2}^m|D_1) \\
&= I(W;D_1) + I(W;\{D_i\}_{i=2}^m) - I(\{D_i\}_{i=2}^m;D_1) + I(\{D_i\}_{i=2}^m;D_1|W) \\
&= I(W;D_1) + I(W;\{D_i\}_{i=2}^m) + I(\{D_i\}_{i=2}^m;D_1|W) \\
&\geq I(W;D_1) + I(W;\{D_i\}_{i=2}^m) \\
&\geq \cdots \\
&\geq \sum_{i=1}^m I(W,D_i).
\end{aligned}
$$

$\square$

### B.3 PROOF OF THEOREM 3

**Theorem 3** (Restate). *Let $W$ be the output of learning algorithm $\mathcal{A}$ under training domains $D_{tr}$. If $\ell(f_w(X), Y)$ is $\beta'$-Lipschitz w.r.t $w$, i.e. $|\ell(f_{w_1}(X), Y) - \ell(f_{w_2}(X), Y)| \leq \beta' c(w_1, w_2)$ for any $w_1, w_2 \in \mathcal{W}$, then*

$$|\mathbb{E}_{W, D_{tr}}[L_{tr}(W)] - \mathbb{E}_W[L(W)]| \leq \frac{\beta'}{m} \sum_{i=1}^{m} \mathbb{E}_{D_i}[\mathbb{W}(P_{W|D_i=d}, P_W)].$$

*Proof.* For any $d \in \mathcal{D}$ and $D_i \in D_{tr}$, let $P = P_{W|D_i=d}$, $Q = P_W$ and $f(w) = L_{D_i}(w)$ in Lemma 5, then

$$\begin{aligned}
|\mathbb{E}_{W, D_{tr}}[L_{tr}(W)] - \mathbb{E}_W[L(W)]| &\leq \frac{1}{m} \mathbb{E}_{D_{tr}} \left[ \sum_{i=1}^{m} |\mathbb{E}_{W|D_i=d}[L_{D_i}(W)] - \mathbb{E}_W[L(W)]| \right] \\
&= \frac{1}{m} \mathbb{E}_{D_{tr}} \left[ \sum_{i=1}^{m} |\mathbb{E}_{W|D_i=d}[L_{D_i}(W)] - \mathbb{E}_W[L_{D_i}(W)]| \right] \\
&\leq \frac{1}{m} \mathbb{E}_{D_{tr}} \left[ \sum_{i=1}^{m} \beta' \mathbb{W}(P_{W|D_i=d}, P_W) \right] \\
&= \frac{\beta'}{m} \sum_{i=1}^{m} \mathbb{E}_{D_i}[\mathbb{W}(P_{W|D_i=d}, P_W)].
\end{aligned}$$

When the metric $d$ is discrete, the Wasserstein distance is equal to the total variation. Combining with Lemma 6, we have the following reductions:

$$\begin{aligned}
\mathbb{E}_{D_i}[\mathbb{W}(P_{W|D_i=d}, P_W)] &= \mathbb{E}_{D_i}[\text{TV}(P_{W|D_i=d}, P_W)] \\
&\leq \mathbb{E}_{D_i} \left[ \sqrt{\frac{1}{2} \text{KL}(P_{W|D_i=d} \parallel P_W)} \right] \\
&\leq \sqrt{\frac{1}{2} I(W; D_i)},
\end{aligned}$$

where the last inequality follows by applying Jensen's inequality on the concave square root function. $\square$

### B.4 PROOF OF THEOREM 4

**Theorem 4** (Restate). *For any $w \in \mathcal{W}$, $\mathbb{E}_{D_{te}}[L_{te}(w)] = L(w)$. Additionally if Assumption 3 holds, then*

$$\Pr\{|L_{te}(w) - L(w)| \geq \epsilon\} \leq \frac{2\sigma^2}{\epsilon^2} I(Z; D).$$

*Furthermore, when the test domains are independent, we have*

$$\Pr\{|L_{te}(w) - L(w)| \geq \epsilon\} \leq \frac{2\sigma^2}{m'\epsilon^2} I(Z; D).$$

*Proof.* By the identical marginal distribution of the test domains $\mathcal{D}_{te} = \{D_k\}_{k=1}^{m'}$, we have

$$\begin{aligned}
\mathbb{E}_{D_{te}}[L_{te}(w)] &= \frac{1}{m'} \sum_{k=1}^{m'} \mathbb{E}_{D_k}[L_{D_k}(w)] = \frac{1}{m'} \sum_{k=1}^{m'} \mathbb{E}_D[L_D(w)] \\
&= \mathbb{E}_D[L_D(w)] = \mathbb{E}_D \mathbb{E}_{Z|D=d}[\ell(f_w(X), Y)] \\
&= \mathbb{E}_{D, Z}[\ell(f_w(X), Y)] = \mathbb{E}_Z[\ell(f_w(X), Y)] = L(w).
\end{aligned}$$

If Assumption 3 holds, then for any $d \in \mathcal{D}$, by setting $P = P_Z$, $Q = P_{Z|D=d}$ and $f(z) = \ell(f_w(x), y)$ in Lemma 4, we have

$$(L_d(w) - L(w))^2 = \left( \mathbb{E}_{Z|D=d}[\ell(f_w(X), Y)] - \mathbb{E}_Z[\ell(f_w(X), Y)] \right)^2$$

$$\leq \left(\sqrt{2\sigma^2 \mathrm{KL}(P_{Z|D=d} \parallel P_Z)}\right)^2$$

$$\leq 2\sigma^2 \mathrm{KL}(P_{Z|D=d} \parallel P_Z).$$

Taking the expectation over $D \sim \nu$, we can get

$$\mathbb{E}_D[(L_D(w) - L(w))^2] \leq \mathbb{E}_D\left[2\sigma^2 \mathrm{KL}(P_{Z|D=d} \parallel P_Z)\right]$$
$$= 2\sigma^2 \mathrm{KL}(P_{Z|D} \parallel P_Z).$$

Furthermore, notice that

$$\mathrm{KL}(P_{Z|D} \parallel P_Z) = \mathbb{E}_D \mathbb{E}_{Z|D=d}\left[\log \frac{P_{Z|D}}{P_Z}\right]$$

$$= \mathbb{E}_{D,Z}\left[\log \frac{P_{Z,D}}{P_Z P_D}\right]$$

$$= I(Z; D).$$

Combining the results above yields the following inequality:

$$\mathbb{E}_D[(L_D(w) - L(w))^2] \leq 2\sigma^2 I(Z; D).$$

When the test domains $\{D_k\}_{i=k}^{m'}$ are independent of each other, they can be regarded as i.i.d copies of $D$, i.e.

$$\mathrm{Var}_{D_{te}}[L_{te}(w)] = \frac{1}{m'^2} \sum_{k=1}^{m'} \mathrm{Var}_{D_k}[L_{D_k}(w)]$$

$$= \frac{1}{m'^2} \sum_{k=1}^{m'} \mathbb{E}_{D_k}[(L_{D_k}(w) - L(w))^2]$$

$$\leq \frac{1}{m'^2} \sum_{k=1}^{m'} 2\sigma^2 I(Z; D)$$

$$= \frac{1}{m'} 2\sigma^2 I(Z; D).$$

Finally, by applying Chebyshev's inequality, we can prove that

$$\Pr\{|L_{te}(w) - L(w)| \geq \epsilon\} \leq \frac{2\sigma^2}{m'\epsilon^2} I(Z; D).$$

Otherwise, when the independent condition is not satisfied, we have that by applying Jensen's inequality:

$$\mathrm{Var}_{D_{te}}[L_{te}(w)] = \mathbb{E}_{D_{te}}\left[\left(\frac{1}{m'} \sum_{k=1}^{m'} L_{D_k}(w) - L(w)\right)^2\right]$$

$$\leq \frac{1}{m'} \sum_{k=1}^{m'} \mathbb{E}_{D_k}[(L_{D_k}(w) - L(w))^2]$$

$$\leq \frac{1}{m'} \sum_{k=1}^{m'} 2\sigma^2 I(Z; D)$$

$$= 2\sigma^2 I(Z; D).$$

One can see that this differs from the independent case above by a factor of $1/m'$. □

## B.5 PROOF OF THEOREM 5

**Theorem 5** (Restate). *If Assumption 4 holds, and $\ell(f_w(X), f_{w'}(X))$ is $\sigma$-subgaussian w.r.t $X \sim P_X$ for any $w, w' \in \mathcal{W}$, then for any $w \in \mathcal{W}$,*

$$\Pr\{L_{te}(w) - L(w) \geq \epsilon + L^*\} \leq \frac{2\sigma^2}{\epsilon^2} I(X; D),$$

*where $L^* = \min_{w^* \in \mathcal{W}}(L_{te}(w^*) + L(w^*))$. Additionally, if the test domains are independent, then*

$$\Pr\{L_{te}(w) - L(w) \geq \epsilon + L^*\} \leq \frac{2\sigma^2}{m'\epsilon^2} I(X; D),$$

*Proof.* For any environment $d \in \mathcal{D}$ and $w, w' \in \mathcal{W}$, denote

$$L_d(w, w') = \mathbb{E}_{X|D=d}[\ell(f_w(X), f_{w'}(X))], \quad \text{and}$$
$$L(w, w') = \mathbb{E}_X[\ell(f_w(X), f_{w'}(X))].$$

By setting $P = P_X$, $Q = P_{X|D=d}$ and $f(x) = \ell(f_w(x), f_{w'}(x))$ in Lemma 4, we have

$$(L_d(w, w') - L(w, w'))^2 \leq 2\sigma^2 \mathrm{KL}(P_{X|D=d} \parallel P_X).$$

By taking the expectation over $D \sim \nu$, we get

$$\mathbb{E}_D\left[(L_d(w, w') - L(w, w'))^2\right] \leq 2\sigma^2 \mathrm{KL}(P_{X|D} \parallel P_X)$$
$$= 2\sigma^2 I(X; D).$$

Through a similar procedure of proving Theorem 4, we have

$$\Pr\{|L_{te}(w, w') - L(w, w')| \geq \epsilon\} \leq \frac{2\sigma^2}{m'\epsilon^2} I(X; D).$$

If Assumption 4 holds, then for any $w, w^* \in \mathcal{W}$, we can prove that

$$L_d(w) = \mathbb{E}_{Z|D=d}[\ell(f_w(X), Y)]$$
$$\leq \mathbb{E}_{Z|D=d}[\ell(f_w(X), f_{w^*}(X)) + \ell(f_{w^*}(X), Y)]$$
$$= L_d(w, w^*) + L_d(w^*).$$

Additionally, for any environment $d \in \mathcal{D}$,

$$L_d(w, w^*) = \mathbb{E}_{X|D=d}[\ell(f_w(X), f_{w^*}(X))]$$
$$\leq \mathbb{E}_{Z|D=d}[\ell(f_w(X), Y) + \ell(Y, f_{w^*}(X))]$$
$$= L_d(w) + L_d(w^*).$$

Combining the results above, we have that with probability at least $1 - \frac{2\sigma^2}{m'\epsilon^2} I(X; D)$,

$$L_{te}(w) \leq L_{te}(w, w^*) + L_{te}(w^*)$$
$$\leq L(w, w^*) + \epsilon + L_{te}(w^*)$$
$$\leq L(w) + L(w^*) + \epsilon + L_{te}(w^*).$$

By minimizing $L_{te}(w^*) + L(w^*)$ over $w^* \in \mathcal{W}$, it follows that

$$\Pr\left\{L_{te}(w) - L(w) \geq \epsilon + \min_{w^* \in \mathcal{W}}(L_{te}(w^*) + L(w^*))\right\} \leq \frac{2\sigma^2}{m'\epsilon^2} I(X; D),$$

which finishes the proof. $\qquad\square$

## C  OMITTED PROOFS IN SECTION 4

### C.1  PROOF OF THEOREM 6

**Theorem 6** (Restate). *Let $G_t^i = -\eta_t g(W_{t-1}, B_t^i)$ and $G_t = \sum_{i=1}^m G_t^i$, then*

$$I(W_T; D_i) \leq \sum_{t=1}^T I(G_t; D_i | W_{t-1}).$$

*Additionally, if the training domains are independent, then*

$$I(W_T; D_i) \leq \sum_{t=1}^T I(G_t^i; D_i | W_{t-1}).$$

*Proof.* Noticing the Markov chain relationship $D_i \to (W_{T-1}, G_T) \to W_{T-1} + G_T$, then by the data processing inequality

$$
\begin{aligned}
I(W_T; D_i) &= I(W_{T-1} + G_T; D_i) \\
&\leq I(W_{T-1}, G_T; D_i) \\
&= I(W_{T-1}; D_i) + I(G_T; D_i | W_{T-1}).
\end{aligned}
$$

where the last equality is by the chain rule of conditional mutual information. By applying the reduction steps above recursively, we can get

$$
\begin{aligned}
I(W_T; D_i) &\leq I(W_{T-1}; D_i) + I(G_T; D_i | W_{T-1}) \\
&\leq I(W_{T-2}; D_i) + I(G_{T-1}; D_i | W_{T-2}) \\
&\quad + I(G_T; D_i | W_{T-1}) \\
&\leq \cdots \\
&\leq \sum_{t=1}^{T} I(G_t; D_i | W_{t-1}).
\end{aligned}
$$

When the training domains are independent, we additionally have

$$
\begin{aligned}
I(W_T; D_i) &\leq I(W_{T-1}; D_i) + I(G_T; D_i | W_{T-1}) \\
&\leq I(W_{T-1}; D_i) + I\big(\{G_T^k\}_{k=1}^m; D_i | W_{T-1}\big) \\
&= I(W_{T-1}; D_i) + I\big(G_T^i; D_i | W_{T-1}\big) + I\big(\{G_T^k\}_{k=1}^m \setminus G_T^i; D_i | W_{T-1}, G_T^i\big) \\
&= I(W_{T-1}; D_i) + I\big(G_T^i; D_i | W_{T-1}\big).
\end{aligned}
$$

Then by the same scheme of recursive reduction, we can prove that

$$
I(W_T; D_i) \leq \sum_{t=1}^{T} I\big(G_t^i; D_i | W_{t-1}\big).
$$

$\square$

## C.2 PROOF OF THEOREM 7

**Theorem 7** (Restate). *Assume the Markov chain relationship $Y \to X \to R$, then for any sufficient $R$ satisfying $H(Y|R, D) \leq H(Y|X, D)$, we have*

$$
I(Y; D|R) \geq I(Y; D|X).
$$

*Proof.* From the Markov chain $Y \to X \to R$, we know that

$$
\begin{aligned}
I(Y; X) - I(Y; R) \geq 0 &\geq H(Y|R, D) - H(Y|X, D) \\
&= H(Y|D) - H(Y|X, D) - (H(Y|D) - H(Y|R, D)) \\
&= I(Y; X|D) - I(Y; R|D).
\end{aligned}
$$

By the definition of interaction information (denoted as $I(\cdot; \cdot; \cdot)$), we have

$$
I(Y; X; D) = I(Y; X) - I(Y; X|D) \geq I(Y; R) - I(Y; R|D) = I(Y; R; D),
$$

which implies

$$
I(Y; D|R) = I(Y; D) - I(Y; R; D) \geq I(Y; D) - I(Y; X; D) = I(Y; D|X).
$$

The proof is complete. $\square$

## C.3 PROOF OF THEOREM 8

**Theorem 8** (Formal). *Let $n$ be the dimensionality of the distribution and $b$ be the number of data points, then*

- If $n > b + 1$, *then for any sampled data points* $s = \{x_i\}_{i=1}^b$, *there exists infinite environments* $d_1$, $d_2$, $\cdots$ *such that* $p(S = s|D = d_1) = p(S = s|D = d_2) = \cdots$.

- If $n > 2b + 1$, *then for any two batch of sampled data points* $s_1 = \{x_i^1\}_{i=1}^b$ *and* $s_2 = \{x_i^2\}_{i=1}^b$, *there exists infinite environments* $d_1$, $d_2$, $\cdots$ *such that for each* $j \in [1, \infty)$, $p(S = s_1|D = d_j) = p(S = s_2|D = d_j)$.

*Proof.* Without loss of generality, we assume that the data-generating distributions $p(X|D)$ are Gaussian with zero means for simplicity, i.e.

$$p(x|D = d) = \frac{1}{\sqrt{(2\pi)^n|\Sigma_d|}} \exp\left(-\frac{1}{2}x^\top \Sigma_d x\right),$$

where $\Sigma_d$ is the corresponding covariance matrix of environment $d$. Let $X \in \mathbb{R}^{n \times b}$ be the data matrix of $S$ such that the $i$-th column of $X$ equals $x_i$, we then have

$$p(S = s|D = d) = \frac{1}{\sqrt{(2\pi)^{bn}|\Sigma_d|^b}} \exp\left(-\frac{1}{2}\text{tr}(X^\top \Sigma_d X)\right).$$

Since the rank of $X$ is at most $b$, one can decompose $\Sigma_d = \Sigma_d^1 + \Sigma_d^2$ with $\text{rank}(\Sigma_d^1) = b$ and $\text{rank}(\Sigma_d^2) = n - b \geq 2$ through eigenvalue decomposition, and let the eigenvector space of $\Sigma_d^1$ cover the column space of $X$. Then we have

$$\text{tr}(X^\top \Sigma_d^1 X) = \text{tr}(X^\top \Sigma_d X), \quad \text{and} \quad \text{tr}(X^\top \Sigma_d^2 X) = 0.$$

Therefore, one can arbitrarily modify the eigenvector space of $\Sigma_d^2$ as long as keeping it orthogonal to that of $\Sigma_d^1$, without changing the value of $\text{tr}(X^\top \Sigma_d X)$. This finishes the proof of the first part.

To prove the second part, similarly we decompose $\Sigma_d$ by $\Sigma_d^1 + \Sigma_d^2$ such that $\text{rank}(\Sigma_d^1) = 2b + 1$ and $\text{rank}(\Sigma_d^2) = n - 2b - 1 \geq 1$, and make the eigenvector space of $\Sigma_d^1$ cover the column space of both $X_1$ and $X_2$, where $X_1$ and $X_2$ are the data matrix of $S_1$ and $S_2$ respectively. We then have

$$\text{tr}(X_1^\top \Sigma_d^1 X_1) = \text{tr}(X_1^\top \Sigma_d X_1),$$
$$\text{tr}(X_2^\top \Sigma_d^1 X_2) = \text{tr}(X_2^\top \Sigma_d X_2),$$
$$\text{and} \quad \text{tr}(X_1^\top \Sigma_d^2 X_1) = \text{tr}(X_2^\top \Sigma_d^2 X_2) = 0.$$

Let $\Sigma_d^1 = U_d^\top \Lambda_d U_d$ be the eigenvalue decomposition of $\Sigma_d^1$, where $U_d \in \mathbb{R}^{(2b+1) \times n}$ and $\Lambda_d = \text{diag}(\lambda_1^d, \cdots, \lambda_{2b+1}^d)$. Notice that for any $x \in \mathbb{R}^n$, we have $x^\top \Sigma_d x = (U_d x)^\top \Lambda_d (U_d x) = \sum_{i=1}^{2b+1}(U_d x)_i^2 \lambda_i$. By assuming that $p(S = s_1|D = d) = p(S = s_2|D = d)$, we have the following homogeneous linear equations:

$$a_1^1 \lambda_1 + a_1^2 \lambda_2 + \cdots + a_1^{2b+1} \lambda_{2b+1} = 0,$$
$$a_2^1 \lambda_1 + a_2^2 \lambda_2 + \cdots + a_2^{2b+1} \lambda_{2b+1} = 0,$$
$$\cdots$$
$$a_b^1 \lambda_1 + a_b^2 \lambda_2 + \cdots + a_b^{2b+1} \lambda_{2b+1} = 0,$$

where $a_i^j = (U_d x_i^1)_j^2 - (U_d x_i^2)_j^2$. Since $2b + 1 > b$, the linear system above has infinite non-zero solutions, which finishes the proof of the second part. $\square$

## C.4 PROOF OF THEOREM 9

**Theorem 9** (Restate). *Let $f$ be a bijection mapping from $[1, b]$ to $[1, b]$, then* $\text{KL}(\bar{P} \parallel \bar{Q}) \leq \frac{1}{b}\sum_{i=1}^b \text{KL}(P_i \parallel Q_{f(i)})$, *and* $\mathbb{W}(\bar{P}, \bar{Q}) \leq \frac{1}{b}\sum_{i=1}^b \mathbb{W}(P_i, Q_{f(i)})$, *where $P_i$ is the probability measure defined by $p_i$ (respectively for $Q_i$).*

*Proof.* Recall that $\bar{p}(x) = \frac{1}{b}\sum_{i=1}^b p_i(x)$ and $\bar{q}(x) = \frac{1}{b}\sum_{i=1}^b q_i(x)$, we then have

$$\text{KL}(\bar{P} \parallel \bar{Q}) = \int_{\mathcal{X}} \bar{p}(x) \log\left(\frac{\bar{p}(x)}{\bar{q}(x)}\right) \mathrm{d}x$$

$$
\begin{aligned}
&= -\int_{\mathcal{X}} \bar{p}(x) \log\left( \frac{1}{b} \sum_{i=1}^{b} \frac{p_i(x)}{\bar{p}(x)} \cdot \frac{q_{f(i)}(x)}{p_i(x)} \right) \mathrm{d}x \\
&\leq -\int_{\mathcal{X}} \bar{p}(x) \frac{1}{b} \sum_{i=1}^{b} \frac{p_i(x)}{\bar{p}(x)} \log\left( \frac{q_{f(i)}(x)}{p_i(x)} \right) \mathrm{d}x \\
&= -\frac{1}{b} \sum_{i=1}^{b} \int_{\mathcal{X}} p_i(x) \log\left( \frac{q_{f(i)}(x)}{p_i(x)} \right) \mathrm{d}x \\
&= \frac{1}{b} \sum_{i=1}^{b} \mathrm{KL}(P_i \parallel Q_{f(i)}),
\end{aligned}
$$

where the only inequality follows by applying Jensen's inequality on the concave logarithmic function. This finishes the proof of the upper bound for KL divergence.

To prove the counterpart for Wasserstein distance, we apply Lemma 5 on $\bar{P}$ and $\bar{Q}$:

$$
\begin{aligned}
\mathbb{W}(\bar{P}, \bar{Q}) &= \sup_{f \in Lip_1} \left\{ \int_{\mathcal{X}} f \, \mathrm{d}\bar{P} - \int_{\mathcal{X}} f \, \mathrm{d}\bar{Q} \right\} \\
&= \sup_{f \in Lip_1} \left\{ \int_{\mathcal{X}} f \, \mathrm{d}\left( \frac{1}{b} \sum_{i=1}^{b} P_i \right) - \int_{\mathcal{X}} f \, \mathrm{d}\left( \frac{1}{b} \sum_{i=1}^{b} Q_{f(i)} \right) \right\} \\
&\leq \frac{1}{b} \sum_{i=1}^{b} \sup_{f \in Lip_1} \left\{ \int_{\mathcal{X}} f \, \mathrm{d}P_i - \int_{\mathcal{X}} f \, \mathrm{d}Q_{f(i)} \right\} \\
&= \frac{1}{b} \sum_{i=1}^{b} \mathbb{W}(P_i, Q_{f(i)}).
\end{aligned}
$$

The proof is complete. $\qquad\square$

## C.5 PROOF OF THEOREM 10

**Theorem 10** (Restate). *Suppose that $\{x_i^1\}_{i=1}^b$, $\{x_i^2\}_{i=1}^b$ are sorted in the same order, then $f(j) = j$ is the minimizer of $\sum_{i=1}^b \mathrm{KL}(P_i \parallel Q_{f(i)})$ and $\sum_{i=1}^b \mathbb{W}(P_i, Q_{f(i)})$.*

*Proof.* For simplicity, we assume that all data points of $\{x_i^1\}_{i=1}^b$ and $\{x_i^2\}_{i=1}^b$ are different from each other. Since $P_i$ and $Q_i$ are Gaussian distributions with the same variance, the KL divergence and Wasserstein distance between them could be analytically acquired:

$$
\mathrm{KL}(P_i \parallel Q_j) = \frac{(x_i^1 - x_j^2)^2}{2\sigma^2}, \quad \text{and} \quad \mathbb{W}(P_i, Q_j) = |x_i^1 - x_j^2|.
$$

Suppose there exists $i \in [1, b]$ such that $f(i) \neq i$. Without loss of generality, we assume that $f(i) > i$. Then by the pigeonhole principle, there exists $j \in (i, b]$ that satisfies $f(j) < f(i)$. Suppose that $\{x_i^1\}_{i=1}^b$, $\{x_i^2\}_{i=1}^b$ are both sorted in ascending order, we have $x_i^1 < x_j^1$ and $x_{f(i)}^2 > x_{f(j)}^2$. For any $p \in \{1, 2\}$, the following 3 cases cover all possible equivalent combinations of the order of $x_i^1$, $x_j^1$, $x_{f(j)}^2$ and $x_{f(i)}^2$:

- When $x_i^1 < x_j^1 < x_{f(j)}^2 < x_{f(i)}^2$ and $p = 2$, we have

$$
\begin{aligned}
&(x_i^1 - x_{f(i)}^2)^2 + (x_j^1 - x_{f(j)}^2)^2 - (x_i^1 - x_{f(j)}^2)^2 - (x_j^1 - x_{f(i)}^2)^2 \\
&= (2x_i^1 - x_{f(i)}^2 - x_{f(j)}^2)(x_{f(j)}^2 - x_{f(i)}^2) - (2x_j^1 - x_{f(j)}^2 - x_{f(i)}^2)(x_{f(j)}^2 - x_{f(i)}^2) \\
&= (x_{f(j)}^2 - x_{f(i)}^2)(2x_i^1 - 2x_j^1) > 0.
\end{aligned}
$$

Elsewise when $p = 1$, we have

$$
|x_i^1 - x_{f(i)}^2| + |x_j^1 - x_{f(j)}^2| = |x_i^1 - x_{f(j)}^2| + |x_j^1 - x_{f(i)}^2|.
$$

- When $x_i^1 < x_{f(j)}^2 < x_j^1 < x_{f(i)}^2$, we have

$$|x_i^1 - x_{f(i)}^2|^p > |x_i^1 - x_{f(j)}^2|^p + |x_j^1 - x_{f(i)}^2|^p.$$

- When $x_i^1 < x_{f(j)}^2 < x_{f(i)}^2 < x_j^1$, we have

$$
\begin{aligned}
|x_i^1 - x_{f(i)}^2|^p + |x_j^1 - x_{f(j)}^2|^p &\geq |x_i^1 - x_{f(j)}^2|^p + |x_{f(i)}^2 - x_{f(j)}^2|^p \\
&\quad + |x_j^1 - x_{f(i)}^2|^p + |x_{f(i)}^2 - x_{f(j)}^2|^p \\
&> |x_i^1 - x_{f(j)}^2|^p + |x_j^1 - x_{f(i)}^2|^p.
\end{aligned}
$$

In conclusion, under all possible circumstances, we have $|x_i^1 - x_{f(i)}^2|^p + |x_j^1 - x_{f(j)}^2|^p \geq |x_i^1 - x_{f(j)}^2|^p + |x_j^1 - x_{f(i)}^2|^p$, which implies that by setting $f'(i) = f(j)$, $f'(j) = f(i)$ and $f'(k) = f(k)$ for $k \notin \{i, j\}$, $f'$ will be a better choice over $f$ to minimize $\mathrm{KL}(\bar{P} \parallel \bar{Q})$ or $\mathbb{W}(\bar{P}, \bar{Q})$. The proof is complete since the existence of a minimizer is obvious. □

## D  FURTHER DISCUSSIONS

### D.1  LEVERAGING THE INDEPENDENCE ASSUMPTION

In the main text, we only assume that the test domains are independent of the training domains, while the training domains are not necessarily independent of each other (same for test domains). This assumption is much weaker than i.i.d. by allowing correlations between training domains, e.g. sampling from a finite set without replacement. While this weaker assumption is preferable, we highlight that we can tighten the previous generalization bounds if the independence assumption is incorporated, such that the training and test domains are i.i.d sampled.

Firstly, when the training domains satisfy the i.i.d condition, we prove in Theorem 13 that $\sum_{i=1}^{m} I(W; D_i) \leq I(W; D_{tr})$. Otherwise, we can only prove that for any $i \in [1, m]$, $I(W; D_i) \leq I(W; D_{tr})$. This indicates that while the model can achieve source-domain generalization by letting $I(W; D_i) \to 0$, it can still learn from the training domain set $D_{tr}$. Notably, having $I(W; D_i) = 0$ for each $i \in [1, m]$ does not mean that the model learns nothing from the training domains, i.e. $I(W; D_{tr})$ can still be positive. To see this, we take $D_i$ as i.i.d random binary variables such that $\Pr(D_i = 0) = \Pr(D_i = 1) = \frac{1}{2}$, and let $W = D_1 \oplus \cdots \oplus D_m$, where $\oplus$ is the XOR operator. Then it is easy to verify that $W$ is independent of each $D_i$ since $P_{W|D_i} = P_W$, implying $I(W; D_i) = 0$. However, $I(W; D_{tr}) = H(W)$ is strictly positive.

Next, the test-domain generalization bounds in Theorem 4 and 5 can be further tightened by a factor of $1/m'$ when the i.i.d condition of test domains is incorporated. Therefore, one can now guarantee better generalization by increasing the number of domains, which is consistent with real-world observations.

### D.2  HIGH-PROBABILITY PROBLEM FORMULATION

Another high-probability formulation of the DG problem is presented by Eastwood et al. (2022), namely Quantile Risk Minimization (QRM). Under our notations, the QRM objective can be expressed as:

$$\min_w \epsilon \quad s.t. \quad \Pr\{L_{te}(w) \geq \epsilon\} \leq \delta.$$

The main difference between our formulation in Problem 1 and QRM is that we not only consider the randomness of $D_{te}$, but also those of $D_{tr}$ and $W$. This randomized nature of training domains and the hypothesis serve as the foundation of our information-theoretic generalization analysis. When the training-domain risks have been observed, i.e. $L_{tr}(W)$ is fixed, our formulation reduces to QRM. The main advantage of our formulation is that it could be directly optimized by learning algorithms without further assumptions or simplifications. On the contrary, Eastwood et al. (2022) needs to further adopt kernel density estimation to approximate the quantile of the risks and transform the QRM problem to the empirical one (EQRM). Further advantages of our formulation include:

- Kernel density estimation required by QRM is challenging when the number of training domains is not sufficiently large. For comparison, IDM could be easily applied as long as there are at least 2 training domains.

- Our formulation aims to find the optimal learning algorithm instead of the optimal hypothesis. This would be essential to analyze the correlations between the hypothesis $W$ and training domains $D_{tr}$, and also is more suitable in robust learning settings when measuring the error bar.

- Our formulation directly characterizes the trade-off between optimization and generalization, which is the main challenge to achieve invariance across different domains (Arjovsky et al., 2019).

### D.3 TIGHTER BOUNDS FOR TARGET-DOMAIN POPULATION RISK

In a similar vein, we provide the following two upper bounds for test-domain generalization error in terms of Wasserstein distances. For the following analysis, we assume the independence between training and test domains.

**Theorem 11.** *If Assumption 5 holds, then for any $w \in \mathcal{W}$,*

$$\Pr\{|L_{te}(w) - L(w)| \geq \epsilon\} \leq \frac{\beta^2}{m'\epsilon^2} \mathbb{E}_D[\mathbb{W}^2(P_{Z|D=d}, P_Z)].$$

*Furthermore, if the metric $d$ is discrete, then*

$$\Pr\{|L_{te}(w) - L(w)| \geq \epsilon\} \leq \frac{\beta^2}{2m'\epsilon^2} I(Z; D).$$

*Proof.* For any $d \in \mathcal{D}$, by setting $P = P_{Z|D=d}$, $Q = P_Z$ and $f(z) = \frac{1}{\beta}\ell(f_w(x), y)$ in Lemma 5, we have

$$(L_d(w) - L(w))^2 \leq \beta^2 \mathbb{W}^2(P_{Z|D=d}, P_Z).$$

Following a similar procedure with the proof of Theorem 4, we have

$$\Pr\{|L_{te}(w) - L(w)| \geq \epsilon\} \leq \frac{\beta^2}{m'\epsilon^2} \mathbb{E}_D[\mathbb{W}^2(P_{Z|D=d}, P_Z)].$$

When the metric $d$ is discrete, Wasserstein distance is equivalent to the total variation. Therefore

$$\begin{aligned}
\Pr\{|L_{te}(w) - L(w)| \geq \epsilon\} &\leq \frac{\beta^2}{m'\epsilon^2} \mathbb{E}_D[\mathrm{TV}^2(P_{Z|D=d}, P_Z)] \\
&\leq \frac{\beta^2}{m'\epsilon^2} \mathbb{E}_D\left[\frac{1}{2}\mathrm{KL}(P_{Z|D=d} \| P_Z)\right] \\
&= \frac{\beta^2}{2m'\epsilon^2} I(Z; D),
\end{aligned}$$

where the second inequality is by applying Lemma 6. The proof is complete. $\square$

**Theorem 12.** *If Assumption 4 holds, and $\ell(f_w(X), f_{w'}(X))$ is $\beta$-Lipschitz for any $w, w' \in \mathcal{W}$, then for any $w \in \mathcal{W}$*

$$\Pr\{L_{te}(w) - L(w) \geq \epsilon + L^*\} \leq \frac{\beta^2}{m'\epsilon^2} \mathbb{E}_D[\mathbb{W}^2(P_{X|D=d}, P_X)],$$

*where $L^* = \min_{w^* \in \mathcal{W}}(L_{te}(w^*) + L(w^*))$. Furthermore, if the metric $d$ is discrete, then*

$$\Pr\{L_{te}(w) - L(w) \geq \epsilon + L^*\} \leq \frac{\beta^2}{2m'\epsilon^2} I(X; D).$$

*Proof.* Following the proof sketch of Theorem 5, by setting $P = P_X$, $Q = P_{X|D=d}$ and $f(x) = \ell(f_w(x), f_{w'}(x))$ in Lemma 5, we have

$$(L_d(w, w') - L(w, w'))^2 \leq \beta^2 \mathbb{W}^2(P_{X|D=d}, P_X),$$

for any $d \in \mathcal{D}$ and $w, w' \in \mathcal{W}$. Similarly, by applying the independence of $\{D_k\}_{k=1}^{m'}$ and Chebyshev's inequality, we have

$$\Pr\{|L_{te}(w, w') - L(w, w')| \geq \epsilon\} \leq \frac{\beta^2}{m'\epsilon^2} \mathbb{E}_D[\mathbb{W}^2(P_{X|D=d}, P_X)].$$

Through a similar procedure of proving Theorem 5 and Theorem 11, we can get

$$\Pr\left\{L_{te}(w) - L(w) \geq \epsilon + \min_{w^* \in \mathcal{W}}(L_{te}(w^*) + L(w^*))\right\} \leq \frac{\beta^2}{m'\epsilon^2} \mathbb{E}_D[\mathbb{W}^2(P_{X|D=d}, P_X)]$$
$$\leq \frac{\beta^2}{2m'\epsilon^2} I(X; D),$$

which finishes the proof. $\qquad\qquad\qquad\qquad\qquad\qquad\qquad\qquad\qquad\qquad\qquad\qquad\qquad\square$

The expected Wasserstein distances can be regarded as analogs to the mutual information terms $I(Z; D)$ and $I(X; D)$ respectively, through a similar reduction procedure as depicted in (2). This also provides alternative perspectives to the distribution shift and the covariate shift in (1).

Moreover, these bounds can be further tightened by considering the risk space distribution shift. Given a hypothesis $w \in \mathcal{W}$ and domain $d \in \mathcal{D}$, let $V = \ell(f_w(X), Y)$ be the risk of predicting some random sample $Z \sim P_{Z|D=d}$. Then by applying Lemma 4 with $P = P_V$, $Q = P_{V|D=d}$, $f(x) = x$ and Assumption 2, we have

$$\mathbb{E}_D[(L_D(w) - L(w))^2] = \mathbb{E}_D[(\mathbb{E}_{V|D=d}[V] - \mathbb{E}_V[V])^2]$$
$$\leq \mathbb{E}_D\left[\left(\sqrt{\frac{M^2}{2}\mathrm{KL}(P_{V|D=d} \| P_V)}\right)^2\right]$$
$$= \frac{M^2}{2} I(V; D).$$

Through the similar sketch of proving Theorem 4, we can prove that

$$\Pr\{|L_{te}(w) - L(w)| \geq \epsilon\} \leq \frac{M^2}{2m'\epsilon^2} I(V; D).$$

By the Markov chain relationship $D \to (X, Y) \to (f_\phi(X), Y) \to (f_w(X), Y) \to V$, this upper bound is strictly tighter than Theorem 4 which uses sample space $I(Z; D)$ or representation space $I(R, Y; D)$ distribution shifts. Also, notice that the mutual information $I(V; D)$ could be rewritten as $\mathrm{KL}(P_{V|D} \| P_V)$, this suggests that matching the inter-domain distributions of the risks helps to generalize on test domains. Considering that $V$ is a scalar while $R$ is a vector, aligning the distributions of the risks avoids high-dimensional distribution matching, and thus enables effective implementation than aligning the representations. We will leave this method for future research.

### D.4 Generalization Bounds for Source-Domain Empirical Risk

The information-theoretic technique adopted in this paper to derive generalization bounds is closely related to the recent advancements of information-theoretic generalization analysis. Specifically, Theorem 2 can be viewed as a multi-domain version of the standard generalization error bound in supervised learning Xu & Raginsky (2017); Bu et al. (2020). In this section, we further consider the effect of finite training samples, which further raises a gap between domain-level population and empirical risks. In addition to the generalization bounds for population risks established in Section 3, upper bounds for the empirical risk can also be derived by incorporating Assumption 2:

**Theorem 13.** *Let $W$ be the output of learning algorithm $\mathcal{A}$ with input $S_{tr}$. If Assumption 2 holds, then*

$$|\mathbb{E}_{W, D_{tr}, S_{tr}}[L'(W)] - \mathbb{E}_W[L(W)]| \leq \frac{1}{m}\sum_{i=1}^{m}\sqrt{\frac{M^2}{2}I(W; D_i)}$$
$$+ \frac{1}{mn}\sum_{i=1}^{m}\sum_{j=1}^{n}\sqrt{\frac{M^2}{2}I(W; Z_j^i|D_i)},$$

$$\Pr\{|\mathbb{E}_{W, D_{tr}, S_{tr}}[L'(W)] - \mathbb{E}_W[L(W)]| \geq \epsilon\} \leq \frac{M^2}{mn\epsilon^2}(I(W; S_{tr}) + \log 3).$$

*Proof.* Recall that any random variables bounded by $[0, M]$ are $\frac{M}{2}$-subgaussian. From assumption 2, we know that $\ell(f_W(X), Y)$ is $\frac{M}{2}$-subgaussian w.r.t $P_W \circ P_Z$. Then by applying Lemma 4, we have

$$|\mathbb{E}_{W,D_{tr},S_{tr}}[L'(W)] - \mathbb{E}_W[L(W)]|$$

$$= \left| \frac{1}{m} \sum_{i=1}^m \frac{1}{n} \sum_{j=1}^n \mathbb{E}_{W,D_i,Z_j^i}[\ell(f_W(X_j^i), Y_j^i)] - \mathbb{E}_{W,D,Z}[\ell(f_W(X), Y)] \right|$$

$$\le \frac{1}{mn} \sum_{i=1}^m \sum_{j=1}^n \left| \mathbb{E}_{W,D_i,Z_j^i}[\ell(f_W(X_j^i), Y_j^i)] - \mathbb{E}_{W,D,Z}[\ell(f_W(X), Y)] \right|$$

$$\le \frac{1}{mn} \sum_{i=1}^m \sum_{j=1}^n \sqrt{\frac{M^2}{2} \mathrm{KL}\left(P_{W,D_i,Z_j^i} \,\big\|\, P_W P_{D_i,Z_j^i}\right)}.$$

Notice that for any $D \in D_{tr}$ and $Z \in S_D$,

$$\mathrm{KL}(P_{W,D,Z} \,\|\, P_W P_{D,Z}) = \mathbb{E}_{W,D,Z}\left[ \log \frac{P_{W,D,Z}}{P_W P_{D,Z}} \right]$$

$$= I(W; D, Z)$$

$$= I(W; D) + I(W; Z|D).$$

Combining our results above, we then get

$$|\mathbb{E}_{W,D_{tr},S_{tr}}[L'(W)] - \mathbb{E}_W[L(W)]| \le \frac{1}{mn} \sum_{i=1}^m \sum_{j=1}^n \sqrt{\frac{M^2}{2}(I(W; D_i) + I(W; Z_j^i|D_i))}$$

$$\le \frac{1}{mn} \sum_{i=1}^m \sum_{j=1}^n \left( \sqrt{\frac{M^2}{2} I(W; D_i)} + \sqrt{\frac{M^2}{2} I(W; Z_j^i|D_i)} \right)$$

$$= \frac{1}{m} \sum_{i=1}^m \sqrt{\frac{M^2}{2} I(W; D_i)} + \frac{1}{mn} \sum_{i=1}^m \sum_{j=1}^n \sqrt{\frac{M^2}{2} I(W; Z_j^i|D_i)}.$$

Similarly, we have

$$\mathbb{E}_{W,S_{tr}}\left[ (L'(W) - \mathbb{E}_W[L(W)])^2 \right]$$

$$= \mathbb{E}_{W,S_{tr}}\left[ \left( \frac{1}{m} \sum_{i=1}^m \frac{1}{n} \sum_{j=1}^n \ell(f_W(X_j^i), Y_j^i) - \mathbb{E}_W[L(W)] \right)^2 \right]$$

$$= \frac{M^2}{mn}(I(W; S_{tr}) + \log 3).$$

This further implies by Lemma 1 that

$$\Pr\{|\mathbb{E}_{W,D_{tr},S_{tr}}[L'(W)] - \mathbb{E}_W[L(W)]| \ge \epsilon\} \le \frac{M^2}{mn\epsilon^2}(I(W; S_{tr}) + \log 3),$$

which completes the proof. $\qquad\square$

**Theorem 14.** *Let $W$ be the output of learning algorithm $\mathcal{A}$ under training domains $D_{tr}$. If $\ell(f_w(X), Y)$ is $\beta'$-Lipschitz w.r.t $w$, then*

$$|\mathbb{E}_{W,D_{tr},S_{tr}}[L'(W)] - \mathbb{E}_W[L(W)]| \le \frac{\beta'}{m} \sum_{i=1}^m \mathbb{E}_{D_i}[\mathbb{W}(P_{W|D_i}, P_W)]$$

$$+ \frac{\beta'}{mn} \sum_{i=1}^m \sum_{j=1}^n \mathbb{E}_{D_i,Z_j^i}[\mathbb{W}(P_{W|D_i,Z_j^i}, P_{W|D_i})].$$

*Proof.* Recall the proof of Theorem 13, we have

$$
\left| \mathbb{E}_{W,D_{tr},S_{tr}}[L'(W)] - \mathbb{E}_W[L(W)] \right|
$$

$$
\leq \frac{1}{mn} \sum_{i=1}^{m} \sum_{j=1}^{n} \left| \mathbb{E}_{W,D_i,Z_j^i}[\ell(f_W(X_j^i), Y_j^i)] - \mathbb{E}_{W,D,Z}[\ell(f_W(X), Y)] \right|
$$

$$
\leq \frac{1}{mn} \sum_{i=1}^{m} \sum_{j=1}^{n} \left| \mathbb{E}_{W,D_i,Z_j^i}[\ell(f_W(X_j^i), Y_j^i)] - \mathbb{E}_{W,D_i,Z}[\ell(f_W(X), Y)] \right|
$$

$$
+ \frac{1}{mn} \sum_{i=1}^{m} \sum_{j=1}^{n} \left| \mathbb{E}_{W,D_i,Z}[\ell(f_W(X), Y)] - \mathbb{E}_{W,D,Z}[\ell(f_W(X), Y)] \right|
$$

$$
\leq \frac{1}{mn} \sum_{i=1}^{m} \sum_{j=1}^{n} \mathbb{E}_{D_i,Z_j^i} \left| \mathbb{E}_{W|D_i,Z_j^i}[\ell(f_W(X_j^i), Y_j^i)] - \mathbb{E}_{W|D_i}[\ell(f_W(X_j^i), Y_j^i)] \right|
$$

$$
+ \frac{1}{mn} \sum_{i=1}^{m} \sum_{j=1}^{n} \mathbb{E}_{D_i} \left| \mathbb{E}_{W|D_i}[L_{D_i}(W)] - \mathbb{E}_W[L_{D_i}(W)] \right|
$$

$$
\leq \frac{\beta'}{mn} \sum_{i=1}^{m} \sum_{j=1}^{n} \left( \mathbb{E}_{D_i,Z_j^i}[\mathbb{W}(P_{W|D_i,Z_j^i}, P_{W|D_i})] + \mathbb{E}_{D_i}[\mathbb{W}(P_{W|D_i}, P_W)] \right)
$$

$$
= \frac{\beta'}{m} \sum_{i=1}^{m} \mathbb{E}_{D_i}[\mathbb{W}(P_{W|D_i}, P_W)] + \frac{\beta'}{mn} \sum_{i=1}^{m} \sum_{j=1}^{n} \mathbb{E}_{D_i,Z_j^i}[\mathbb{W}(P_{W|D_i,Z_j^i}, P_{W|D_i})],
$$

where the last inequality is by applying Lemma 5. The proof is complete. $\qquad\square$

The theorems above provide upper bounds for the empirical generalization risk by exploiting the mutual information between the hypothesis and the samples (or the Wasserstein distance counterparts). Compared to Theorems 2 and 3, these results additionally consider the randomness of the sampled data points $Z_j^i$, and indicate that traditional techniques that improve the generalization of deep learning algorithms by minimizing $I(W; S_{tr})$, such as gradient clipping (Wang & Mao, 2021; 2022) and stochastic gradient perturbation (Pensia et al., 2018; Wang et al., 2021) methods, also enhance the capability of learning algorithm $\mathcal{A}$ to generalize on target domains under our high-probability problem setting by preventing overfitting to training samples. This observation is also verified in (Wang & Mao, 2022). Relevant analysis may also motivate information-theoretic generalization analysis for meta-learning tasks (Chen et al., 2021; Jose & Simeone, 2021; Hellström & Durisi, 2022; Bu et al., 2023). We do not consider these approaches in this paper, as they are beyond the scope of solving Problem 1.

# E    EXPERIMENT DETAILS

In this paper, deep learning models are trained with an Intel Xeon CPU (2.10GHz, 48 cores), 256GB memory, and 4 Nvidia Tesla V100 GPUs (32GB).

## E.1    IMPLEMENTATION OF IDM

We provide the pseudo-code for PDM in Algorithm 1, where the moving averages $X_{ma}^i$ are initialized with 0. The input data points for distribution alignment are represented as matrices $X^i \in \mathbb{R}^{b \times d}$, where $b$ denotes the batch size and $d$ represents the dimensionality. Each row of $X$ then corresponds to an individual data point. We also present the pseudo-code for IDM in Algorithm 2 for completeness.

We follow the experiment settings of (Rame et al., 2022) and utilize a moving average to increase the equivalent number of data points for more accurate probability density estimation in distribution alignments. This does not invalidate our analysis in Theorem 8, as the maximum equivalent batch size ($b/(1 - \gamma) \approx 32/(1 - 0.95) = 640$) remains significantly smaller than the dimensionality of the representation (2048 for ResNet-50 in DomainBed) or the gradient ($2048 \times c$, the number of

---

**Algorithm 1** PDM for distribution matching.

---

1: **Input:** Data matrices $\{X^i\}_{i=1}^m$, moving average $\gamma$.
2: **Output:** Penalty of distribution matching.
3: **for** $i$ from 1 **to** $m$ **do**
4:     Sort the elements of $X^i$ in each column in ascending order.
5:     Calculate moving average $X_{ma}^i = \gamma X_{ma}^i + (1-\gamma)X^i$.
6: **end for**
7: Calculate the mean of data points across domains: $X_{ma} = \frac{1}{m}\sum_{i=1}^m X_{ma}^i$.
8: **Output:** $\mathcal{L}_{\text{PDM}} = \frac{1}{mdb}\sum_{i=1}^m \|X_{ma} - X_{ma}^i\|_F^2$.

---

**Algorithm 2** IDM for high-probability DG.

---

1: **Input:** Model $W$, training dataset $S_{tr}$, hyper-parameters $\lambda_1, \lambda_2, t_1, t_2, \gamma_1, \gamma_2$.
2: **for** $t$ from 1 **to** #steps **do**
3:     **for** $i$ from 1 **to** $m$ **do**
4:         Randomly sample a batch $B_t^i = (X_t^i, Y_t^i)$ from $S_{D_i}$ of size $b$.
5:         Compute individual representations: $(R_t^i)_j = f_\Phi\big((X_t^i)_j\big)$, for $j \in [1, b]$.
6:         Compute individual risks: $(L_t^i)_j = \ell\big(f_\Psi\big((R_t^i)_j\big), (Y_t^i)_j\big)$, for $j \in [1, b]$.
7:         Compute individual gradients: $(G_t^i)_j = \nabla_\Psi(L_t^i)_j$, for $j \in [1, b]$.
8:     **end for**
9:     Compute total empirical risk: $\mathcal{L}_{\text{IDM}} = \frac{1}{mn}\sum_{i=1}^m\sum_{j=1}^n(L_t^i)_j$.
10:     **if** $t \geq t_1$ **then**
11:         Compute gradient alignment risk: $\mathcal{L}_{\text{G}} = \text{PDM}(\{G_t^i\}_{i=1}^m, \gamma_1)$.
12:         $\mathcal{L}_{\text{IDM}} = \mathcal{L}_{\text{IDM}} + \lambda_1\mathcal{L}_{\text{G}}$.
13:     **end if**
14:     **if** $t \geq t_2$ **then**
15:         Compute representation alignment risk: $\mathcal{L}_{\text{R}} = \text{PDM}(\{R_t^i\}_{i=1}^m, \gamma_2)$.
16:         $\mathcal{L}_{\text{IDM}} = \mathcal{L}_{\text{IDM}} + \lambda_2\mathcal{L}_{\text{R}}$.
17:     **end if**
18:     Back-propagate gradients $\nabla_W\mathcal{L}_{\text{IDM}}$ and update the model $W$.
19: **end for**

---

classes) and satisfies $d > 2b+1$. Therefore, it is still impossible to distinguish different inter-domain distributions as indicated by Theorem 8. However, this moving average technique indeed helps to improve the empirical performance, as shown by our ablation studies.

## E.2 COLORED MNIST

The Colored MNIST dataset is a binary classification task introduced by IRM (Arjovsky et al., 2019). The main difference between Colored MNIST and the original MNIST dataset is the manually introduced strong correlation between the label and image colors. Colored MNIST is generated according to the following procedure:

- Give each sample an initial label by whether the digit is greater than 4 (i.e. label 0 for 0-4 digits and label 1 for 5-9 digits.

- Randomly flip the label with probability 0.25, so an oracle predictor that fully relies on the shape of the digits would achieve a 75% accuracy.

- Each environment is assigned a probability $P_e$, which characterizes the correlation between the label and the color: samples with label 0 have $P_e$ chance to be red, and $1 - P_e$ chance to be green, while samples with label 1 have $P_e$ chance to be green, and $1 - P_e$ chance to be red.

The original environment setting of (Arjovsky et al., 2019) includes two training domains $D_{tr} = \{P_1 = 90\%, P_2 = 80\%\}$, such that the predictive power of the color superiors that of the actual digits. This correlation is reversed in the test domain $D_{te} = \{P_3 = 10\%\}$, thus fooling algorithms without causality inference abilities to overfit the color features and generalize poorly on test environments.

The original implementation[2] uses a 3-layer MLP network with ReLU activation. The model is trained for 501 epochs in a full gradient descent scheme, such that the batch size equals the number of training samples $25,000$. We follow the hyper-parameter selection strategy of (Arjovsky et al., 2019) through a random search over 50 independent trials, as reported in Table 3 along with the parameters selected for IDM. Considering that the covariate shift is not prominent according to the dataset construction procedure, we only apply gradient alignment without feature alignment in this experiment.

Table 3: The hyper-parameters of Colored MNIST.

| Parameter | Random Distribution | Selected Value |
|---|---|---|
| dimension of hidden layer | $2^{\text{Uniform}(6,9)}$ | 433 |
| weight decay | $10^{\text{Uniform}(-2,-5)}$ | 0.00034 |
| learning rate | $10^{\text{Uniform}(-2.5,-3.5)}$ | 0.000449 |
| warmup iterations | $\text{Uniform}(50, 250)$ | 154 |
| regularization strength | $10^{\text{Uniform}(4,8)}$ | 2888595.180638 |

### E.3 DomainBed Benchmark

DomainBed (Gulrajani & Lopez-Paz, 2020) is an extensive benchmark for both DA and DG algorithms, which involves various synthetic and real-world datasets mainly focusing on image classification:

- Colored MNIST (Arjovsky et al., 2019) is a variant of the MNIST dataset. As discussed previously, Colored MNIST includes 3 domains $\{90\%, 80\%, 10\%\}$, $70,000$ samples of dimension $(2, 28, 28)$ and 2 classes.

- Rotated MNIST (Ghifary et al., 2015) is a variant of the MNIST dataset with 7 domains $\{0, 15, 30, 45, 60, 75\}$ representing the rotation degrees, $70,000$ samples of dimension $(28, 28)$ and 10 classes.

- VLCS (Fang et al., 2013) includes 4 domains $\{\text{Caltech101}, \text{LabelMe}, \text{SUN09}, \text{VOC2007}\}$, $10,729$ samples of dimension $(3, 224, 224)$ and 5 classes.

- PACS (Li et al., 2017) includes 4 domains $\{\text{art}, \text{cartoons}, \text{photos}, \text{sketches}\}$, $9,991$ samples of dimension $(3, 224, 224)$ and 7 classes.

- OfficeHome (Venkateswara et al., 2017) includes 4 domains $\{\text{art}, \text{clipart}, \text{product}, \text{real}\}$, $15,588$ samples of dimension $(3, 224, 224)$ and 65 classes.

- TerraIncognita (Beery et al., 2018) includes 4 domains $\{\text{L100}, \text{L38}, \text{L43}, 46\}$ representing locations of photographs, $24,788$ samples of dimension $(3, 224, 224)$ and 10 classes.

- DomainNet (Peng et al., 2019) includes 6 domains $\{\text{clipart}, \text{infograph}, \text{painting}, \text{quickdraw}, \text{real}, \text{sketch}\}$, $586,575$ samples of dimension $(3, 224, 224)$ and 345 classes.

We list all competitive DG approaches below. Note that some recent progress is omitted (Cha et al., 2021; Eastwood et al., 2022; Wang et al., 2022; 2023; Setlur et al., 2023; Chen et al., 2023), which either contributes complementary approaches, does not report full DomainBed results, or does not report the test-domain validation scores. Due to the limitation of computational resources, we are not able to reproduce the full results of these works on DomainBed.

- ERM: Empirical Risk Minimization.
- IRM: Invariant Risk Minimization (Arjovsky et al., 2019).
- GroupDRO: Group Distributionally Robust Optimization (Sagawa et al., 2019).
- Mixup: Interdomain Mixup (Yan et al., 2020).
- MLDG: Meta Learning Domain Generalization (Li et al., 2018a).
- CORAL: Deep CORAL (Sun & Saenko, 2016).

---

[2]`https://github.com/facebookresearch/InvariantRiskMinimization`

- MMD: Maximum Mean Discrepancy (Li et al., 2018b).
- DANN: Domain Adversarial Neural Network (Ganin et al., 2016).
- CDANN: Conditional Domain Adversarial Neural Network (Li et al., 2018c).
- MTL: Marginal Transfer Learning (Blanchard et al., 2021).
- SagNet: Style Agnostic Networks (Nam et al., 2021).
- ARM: Adaptive Risk Minimization (Zhang et al., 2021).
- V-REx: Variance Risk Extrapolation (Krueger et al., 2021).
- RSC: Representation Self-Challenging (Huang et al., 2020).
- AND-mask: Learning Explanations that are Hard to Vary (Parascandolo et al., 2020).
- SAND-mask: Smoothed-AND mask (Shahtalebi et al., 2021).
- Fish: Gradient Matching for Domain Generalization (Shi et al., 2021).
- Fishr: Invariant Gradient Variances for Out-of-distribution Generalization (Rame et al., 2022).
- SelfReg: Self-supervised Contrastive Regularization (Kim et al., 2021).
- CausIRL: Invariant Causal Mechanisms through Distribution Matching (Chevalley et al., 2022).

The same fine-tuning procedure is applied to all approaches: The network is a multi-layer CNN for synthetic MNIST datasets and is a pre-trained ResNet-50 for other real-world datasets. The hyper-parameters are selected by a random search over 20 independent trials for each target domain, and each evaluation score is reported after 3 runs with different initialization seeds[3]. The hyper-parameter selection criteria are shown in Table 4. Note that warmup iterations and moving average techniques are not adopted for representation alignment.

Table 4: The hyper-parameters of DomainBed.

| Condition | Parameter | Default Value | Random Distribution |
|---|---|---|---|
| MNIST Datasets | learning rate | 0.001 | $10^{\text{Uniform}(-4.5,-3.5)}$ |
| | batch size | 64 | $2^{\text{Uniform}(3,9)}$ |
| Real-world Datasets | learning rate | 0.00005 | $10^{\text{Uniform}(-5,-3.5)}$ |
| | batch size | 32 | $2^{\text{Uniform}(3,5)}$ (DomainNet) / $2^{\text{Uniform}(3,5.5)}$ (others) |
| | weight decay | 0 | $10^{\text{Uniform}(-6,-2)}$ |
| | dropout | 0 | $\text{Uniform}(\{0, 0.1, 0.5\})$ |
| - | steps | 5000 | 5000 |
| IDM | gradient penalty | 1000 | $10^{\text{Uniform}(1,5)}$ |
| | gradient warmup | 1500 | $\text{Uniform}(0, 5000)$ |
| | representation penalty | 1 | $10^{\text{Uniform}(-1,1)}$ |
| | moving average | 0.95 | $\text{Uniform}(0.9, 0.99)$ |

Note that although the same Colored MNIST dataset is adopted by DomainBed, the experimental settings are completely different from the previous one (Arjovsky et al., 2019). The main difference is the batch size (25000 for IRM, less than 512 for DomainBed), making it much harder to learn invariance for causality inference and distribution matching methods since fewer samples are available for probability density estimation. This explains the huge performance drop between these two experiments using the same DG algorithms.

## F  ADDITIONAL EXPERIMENTAL RESULTS

### F.1  COMPONENT ANALYSIS

In this section, we conduct ablation studies to demonstrate the effect of each component of the proposed IDM algorithm. Specifically, we analyze the effect of gradient alignment (GA), representation alignment (RA), warmup iterations (WU), moving average (MA), and the proposed PDM method for distribution matching.

---

[3]https://github.com/facebookresearch/DomainBed

Table 5: Component Analysis on ColoredMNIST of DomainBed.

| Algorithm | GA | RA | WU | MA | 90% | 80% | 10% | Average |
|---|---|---|---|---|---|---|---|---|
| ERM | | | - | | $71.8 \pm 0.4$ | $72.9 \pm 0.1$ | $28.7 \pm 0.5$ | 57.8 |
| IDM | ✗ | ✓ | ✗ | ✗ | $71.9 \pm 0.4$ | $72.5 \pm 0.0$ | $28.8 \pm 0.7$ | 57.7 |
| | ✓ | ✗ | ✓ | ✓ | $73.1 \pm 0.2$ | $72.7 \pm 0.3$ | $67.4 \pm 1.6$ | 71.1 |
| | ✓ | ✓ | ✗ | ✓ | $72.9 \pm 0.2$ | $72.7 \pm 0.1$ | $60.8 \pm 2.1$ | 68.8 |
| | ✓ | ✓ | ✓ | ✗ | $72.0 \pm 0.1$ | $71.5 \pm 0.3$ | $48.7 \pm 7.1$ | 64.0 |
| | ✓ | ✓ | ✓ | ✓ | $\mathbf{74.2} \pm 0.6$ | $\mathbf{73.5} \pm 0.2$ | $\mathbf{68.3} \pm 2.5$ | **72.0** |

Table 6: Component Analysis on OfficeHome of DomainBed.

| Algorithm | GA | RA | WU | MA | A | C | P | R | Average |
|---|---|---|---|---|---|---|---|---|---|
| ERM | | | - | | $61.7 \pm 0.7$ | $53.4 \pm 0.3$ | $74.1 \pm 0.4$ | $76.2 \pm 0.6$ | 66.4 |
| IDM | ✗ | ✓ | ✗ | ✗ | $\mathbf{64.7} \pm 0.5$ | $\mathbf{54.6} \pm 0.3$ | $76.2 \pm 0.4$ | $\mathbf{78.1} \pm 0.5$ | **68.4** |
| | ✓ | ✗ | ✓ | ✓ | $61.9 \pm 0.4$ | $53.0 \pm 0.3$ | $75.5 \pm 0.2$ | $77.9 \pm 0.2$ | 67.1 |
| | ✓ | ✓ | ✗ | ✓ | $62.5 \pm 0.1$ | $53.0 \pm 0.7$ | $75.0 \pm 0.4$ | $77.2 \pm 0.7$ | 66.9 |
| | ✓ | ✓ | ✓ | ✗ | $64.2 \pm 0.3$ | $53.5 \pm 0.6$ | $76.1 \pm 0.4$ | $\mathbf{78.1} \pm 0.4$ | 68.0 |
| | ✓ | ✓ | ✓ | ✓ | $64.4 \pm 0.3$ | $54.4 \pm 0.6$ | $\mathbf{76.5} \pm 0.3$ | $78.0 \pm 0.4$ | 68.3 |

### F.1.1 GRADIENT ALIGNMENT

According to our theoretical analysis, gradient alignment promotes training-domain generalization, especially when concept shift is prominent. As can be seen in Table 5, IDM without gradient alignment (57.7%) performs similarly to ERM (57.8%), which is unable to learn invariance across training domains. Gradient alignment also significantly boosts the performance on VLCS (77.4% to 78.1%) and PACS (86.8% to 87.6%), as seen in Table 7 and 8. However, for datasets where concept shift is not prominent e.g. OfficeHome, gradient alignment cannot help to improve performance as shown in Table 6. It is worth noting that gradient alignment also penalizes a lower bound for the representation space distribution shift: In the $t$-th step of gradient descent, the Markov chain relationship $D_i \to B_t^i \to (R_t^i, Y_t^i) \to G_t^i$ holds conditioned on the current predictor $W_{t-1}$, which implies the lower bound $I(G_t^i; D_i|W_{t-1}) \leq I(R_t^i, Y_t^i; D_i|W_{t-1})$ by the data processing inequality. This indicates that gradient alignment also helps to address the covariate shift, which explains the promising performance of gradient-based DG algorithms e.g. Fish and Fishr. However, since this is a lower bound rather than an upper bound, gradient manipulation is insufficient to fully address representation space covariate shifts, as seen in the following analysis for representation alignment.

Table 7: Effect of gradient alignment (GA) on VLCS of DomainBed.

| Algorithm | GA | A | C | P | S | Average |
|---|---|---|---|---|---|---|
| ERM | - | $\mathbf{97.6} \pm 0.3$ | $\mathbf{67.9} \pm 0.7$ | $70.9 \pm 0.2$ | $74.0 \pm 0.6$ | 77.6 |
| IDM | ✗ | $97.1 \pm 0.7$ | $67.2 \pm 0.4$ | $69.9 \pm 0.4$ | $75.6 \pm 0.8$ | 77.4 |
| IDM | ✓ | $\mathbf{97.6} \pm 0.3$ | $66.9 \pm 0.3$ | $\mathbf{71.8} \pm 0.5$ | $\mathbf{76.0} \pm 1.3$ | **78.1** |

### F.1.2 REPRESENTATION ALIGNMENT

Representation alignment promotes test-domain generalization by minimizing the representation level covariate shift. As shown in Table 5 - 9, representation alignment is effective in OfficeHome (67.1% to 68.3%) and RotatedMNIST (97.8% to 98.0%), and still enhances the performance even though covariate shift is not prominent in ColoredMNIST (71.1% to 72.0%). This verifies our claim that representation alignment complements gradient alignment in solving Problem 1, and is necessary for achieving high-probability DG.

Table 8: Effect of gradient alignment (GA) on PACS of DomainBed.

| Algorithm | GA | A | C | P | S | Average |
|---|---|---|---|---|---|---|
| ERM | - | $86.5 \pm 1.0$ | $81.3 \pm 0.6$ | $96.2 \pm 0.3$ | $\mathbf{82.7} \pm 1.1$ | 86.7 |
| IDM | ✗ | $87.8 \pm 0.6$ | $81.6 \pm 0.3$ | $97.4 \pm 0.2$ | $80.6 \pm 1.3$ | 86.8 |
| IDM | ✓ | $\mathbf{88.0} \pm 0.3$ | $\mathbf{82.6} \pm 0.6$ | $\mathbf{97.6} \pm 0.4$ | $82.3 \pm 0.6$ | $\mathbf{87.6}$ |

Table 9: Effect of representation alignment (RA) on RotatedMNIST of DomainBed.

| Algorithm | RA | 0 | 15 | 30 | 45 | 60 | 75 | Average |
|---|---|---|---|---|---|---|---|---|
| ERM | - | $95.3 \pm 0.2$ | $\mathbf{98.7} \pm 0.1$ | $98.9 \pm 0.1$ | $98.7 \pm 0.2$ | $\mathbf{98.9} \pm 0.0$ | $96.2 \pm 0.2$ | 97.8 |
| IDM | ✗ | $95.6 \pm 0.1$ | $98.4 \pm 0.1$ | $98.7 \pm 0.2$ | $\mathbf{99.1} \pm 0.0$ | $98.7 \pm 0.1$ | $\mathbf{96.6} \pm 0.4$ | 97.8 |
| IDM | ✓ | $\mathbf{96.1} \pm 0.3$ | $\mathbf{98.7} \pm 0.1$ | $\mathbf{99.1} \pm 0.1$ | $98.9 \pm 0.1$ | $\mathbf{98.9} \pm 0.1$ | $\mathbf{96.6} \pm 0.1$ | $\mathbf{98.0}$ |

### F.1.3 WARMUP ITERATIONS

Following the experimental settings of (Arjovsky et al., 2019; Rame et al., 2022), we do not apply the penalties of gradient or representation alignment until the number of epochs reaches a certain value. This is inspired by the observation that forcing invariance in early steps may hinder the models from extracting useful correlations. By incorporating these warmup iterations, predictors are allowed to extract all possible correlations between the inputs and the labels at the beginning, and then discard spurious ones in later updates. As can be seen in Table 5 and 6, this strategy helps to enhance the final performances on ColoredMNIST (68.8% to 72.0%) and OfficeHome (66.9% to 68.3%).

### F.1.4 MOVING AVERAGE

Following Rame et al. (2022); Pooladzandi et al. (2022), we use an exponential moving average when computing the gradients or the representations. This strategy helps when the batch size is not sufficiently large to sketch the probability distributions. In the IRM experiment setup where the batch size is 25000, Fishr (70.2%) and IDM (70.5%) both easily achieve near-optimal accuracy compared to Oracle (71.0%). In the DomainBed setup, the batch size $2^{\text{Uniform}(3,9)}$ is significantly diminished, resulting in worse test-domain accuracy of Fishr (68.8%). As shown in Table 5 and 6, this moving average strategy greatly enhances the performance of IDM on ColoredMNIST (64.0% to 72.0%) and OfficeHome (68.0% to 68.3%).

### F.1.5 PDM FOR DISTRIBUTION MATCHING

We then demonstrate the superiority of our PDM method over moment-based distribution alignment techniques. Specifically, we compare IGA (Koyama & Yamaguchi, 2020) which matches the empirical expectation of the gradients, Fishr (Rame et al., 2022) which proposes to align the gradient variance, the combination of IGA + Fishr (i.e. aligning the expectation and variance simultaneously), and our approach IDM (without representation space alignment). The performance gain of IDM on the Colored MNIST task in (Arjovsky et al., 2019) is not significant, since it is relatively easier to learn invariance with a large batch size (25000). In the DomainBed setting, the batch size is significantly reduced (8-512), making this learning task much harder. The results are reported in Table 10.

Table 10: Superiority of PDM on Colored MNIST of DomainBed.

| Algorithm | 90% | 80% | 10% | Average |
|---|---|---|---|---|
| ERM | $71.8 \pm 0.4$ | $72.9 \pm 0.1$ | $28.7 \pm 0.5$ | 57.8 |
| IGA | $72.6 \pm 0.3$ | $72.9 \pm 0.2$ | $50.0 \pm 1.2$ | 65.2 |
| Fishr | $74.1 \pm 0.6$ | $73.3 \pm 0.1$ | $58.9 \pm 3.7$ | 68.8 |
| IGA + Fishr | $73.3 \pm 0.0$ | $72.6 \pm 0.5$ | $66.3 \pm 2.9$ | 70.7 |
| IDM | $\mathbf{74.2} \pm 0.6$ | $\mathbf{73.5} \pm 0.2$ | $\mathbf{68.3} \pm 2.5$ | $\mathbf{72.0}$ |

Table 11: Computational overhead of IDM using default batch size.

| Dataset | Training Time (h) | | | Memory Requirement (GB) | | |
|---|---|---|---|---|---|---|
| | ERM | IDM | Overhead | ERM | IDM | Overhead |
| ColoredMNIST | 0.076 | 0.088 | 14.6% | 0.138 | 0.139 | 0.2% |
| RotatedMNIST | 0.101 | 0.110 | 9.3% | 0.338 | 0.342 | 1.0% |
| VLCS | 0.730 | 0.744 | 2.0% | 8.189 | 8.199 | 0.1% |
| PACS | 0.584 | 0.593 | 1.5% | 8.189 | 8.201 | 0.1% |
| OfficeHome | 0.690 | 0.710 | 2.9% | 8.191 | 8.506 | 3.8% |
| TerraIncognita | 0.829 | 0.840 | 1.3% | 8.189 | 8.208 | 0.2% |
| DomainNet | 2.805 | 2.947 | 5.0% | 13.406 | 16.497 | 23.1% |

As can be seen, IDM achieves significantly higher performance on Colored MNIST (72.0%) even compared to the combination of IGA + Fishr (70.7%). This verifies our conclusion that matching the expectation and the variance is not sufficient for complex probability distributions, and demonstrates the superiority of the proposed PDM method for distribution alignment.

## F.2 RUNNING TIME COMPARISON

Since IDM only stores historical gradients and representations for a single batch from each training domain, the storage and computation overhead is marginal compared to training the entire network. As shown in Table 11, the training time is only 5% longer compared to ERM on the largest Domain-Net dataset.

## F.3 TRAINING-DOMAIN MODEL SELECTION

Table 12: DomainBed using training-domain validation. We format **best**, second best and worse than ERM results.

| Algorithm | Accuracy (↑) | | | | | | | | Ranking (↓) | | |
|---|---|---|---|---|---|---|---|---|---|---|---|
| | CMNIST | RMNIST | VLCS | PACS | OffHome | TerraInc | DomNet | Avg | Mean | Median | Worst |
| ERM | 51.5 ± 0.1 | 98.0 ± 0.0 | 77.5 ± 0.4 | 85.5 ± 0.2 | 66.5 ± 0.3 | 46.1 ± 1.8 | 40.9 ± 0.1 | 66.6 | 9.6 | 10 | 15 |
| IRM | 52.0 ± 0.1 | 97.7 ± 0.1 | 78.5 ± 0.5 | 83.5 ± 0.8 | 64.3 ± 2.2 | 47.6 ± 0.8 | 33.9 ± 2.8 | 65.4 | 13.1 | 18 | 22 |
| GroupDRO | 52.1 ± 0.0 | 98.0 ± 0.0 | 76.7 ± 0.6 | 84.4 ± 0.8 | 66.0 ± 0.7 | 43.2 ± 1.1 | 33.3 ± 0.2 | 64.8 | 13.9 | 17 | 22 |
| Mixup | 52.1 ± 0.2 | 98.0 ± 0.1 | 77.4 ± 0.6 | 84.6 ± 0.6 | 68.1 ± 0.3 | 47.9 ± 0.8 | 39.2 ± 0.1 | 66.7 | 7.4 | 4 | 17 |
| MLDG | 51.5 ± 0.1 | 97.9 ± 0.0 | 77.2 ± 0.4 | 84.9 ± 1.0 | 66.8 ± 0.6 | 47.7 ± 0.9 | 41.2 ± 0.1 | 66.7 | 10.6 | 10 | 19 |
| CORAL | 51.5 ± 0.1 | 98.0 ± 0.1 | **78.8** ± 0.6 | 86.2 ± 0.3 | **68.7** ± 0.3 | 47.6 ± 1.0 | 41.5 ± 0.1 | **67.5** | 4.6 | 3 | 15 |
| MMD | 51.5 ± 0.2 | 97.9 ± 0.0 | 77.5 ± 0.9 | 84.6 ± 0.5 | 66.3 ± 0.1 | 42.2 ± 1.6 | 23.4 ± 9.5 | 63.3 | 15.3 | 13 | 22 |
| DANN | 51.5 ± 0.3 | 97.8 ± 0.1 | 78.6 ± 0.4 | 83.6 ± 0.4 | 65.9 ± 0.6 | 46.7 ± 0.5 | 38.3 ± 0.1 | 66.1 | 13.1 | 15 | 20 |
| CDANN | 51.7 ± 0.1 | 97.9 ± 0.1 | 77.5 ± 0.1 | 82.6 ± 0.9 | 65.8 ± 1.3 | 45.8 ± 1.6 | 38.3 ± 0.3 | 65.6 | 14.0 | 14 | 22 |
| MTL | 51.4 ± 0.1 | 97.9 ± 0.0 | 77.2 ± 0.4 | 84.6 ± 0.5 | 66.4 ± 0.5 | 45.6 ± 1.2 | 40.6 ± 0.1 | 66.2 | 14.1 | 13 | 21 |
| SagNet | 51.7 ± 0.0 | 98.0 ± 0.0 | 77.8 ± 0.5 | 86.3 ± 0.2 | 68.1 ± 0.1 | **48.6** ± 1.0 | 40.3 ± 0.1 | 67.2 | 4.9 | 4 | 10 |
| ARM | **56.2** ± 0.2 | **98.2** ± 0.1 | 77.6 ± 0.3 | 85.1 ± 0.4 | 64.8 ± 0.3 | 45.5 ± 0.3 | 35.5 ± 0.2 | 66.1 | 11.0 | 10 | 21 |
| V-REx | 51.8 ± 0.1 | 97.9 ± 0.1 | 78.3 ± 0.2 | 84.9 ± 0.6 | 66.4 ± 0.6 | 46.4 ± 0.6 | 33.6 ± 2.9 | 65.6 | 10.4 | 11 | 19 |
| RSC | 51.7 ± 0.2 | 97.6 ± 0.1 | 77.1 ± 0.5 | 85.2 ± 0.9 | 65.5 ± 0.9 | 46.6 ± 1.0 | 38.9 ± 0.5 | 66.1 | 14.6 | 13 | 21 |
| AND-mask | 51.3 ± 0.2 | 97.6 ± 0.1 | 78.1 ± 0.9 | 84.4 ± 0.9 | 65.6 ± 0.4 | 44.6 ± 0.3 | 37.2 ± 0.6 | 65.5 | 16.9 | 19 | 22 |
| SAND-mask | 51.8 ± 0.2 | 97.4 ± 0.1 | 77.4 ± 0.2 | 84.6 ± 0.9 | 65.8 ± 0.4 | 42.9 ± 1.7 | 32.1 ± 0.6 | 64.6 | 16.7 | 17 | 22 |
| Fish | 51.6 ± 0.1 | 98.0 ± 0.0 | 77.8 ± 0.3 | 85.5 ± 0.3 | 68.6 ± 0.4 | 45.1 ± 1.3 | 42.7 ± 0.2 | 67.1 | 7.0 | 6 | 18 |
| Fishr | 52.0 ± 0.2 | 97.8 ± 0.0 | 77.8 ± 0.1 | 85.5 ± 0.4 | 67.8 ± 0.1 | 47.4 ± 1.6 | 41.7 ± 0.0 | 67.1 | 7.6 | 6 | 17 |
| SelfReg | 52.1 ± 0.2 | 98.0 ± 0.2 | 77.8 ± 0.9 | **86.5** ± 0.3 | 67.9 ± 0.7 | 47.0 ± 0.3 | **42.8** ± 0.0 | 67.3 | **3.7** | **2** | **8** |
| CausIRLCORAL | 51.7 ± 0.1 | 97.9 ± 0.1 | 77.5 ± 0.6 | 85.8 ± 0.1 | 68.6 ± 0.5 | 47.3 ± 0.8 | 41.9 ± 0.1 | 67.3 | 6.9 | 7 | 12 |
| CausIRLMMD | 51.6 ± 0.1 | 97.9 ± 0.0 | 77.6 ± 0.4 | 84.0 ± 0.8 | 65.7 ± 0.6 | 46.3 ± 0.9 | 40.3 ± 0.2 | 66.2 | 13.1 | 12 | 19 |
| IDM | 51.5 ± 0.1 | 98.0 ± 0.1 | 77.5 ± 0.6 | 85.9 ± 0.3 | 67.9 ± 0.1 | 46.2 ± 1.5 | 41.8 ± 0.2 | 67.0 | 8.0 | 6 | 15 |

We focus on the test-domain model selection criterion in the main text, where the validation set follows the same distribution as the test domains. Our choice is well-motivated for the following reasons:

- Test-domain validation is provided by the DomainBed benchmark as one of the default model-selection methods, and is also widely adopted in the literature in many significant works like IRM (Arjovsky et al., 2019), V-Rex (Krueger et al., 2021), and Fishr (Rame et al., 2022).

- As suggested by Theorem 1, any algorithm that fits well on training domains will suffer from strictly positive risks in test domains once concept shift is induced. Therefore, training-domain validation would result in sub-optimal selection results.

- Training-domain validation may render efforts to address concept shift useless, as spurious features are often more predictive than invariant ones. This is particularly unfair for algorithms that aim to tackle the concept shift. As shown in Table 12, no algorithm can significantly outperform ERM on Colored MNIST using training-domain validation (an exception is ARM which uses test-time adaptation, and thus cannot be directly compared), even though ERM is shown to perform much worse than random guessing (10% v.s. 50% accuracy) for the last domain (see Table 1 in (Arjovsky et al., 2019) and Appendix D.4.1 in (Rame et al., 2022)). As a result, models selected by training-domain validation may not generalize well when concept shift is substantial.

- As mentioned by D'Amour et al. (2022), training-domain validation suffers from underspecification, where predictors with equivalently strong performances in training domains may behave very differently during testing. It is also emphasized by Teney et al. (2022) that OOD performance cannot, by definition, be performed with a validation set from the same distribution as the training data. This further raises concerns about the validity of using training-domain accuracies for validation purposes.

- Moreover, test-domain validation is also applicable in practice, as it is feasible to label a few test-domain samples for validation purposes. It is also unrealistic to deploy models in target environments without any form of verification, making such efforts necessary in practice.

In Table 12, we report the results for training-domain model selection just for completeness, where the proposed IDM algorithm consistently outperforms ERM. Yet, such a training-domain selection strategy is flawed and has clear limitations, and we believe the test-domain results are sufficient to demonstrate the effectiveness of our approach in real-world learning scenarios.

## F.4 FULL DOMAINBED RESULTS

Finally, we report detailed results of IDM for each domain in each dataset of the DomainBed benchmark under test-domain model selection for a complete evaluation in Table 13 - 19. Note that detailed scores of certain algorithms (Fish, CausIRL) are not available.

Table 13: Detailed results on Colored MNIST in DomainBed.

| Algorithm | 90% | 80% | 10% | Average |
|---|---|---|---|---|
| ERM | 71.8 ± 0.4 | 72.9 ± 0.1 | 28.7 ± 0.5 | 57.8 |
| IRM | 72.0 ± 0.1 | 72.5 ± 0.3 | 58.5 ± 3.3 | 67.7 |
| GroupDRO | 73.5 ± 0.3 | 73.0 ± 0.3 | 36.8 ± 2.8 | 61.1 |
| Mixup | 72.5 ± 0.2 | 73.9 ± 0.4 | 28.6 ± 0.2 | 58.4 |
| MLDG | 71.9 ± 0.3 | 73.5 ± 0.2 | 29.1 ± 0.9 | 58.2 |
| CORAL | 71.1 ± 0.2 | 73.4 ± 0.2 | 31.1 ± 1.6 | 58.6 |
| MMD | 69.0 ± 2.3 | 70.4 ± 1.6 | 50.6 ± 0.2 | 63.3 |
| DANN | 72.4 ± 0.5 | 73.9 ± 0.5 | 24.9 ± 2.7 | 57.0 |
| CDANN | 71.8 ± 0.5 | 72.9 ± 0.1 | 33.8 ± 6.4 | 59.5 |
| MTL | 71.2 ± 0.2 | 73.5 ± 0.2 | 28.0 ± 0.6 | 57.6 |
| SagNet | 72.1 ± 0.3 | 73.2 ± 0.3 | 29.4 ± 0.5 | 58.2 |
| ARM | 84.9 ± 0.9 | 76.8 ± 0.6 | 27.9 ± 2.1 | 63.2 |
| V-REx | 72.8 ± 0.3 | 73.0 ± 0.3 | 55.2 ± 4.0 | 67.0 |
| RSC | 72.0 ± 0.1 | 73.2 ± 0.1 | 30.2 ± 1.6 | 58.5 |
| AND-mask | 71.9 ± 0.6 | 73.6 ± 0.5 | 30.2 ± 1.4 | 58.6 |
| SAND-mask | 79.9 ± 3.8 | 75.9 ± 1.6 | 31.6 ± 1.1 | 62.3 |
| Fishr | 74.1 ± 0.6 | 73.3 ± 0.1 | 58.9 ± 3.7 | 68.8 |
| SelfReg | 71.3 ± 0.4 | 73.4 ± 0.2 | 29.3 ± 2.1 | 58.0 |
| IDM | 74.2 ± 0.6 | 73.5 ± 0.2 | 68.3 ± 2.5 | 72.0 |

Table 14: Detailed results on Rotated MNIST in DomainBed.

| Algorithm | 0 | 15 | 30 | 45 | 60 | 75 | Average |
|---|---|---|---|---|---|---|---|
| ERM | $95.3 \pm 0.2$ | $98.7 \pm 0.1$ | $98.9 \pm 0.1$ | $98.7 \pm 0.2$ | $98.9 \pm 0.0$ | $96.2 \pm 0.2$ | 97.8 |
| IRM | $94.9 \pm 0.6$ | $98.7 \pm 0.2$ | $98.6 \pm 0.1$ | $98.6 \pm 0.2$ | $98.7 \pm 0.1$ | $95.2 \pm 0.3$ | 97.5 |
| GroupDRO | $95.9 \pm 0.1$ | $99.0 \pm 0.1$ | $98.9 \pm 0.1$ | $98.8 \pm 0.1$ | $98.6 \pm 0.1$ | $96.3 \pm 0.4$ | 97.9 |
| Mixup | $95.8 \pm 0.3$ | $98.7 \pm 0.0$ | $99.0 \pm 0.1$ | $98.8 \pm 0.1$ | $98.8 \pm 0.1$ | $96.6 \pm 0.2$ | 98.0 |
| MLDG | $95.7 \pm 0.2$ | $98.9 \pm 0.1$ | $98.8 \pm 0.1$ | $98.9 \pm 0.1$ | $98.6 \pm 0.1$ | $95.8 \pm 0.4$ | 97.8 |
| CORAL | $96.2 \pm 0.2$ | $98.8 \pm 0.1$ | $98.8 \pm 0.1$ | $98.8 \pm 0.1$ | $98.9 \pm 0.1$ | $96.4 \pm 0.2$ | 98.0 |
| MMD | $96.1 \pm 0.2$ | $98.9 \pm 0.0$ | $99.0 \pm 0.0$ | $98.8 \pm 0.0$ | $98.9 \pm 0.0$ | $96.4 \pm 0.2$ | 98.0 |
| DANN | $95.9 \pm 0.1$ | $98.9 \pm 0.1$ | $98.6 \pm 0.2$ | $98.7 \pm 0.1$ | $98.9 \pm 0.0$ | $96.3 \pm 0.3$ | 97.9 |
| CDANN | $95.9 \pm 0.2$ | $98.8 \pm 0.0$ | $98.7 \pm 0.1$ | $98.9 \pm 0.1$ | $98.8 \pm 0.1$ | $96.1 \pm 0.3$ | 97.9 |
| MTL | $96.1 \pm 0.2$ | $98.9 \pm 0.0$ | $99.0 \pm 0.0$ | $98.7 \pm 0.1$ | $99.0 \pm 0.0$ | $95.8 \pm 0.3$ | 97.9 |
| SagNet | $95.9 \pm 0.1$ | $99.0 \pm 0.1$ | $98.9 \pm 0.1$ | $98.6 \pm 0.1$ | $98.8 \pm 0.1$ | $96.3 \pm 0.1$ | 97.9 |
| ARM | $95.9 \pm 0.4$ | $99.0 \pm 0.1$ | $98.8 \pm 0.1$ | $98.9 \pm 0.1$ | $99.1 \pm 0.1$ | $96.7 \pm 0.2$ | 98.1 |
| V-REx | $95.5 \pm 0.2$ | $99.0 \pm 0.0$ | $98.7 \pm 0.2$ | $98.8 \pm 0.1$ | $98.8 \pm 0.0$ | $96.4 \pm 0.0$ | 97.9 |
| RSC | $95.4 \pm 0.1$ | $98.6 \pm 0.1$ | $98.6 \pm 0.1$ | $98.9 \pm 0.0$ | $98.8 \pm 0.1$ | $95.4 \pm 0.3$ | 97.6 |
| AND-mask | $94.9 \pm 0.1$ | $98.8 \pm 0.1$ | $98.8 \pm 0.1$ | $98.7 \pm 0.2$ | $98.6 \pm 0.2$ | $95.5 \pm 0.2$ | 97.5 |
| SAND-mask | $94.7 \pm 0.2$ | $98.5 \pm 0.2$ | $98.6 \pm 0.1$ | $98.6 \pm 0.1$ | $98.5 \pm 0.1$ | $95.2 \pm 0.1$ | 97.4 |
| Fishr | $95.8 \pm 0.1$ | $98.3 \pm 0.1$ | $98.8 \pm 0.1$ | $98.6 \pm 0.3$ | $98.7 \pm 0.1$ | $96.5 \pm 0.1$ | 97.8 |
| SelfReg | $96.0 \pm 0.3$ | $98.9 \pm 0.1$ | $98.9 \pm 0.1$ | $98.9 \pm 0.1$ | $98.9 \pm 0.1$ | $96.8 \pm 0.1$ | 98.1 |
| IDM | $96.1 \pm 0.3$ | $98.7 \pm 0.1$ | $99.1 \pm 0.1$ | $98.9 \pm 0.1$ | $98.9 \pm 0.1$ | $96.6 \pm 0.1$ | 98.0 |

Table 15: Detailed results on VLCS in DomainBed.

| Algorithm | C | L | S | V | Average |
|---|---|---|---|---|---|
| ERM | $97.6 \pm 0.3$ | $67.9 \pm 0.7$ | $70.9 \pm 0.2$ | $74.0 \pm 0.6$ | 77.6 |
| IRM | $97.3 \pm 0.2$ | $66.7 \pm 0.1$ | $71.0 \pm 2.3$ | $72.8 \pm 0.4$ | 76.9 |
| GroupDRO | $97.7 \pm 0.2$ | $65.9 \pm 0.2$ | $72.8 \pm 0.8$ | $73.4 \pm 1.3$ | 77.4 |
| Mixup | $97.8 \pm 0.4$ | $67.2 \pm 0.4$ | $71.5 \pm 0.2$ | $75.7 \pm 0.6$ | 78.1 |
| MLDG | $97.1 \pm 0.5$ | $66.6 \pm 0.5$ | $71.5 \pm 0.1$ | $75.0 \pm 0.9$ | 77.5 |
| CORAL | $97.3 \pm 0.2$ | $67.5 \pm 0.6$ | $71.6 \pm 0.6$ | $74.5 \pm 0.0$ | 77.7 |
| MMD | $98.8 \pm 0.0$ | $66.4 \pm 0.4$ | $70.8 \pm 0.5$ | $75.6 \pm 0.4$ | 77.9 |
| DANN | $99.0 \pm 0.2$ | $66.3 \pm 1.2$ | $73.4 \pm 1.4$ | $80.1 \pm 0.5$ | 79.7 |
| CDANN | $98.2 \pm 0.1$ | $68.8 \pm 0.5$ | $74.3 \pm 0.6$ | $78.1 \pm 0.5$ | 79.9 |
| MTL | $97.9 \pm 0.7$ | $66.1 \pm 0.7$ | $72.0 \pm 0.4$ | $74.9 \pm 1.1$ | 77.7 |
| SagNet | $97.4 \pm 0.3$ | $66.4 \pm 0.4$ | $71.6 \pm 0.1$ | $75.0 \pm 0.8$ | 77.6 |
| ARM | $97.6 \pm 0.6$ | $66.5 \pm 0.3$ | $72.7 \pm 0.6$ | $74.4 \pm 0.7$ | 77.8 |
| V-REx | $98.4 \pm 0.2$ | $66.4 \pm 0.7$ | $72.8 \pm 0.1$ | $75.0 \pm 1.4$ | 78.1 |
| RSC | $98.0 \pm 0.4$ | $67.2 \pm 0.3$ | $70.3 \pm 1.3$ | $75.6 \pm 0.4$ | 77.8 |
| AND-mask | $98.3 \pm 0.3$ | $64.5 \pm 0.2$ | $69.3 \pm 1.3$ | $73.4 \pm 1.3$ | 76.4 |
| SAND-mask | $97.6 \pm 0.3$ | $64.5 \pm 0.6$ | $69.7 \pm 0.6$ | $73.0 \pm 1.2$ | 76.2 |
| Fishr | $97.6 \pm 0.7$ | $67.3 \pm 0.5$ | $72.2 \pm 0.9$ | $75.7 \pm 0.3$ | 78.2 |
| SelfReg | $97.9 \pm 0.4$ | $66.7 \pm 0.1$ | $73.5 \pm 0.7$ | $74.7 \pm 0.7$ | 78.2 |
| IDM | $97.6 \pm 0.3$ | $66.9 \pm 0.3$ | $71.8 \pm 0.5$ | $76.0 \pm 1.3$ | 78.1 |

Table 16: Detailed results on PACS in DomainBed.

| Algorithm | A | C | P | S | Average |
|---|---|---|---|---|---|
| ERM | $86.5 \pm 1.0$ | $81.3 \pm 0.6$ | $96.2 \pm 0.3$ | $82.7 \pm 1.1$ | 86.7 |
| IRM | $84.2 \pm 0.9$ | $79.7 \pm 1.5$ | $95.9 \pm 0.4$ | $78.3 \pm 2.1$ | 84.5 |
| GroupDRO | $87.5 \pm 0.5$ | $82.9 \pm 0.6$ | $97.1 \pm 0.3$ | $81.1 \pm 1.2$ | 87.1 |
| Mixup | $87.5 \pm 0.4$ | $81.6 \pm 0.7$ | $97.4 \pm 0.2$ | $80.8 \pm 0.9$ | 86.8 |
| MLDG | $87.0 \pm 1.2$ | $82.5 \pm 0.9$ | $96.7 \pm 0.3$ | $81.2 \pm 0.6$ | 86.8 |
| CORAL | $86.6 \pm 0.8$ | $81.8 \pm 0.9$ | $97.1 \pm 0.5$ | $82.7 \pm 0.6$ | 87.1 |
| MMD | $88.1 \pm 0.8$ | $82.6 \pm 0.7$ | $97.1 \pm 0.5$ | $81.2 \pm 1.2$ | 87.2 |
| DANN | $87.0 \pm 0.4$ | $80.3 \pm 0.6$ | $96.8 \pm 0.3$ | $76.9 \pm 1.1$ | 85.2 |
| CDANN | $87.7 \pm 0.6$ | $80.7 \pm 1.2$ | $97.3 \pm 0.4$ | $77.6 \pm 1.5$ | 85.8 |
| MTL | $87.0 \pm 0.2$ | $82.7 \pm 0.8$ | $96.5 \pm 0.7$ | $80.5 \pm 0.8$ | 86.7 |
| SagNet | $87.4 \pm 0.5$ | $81.2 \pm 1.2$ | $96.3 \pm 0.8$ | $80.7 \pm 1.1$ | 86.4 |
| ARM | $85.0 \pm 1.2$ | $81.4 \pm 0.2$ | $95.9 \pm 0.3$ | $80.9 \pm 0.5$ | 85.8 |
| V-REx | $87.8 \pm 1.2$ | $81.8 \pm 0.7$ | $97.4 \pm 0.2$ | $82.1 \pm 0.7$ | 87.2 |
| RSC | $86.0 \pm 0.7$ | $81.8 \pm 0.9$ | $96.8 \pm 0.7$ | $80.4 \pm 0.5$ | 86.2 |
| AND-mask | $86.4 \pm 1.1$ | $80.8 \pm 0.9$ | $97.1 \pm 0.2$ | $81.3 \pm 1.1$ | 86.4 |
| SAND-mask | $86.1 \pm 0.6$ | $80.3 \pm 1.0$ | $97.1 \pm 0.3$ | $80.0 \pm 1.3$ | 85.9 |
| Fishr | $87.9 \pm 0.6$ | $80.8 \pm 0.5$ | $97.9 \pm 0.4$ | $81.1 \pm 0.8$ | 86.9 |
| SelfReg | $87.5 \pm 0.1$ | $83.0 \pm 0.1$ | $97.6 \pm 0.1$ | $82.8 \pm 0.2$ | 87.7 |
| IDM | $88.0 \pm 0.3$ | $82.6 \pm 0.6$ | $97.6 \pm 0.4$ | $82.3 \pm 0.6$ | 87.6 |

Table 17: Detailed results on OfficeHome in DomainBed.

| Algorithm | A | C | P | R | Average |
|---|---|---|---|---|---|
| ERM | $61.7 \pm 0.7$ | $53.4 \pm 0.3$ | $74.1 \pm 0.4$ | $76.2 \pm 0.6$ | 66.4 |
| IRM | $56.4 \pm 3.2$ | $51.2 \pm 2.3$ | $71.7 \pm 2.7$ | $72.7 \pm 2.7$ | 63.0 |
| GroupDRO | $60.5 \pm 1.6$ | $53.1 \pm 0.3$ | $75.5 \pm 0.3$ | $75.9 \pm 0.7$ | 66.2 |
| Mixup | $63.5 \pm 0.2$ | $54.6 \pm 0.4$ | $76.0 \pm 0.3$ | $78.0 \pm 0.7$ | 68.0 |
| MLDG | $60.5 \pm 0.7$ | $54.2 \pm 0.5$ | $75.0 \pm 0.2$ | $76.7 \pm 0.5$ | 66.6 |
| CORAL | $64.8 \pm 0.8$ | $54.1 \pm 0.9$ | $76.5 \pm 0.4$ | $78.2 \pm 0.4$ | 68.4 |
| MMD | $60.4 \pm 1.0$ | $53.4 \pm 0.5$ | $74.9 \pm 0.1$ | $76.1 \pm 0.7$ | 66.2 |
| DANN | $60.6 \pm 1.4$ | $51.8 \pm 0.7$ | $73.4 \pm 0.5$ | $75.5 \pm 0.9$ | 65.3 |
| CDANN | $57.9 \pm 0.2$ | $52.1 \pm 1.2$ | $74.9 \pm 0.7$ | $76.2 \pm 0.2$ | 65.3 |
| MTL | $60.7 \pm 0.8$ | $53.5 \pm 1.3$ | $75.2 \pm 0.6$ | $76.6 \pm 0.6$ | 66.5 |
| SagNet | $62.7 \pm 0.5$ | $53.6 \pm 0.5$ | $76.0 \pm 0.3$ | $77.8 \pm 0.1$ | 67.5 |
| ARM | $58.8 \pm 0.5$ | $51.8 \pm 0.7$ | $74.0 \pm 0.1$ | $74.4 \pm 0.2$ | 64.8 |
| V-REx | $59.6 \pm 1.0$ | $53.3 \pm 0.3$ | $73.2 \pm 0.5$ | $76.6 \pm 0.4$ | 65.7 |
| RSC | $61.7 \pm 0.8$ | $53.0 \pm 0.9$ | $74.8 \pm 0.8$ | $76.3 \pm 0.5$ | 66.5 |
| AND-mask | $60.3 \pm 0.5$ | $52.3 \pm 0.6$ | $75.1 \pm 0.2$ | $76.6 \pm 0.3$ | 66.1 |
| SAND-mask | $59.9 \pm 0.7$ | $53.6 \pm 0.8$ | $74.3 \pm 0.4$ | $75.8 \pm 0.5$ | 65.9 |
| Fishr | $63.4 \pm 0.8$ | $54.2 \pm 0.3$ | $76.4 \pm 0.3$ | $78.5 \pm 0.2$ | 68.2 |
| SelfReg | $64.2 \pm 0.6$ | $53.6 \pm 0.7$ | $76.7 \pm 0.3$ | $77.9 \pm 0.5$ | 68.1 |
| IDM | $64.4 \pm 0.3$ | $54.4 \pm 0.6$ | $76.5 \pm 0.3$ | $78.0 \pm 0.4$ | 68.3 |

Table 18: Detailed results on TerraIncognita in DomainBed.

| Algorithm | L100 | L38 | L43 | L46 | Average |
|---|---|---|---|---|---|
| ERM | $59.4 \pm 0.9$ | $49.3 \pm 0.6$ | $60.1 \pm 1.1$ | $43.2 \pm 0.5$ | 53.0 |
| IRM | $56.5 \pm 2.5$ | $49.8 \pm 1.5$ | $57.1 \pm 2.2$ | $38.6 \pm 1.0$ | 50.5 |
| GroupDRO | $60.4 \pm 1.5$ | $48.3 \pm 0.4$ | $58.6 \pm 0.8$ | $42.2 \pm 0.8$ | 52.4 |
| Mixup | $67.6 \pm 1.8$ | $51.0 \pm 1.3$ | $59.0 \pm 0.0$ | $40.0 \pm 1.1$ | 54.4 |
| MLDG | $59.2 \pm 0.1$ | $49.0 \pm 0.9$ | $58.4 \pm 0.9$ | $41.4 \pm 1.0$ | 52.0 |
| CORAL | $60.4 \pm 0.9$ | $47.2 \pm 0.5$ | $59.3 \pm 0.4$ | $44.4 \pm 0.4$ | 52.8 |
| MMD | $60.6 \pm 1.1$ | $45.9 \pm 0.3$ | $57.8 \pm 0.5$ | $43.8 \pm 1.2$ | 52.0 |
| DANN | $55.2 \pm 1.9$ | $47.0 \pm 0.7$ | $57.2 \pm 0.9$ | $42.9 \pm 0.9$ | 50.6 |
| CDANN | $56.3 \pm 2.0$ | $47.1 \pm 0.9$ | $57.2 \pm 1.1$ | $42.4 \pm 0.8$ | 50.8 |
| MTL | $58.4 \pm 2.1$ | $48.4 \pm 0.8$ | $58.9 \pm 0.6$ | $43.0 \pm 1.3$ | 52.2 |
| SagNet | $56.4 \pm 1.9$ | $50.5 \pm 2.3$ | $59.1 \pm 0.5$ | $44.1 \pm 0.6$ | 52.5 |
| ARM | $60.1 \pm 1.5$ | $48.3 \pm 1.6$ | $55.3 \pm 0.6$ | $40.9 \pm 1.1$ | 51.2 |
| V-REx | $56.8 \pm 1.7$ | $46.5 \pm 0.5$ | $58.4 \pm 0.3$ | $43.8 \pm 0.3$ | 51.4 |
| RSC | $59.9 \pm 1.4$ | $46.7 \pm 0.4$ | $57.8 \pm 0.5$ | $44.3 \pm 0.6$ | 52.1 |
| AND-mask | $54.7 \pm 1.8$ | $48.4 \pm 0.5$ | $55.1 \pm 0.5$ | $41.3 \pm 0.6$ | 49.8 |
| SAND-mask | $56.2 \pm 1.8$ | $46.3 \pm 0.3$ | $55.8 \pm 0.4$ | $42.6 \pm 1.2$ | 50.2 |
| Fishr | $60.4 \pm 0.9$ | $50.3 \pm 0.3$ | $58.8 \pm 0.5$ | $44.9 \pm 0.5$ | 53.6 |
| SelfReg | $60.0 \pm 2.3$ | $48.8 \pm 1.0$ | $58.6 \pm 0.8$ | $44.0 \pm 0.6$ | 52.8 |
| IDM | $60.1 \pm 1.4$ | $48.8 \pm 1.9$ | $57.9 \pm 0.2$ | $44.3 \pm 1.2$ | 52.8 |

Table 19: Detailed results on DomainNet in DomainBed.

| Algorithm | clip | info | paint | quick | real | sketch | Average |
|---|---|---|---|---|---|---|---|
| ERM | $58.6 \pm 0.3$ | $19.2 \pm 0.2$ | $47.0 \pm 0.3$ | $13.2 \pm 0.2$ | $59.9 \pm 0.3$ | $49.8 \pm 0.4$ | 41.3 |
| IRM | $40.4 \pm 6.6$ | $12.1 \pm 2.7$ | $31.4 \pm 5.7$ | $9.8 \pm 1.2$ | $37.7 \pm 9.0$ | $36.7 \pm 5.3$ | 28.0 |
| GroupDRO | $47.2 \pm 0.5$ | $17.5 \pm 0.4$ | $34.2 \pm 0.3$ | $9.2 \pm 0.4$ | $51.9 \pm 0.5$ | $40.1 \pm 0.6$ | 33.4 |
| Mixup | $55.6 \pm 0.1$ | $18.7 \pm 0.4$ | $45.1 \pm 0.5$ | $12.8 \pm 0.3$ | $57.6 \pm 0.5$ | $48.2 \pm 0.4$ | 39.6 |
| MLDG | $59.3 \pm 0.1$ | $19.6 \pm 0.2$ | $46.8 \pm 0.2$ | $13.4 \pm 0.2$ | $60.1 \pm 0.4$ | $50.4 \pm 0.3$ | 41.6 |
| CORAL | $59.2 \pm 0.1$ | $19.9 \pm 0.2$ | $47.4 \pm 0.2$ | $14.0 \pm 0.4$ | $59.8 \pm 0.2$ | $50.4 \pm 0.4$ | 41.8 |
| MMD | $32.2 \pm 13.3$ | $11.2 \pm 4.5$ | $26.8 \pm 11.3$ | $8.8 \pm 2.2$ | $32.7 \pm 13.8$ | $29.0 \pm 11.8$ | 23.5 |
| DANN | $53.1 \pm 0.2$ | $18.3 \pm 0.1$ | $44.2 \pm 0.7$ | $11.9 \pm 0.1$ | $55.5 \pm 0.4$ | $46.8 \pm 0.6$ | 38.3 |
| CDANN | $54.6 \pm 0.4$ | $17.3 \pm 0.1$ | $44.2 \pm 0.7$ | $12.8 \pm 0.2$ | $56.2 \pm 0.4$ | $45.9 \pm 0.5$ | 38.5 |
| MTL | $58.0 \pm 0.4$ | $19.2 \pm 0.2$ | $46.2 \pm 0.1$ | $12.7 \pm 0.2$ | $59.9 \pm 0.1$ | $49.0 \pm 0.0$ | 40.8 |
| SagNet | $57.7 \pm 0.3$ | $19.1 \pm 0.1$ | $46.3 \pm 0.5$ | $13.5 \pm 0.4$ | $58.9 \pm 0.4$ | $49.5 \pm 0.2$ | 40.8 |
| ARM | $49.6 \pm 0.4$ | $16.5 \pm 0.3$ | $41.5 \pm 0.8$ | $10.8 \pm 0.1$ | $53.5 \pm 0.3$ | $43.9 \pm 0.4$ | 36.0 |
| V-REx | $43.3 \pm 4.5$ | $14.1 \pm 1.8$ | $32.5 \pm 5.0$ | $9.8 \pm 1.1$ | $43.5 \pm 5.6$ | $37.7 \pm 4.5$ | 30.1 |
| RSC | $55.0 \pm 1.2$ | $18.3 \pm 0.5$ | $44.4 \pm 0.6$ | $12.5 \pm 0.1$ | $55.7 \pm 0.7$ | $47.8 \pm 0.9$ | 38.9 |
| AND-mask | $52.3 \pm 0.8$ | $17.3 \pm 0.5$ | $43.7 \pm 1.1$ | $12.3 \pm 0.4$ | $55.8 \pm 0.4$ | $46.1 \pm 0.8$ | 37.9 |
| SAND-mask | $43.8 \pm 1.3$ | $15.2 \pm 0.2$ | $38.2 \pm 0.6$ | $9.0 \pm 0.2$ | $47.1 \pm 1.1$ | $39.9 \pm 0.6$ | 32.2 |
| Fishr | $58.3 \pm 0.5$ | $20.2 \pm 0.2$ | $47.9 \pm 0.2$ | $13.6 \pm 0.3$ | $60.5 \pm 0.3$ | $50.5 \pm 0.3$ | 41.8 |
| SelfReg | $60.7 \pm 0.1$ | $21.6 \pm 0.1$ | $49.5 \pm 0.1$ | $14.2 \pm 0.3$ | $60.7 \pm 0.1$ | $51.7 \pm 0.1$ | 43.1 |
| IDM | $58.8 \pm 0.3$ | $20.7 \pm 0.2$ | $48.3 \pm 0.1$ | $13.7 \pm 0.4$ | $59.1 \pm 0.1$ | $50.2 \pm 0.3$ | 41.8 |

