# OpenReview forum: "Provable Domain Generalization via Information Theory Guided Distribution Matching"
_ICLR.cc/2024/Conference — Submitted to ICLR 2024_

### Official Review · Reviewer_rTAB · 2023-10-31

**Soundness:** 3 good
**Presentation:** 3 good
**Contribution:** 3 good
**Rating:** 6
**Confidence:** 4

**Summary:**

This paper provides an information-theoretic analysis of the difference between the training and test-domain population risks for the domain generalization (DG) problem. Specifically, different upper bounds for both source and target-domain generalization errors are presented, revealing the key information quantities that control the capability of learning algorithms to generalize on unseen domains. Motivated by the theoretical analysis, this paper proposes the Inter-domain Distribution Matching (IDM) algorithm for high-probability DG by simultaneously aligning inter-domain gradients and representations, and Per-sample Distribution Matching (PDM) for high-dimensional and complex data distribution alignment. Experimental results are provided to validate the efficacy of the proposed algorithm.

**Strengths:**

1. The information-theoretic analysis provided by the paper is quite inspiring. It separates the difference between L_tr and L_te using L(W), and constructs generalization error bounds for source-domain/target domain population risk, respectively.
2. The Per-sample Distribution Matching (PDM) idea is cute, and the connection to the slicing technique is quite interesting.
3. The proposed algorithm is justified by the information-theoretic analysis, and it works well on multiple datasets.

**Weaknesses:**

1. From the definitions of L_tr and L_te, this paper assumes that we have an infinite number of samples for each domain so that we can evaluate the population risk of each domain. This means that only the generalization gap caused by finite number of domains is considered, but the generalization gap due to finite number of samples is ignored. It is fine to only focus on the gap between L_tr and L_te, but it should be mentioned that in practice we also need to deal with the standard generalization error gap, which cannot be handled by the method proposed in the paper.

2. It seems that the assumption in Theorem 7 seldom holds in practice. Note that R is a representation of X, so for any fixed d, $I(Y;X|D=d)\ge I(Y;R|D=d)$ by data processing inequality. Take expectation over D, we have $I(Y;X|D)\ge I(Y;R|D)$, and therefore $H(Y|X,D)\le H(Y|R,D)$.
In the discussion after Theorem 7, it is said that “$I(Y;D|R)$ is hard to estimate, and minimizing the covariate shift $I(R;D)$ solely is sufficient.” Note that $I(Y;D|X)$ is not a lower bound for $I(Y;D|R)$ in general. Why does the proposed algorithm only minimize $I(R_i;D_i)$?  From my understanding, we should minimize $I(Y;D|R)$ together with $I(R;D)$, which corresponds to the invariant risk minimization or sufficiency condition in fairness literature.

**Questions:**

1. The information-theoretic generalization bound presented in Theorem 2 looks similar to Propositions 1 and 2 in the following paper. Roughly speaking, the result presented in this paper can be viewed as the multi-domain version of the standard generalization error bound in supervised learning by replacing the mutual information $I(W;Z_i)$ with $I(W;D_i)$. Such a connection should be discussed in the paper.

Bu, Yuheng, Shaofeng Zou, and Venugopal V. Veeravalli. "Tightening mutual information-based bounds on generalization error." IEEE Journal on Selected Areas in Information Theory 1, no. 1 (2020): 121-130.

2. The DG setting considered here is also related to meta-learning. Some other references on information-theoretic analysis for meta-learning:

Jose, Sharu Theresa, and Osvaldo Simeone. "Information-theoretic generalization bounds for meta-learning and applications." Entropy 23, no. 1 (2021): 126.

Chen, Qi, Changjian Shui, and Mario Marchand. "Generalization bounds for meta-learning: An information-theoretic analysis." Advances in Neural Information Processing Systems 34 (2021): 25878-25890.

Hellström, Fredrik, and Giuseppe Durisi. "Evaluated CMI bounds for meta learning: Tightness and expressiveness." Advances in Neural Information Processing Systems 35 (2022): 20648-20660.

Bu, Yuheng, Harsha Vardhan Tetali, Gholamali Aminian, Miguel Rodrigues, and Gregory Wornell. "On the Generalization Error of Meta Learning for the Gibbs Algorithm." arXiv preprint arXiv:2304.14332 (2023).


Minor comments:
1. After equation (4) line 7, “Secondly, gradient alignment is not required when I(Y ;D|X)”, it should be when I(Y ;D|X)=0 ?

---

> ### Author Response · Authors · 2023-11-17
> **Response to Reviewer rTAB**
>
> Dear Reviewer rTAB, thank you for the valuable and insightful suggestions! We address the raised concerns as follows:
>
> **From the definitions of $L_{tr}$ and $L_{te}$, this paper assumes that we have an infinite number of samples for each domain so that we can evaluate the population risk of each domain. This means that only the generalization gap caused by finite number of domains is considered, but the generalization gap due to finite number of samples is ignored. It is fine to only focus on the gap between $L_{tr}$ and $L_{te}$, but it should be mentioned that in practice we also need to deal with the standard generalization error gap, which cannot be handled by the method proposed in the paper.**
>
> In fact, the population risks $L_{tr}$ and $L_{te}$ are only for the purpose of theoretical analysis. There is no need to evaluate them in practice, as our regularization does not involve these quantities. We agree that there exists a generalization gap caused by finite samples, but this is of different research interest and is beyond the scope of domain generalization. For completeness, we also provide generalization bounds considering the effect of finite samples, as discussed in Section D.4. But for simplicity and conciseness, we only focus on the domain generalization gap and do not include these results in the main text.
>
> **It seems that the assumption in Theorem 7 seldom holds in practice. Note that R is a representation of X, so for any fixed d, $I(Y;X|D=d)\ge I(Y;R|D=d)$ by data processing inequality. Take expectation over D, we have $I(Y;X|D)\ge I(Y;R|D)$, and therefore $H(Y|X,D)\le H(Y|R,D)$. In the discussion after Theorem 7, it is said that “$I(Y;D|R)$ is hard to estimate, and minimizing the covariate shift $I(R;D)$ solely is sufficient.” Note that $I(Y;D|X)$ is not a lower bound for $I(Y;D|R)$ in general. Why does the proposed algorithm only minimize $I(R_i;D_i)$? From my understanding, we should minimize $I(Y;D|R)$ together with $I(R;D)$, which corresponds to the invariant risk minimization or sufficiency condition in fairness literature.**
>
> As suggested by the learning theory of information bottleneck (Tishby et al., 2015), the network is encouraged to maximize $I(R;Y)$ while minimizing $I(R;X)$, and the representation $R$ should be the minimal sufficient statistic of $X$ w.r.t. $Y$ for a well-trained network. Therefore, it is likely that at the end of the training progress, we have $I(R;Y|D) \approx I(X;Y|D)$.
>
> The main motivation for minimizing $I(R;D)$ solely is due to Theorem 5, which suggests that minimizing the covariate shift solely is sufficient to upper bound the target-domain population risk. This also applies to the representation space covariate shift, i.e. $I(R;D)$. Note that this conclusion does not require $I(Y;D|R) \ge I(Y;D|X)$ to hold.
>
> It is noteworthy that the concept shift $I(Y;D|R)$ is instead addressed by gradient matching. From the definition, $I(Y;D|R)$ is raised by the inconsistency of the correlations between the representation $R$ and the label $Y$ (i.e. $P_{Y|R}$) across different domains. As we discussed after Theorem 2, minimizing $I(W;D_i)$ (i.e. gradient matching) encourages the model to discard such domain-specific correlations and achieve satisfactory generalization performance.
>
> Tishby N, Zaslavsky N. Deep learning and the information bottleneck principle.
>
> **The information-theoretic generalization bound presented in Theorem 2 looks similar to Propositions 1 and 2 in the following paper. Roughly speaking, the result presented in this paper can be viewed as the multi-domain version of the standard generalization error bound in supervised learning by replacing the mutual information $I(W;Z_i)$ with $I(W;D_i)$. Such a connection should be discussed in the paper. The DG setting considered here is also related to meta-learning.**
>
> We replenish further discussions in Section D.4 about the connection between our results and standard generalization error bounds, and also the connection to meta-learning.

---

### Official Review · Reviewer_pEKy · 2023-11-01

**Soundness:** 1 poor
**Presentation:** 2 fair
**Contribution:** 1 poor
**Rating:** 1
**Confidence:** 5

**Summary:**

Authors propose a new probabilistic framework of domain generalization and propose IDM and PDM with theoretical analyses on information theory. They conduct experiments on DomainBed to confirm the effectiveness.

**Strengths:**

The idea is generally easy to follow and theoretical contributions are made.

**Weaknesses:**

Theory:

- Assumption 1 is quite confusing "The target domains $D_{te}$ are independent of source domains $D_{tr}$" Here, "independent" does not seem well-defined.

Algorithm:

- The operation of diving data points into separate dimensions assume the independence between these dimensions. It is not obvious whether this assumption is realistic or not.
- If the representation has already been aligned, why is the gradient alignment needed?
- Fishr [1] also aligns the gradients across domains, reducing the algorithm's novelty.

Experiments:

- An important baseline SWAD [2] is missing in Table 2.
- In the ablation study of Table 6, removing GA (gradient alignment) brings increase to the performance. Is gradient alignment really needed?
- In Table 2, IDM brings the biggest increase on CMNIST compared with Fishr, while CMNIST is a half-synthetic and simple dataset. This weakens the demonstrations of the algorithm's effectiveness.

Others:

- Writing and presentation could be improved. For example, the implementation of algorithms should be put in the main body instead of the appendix to make it clearer. The subscript $ma$ in $X_{ma}$ in Algorithm 1 may be mistakenly interpreted as $m$ and $a$ since $m$ represents the number of training domains.


[1] Rame A, Dancette C, Cord M. Fishr: Invariant gradient variances for out-of-distribution generalization[C]//International Conference on Machine Learning. PMLR, 2022: 18347-18377.

[2] Cha J, Chun S, Lee K, et al. Swad: Domain generalization by seeking flat minima[J]. Advances in Neural Information Processing Systems, 2021, 34: 22405-22418.

**Questions:**

Please refer to weaknesses.

## Post-Rebuttal

After going through authors' response to me and other reviewers' feedback along with authors' corresponding responses, I find new major concerns, stated in detail in my responses. Considering the new concerns along with the non-technical issues, I find that I still overestimated the contributions of this work in my initial review, so I decide to lower my score from 3 to 1 after a more careful review.

---

> ### Author Response · Authors · 2023-11-17
> **Response to Reviewer pEKy (1/2)**
>
> Dear Reviewer pEKy, thanks for your valuable comments! We would like to clarify certain misconceptions about our work:
>
> **Assumption 1 is quite confusing "The target domains $D_{te}$ are independent of source domains $D_{tr}$" Here, "independent" does not seem well-defined.**
>
> Independence is an essential concept in probability theory. When two random variables $X$, $Y$ are independent, it means that their joint distribution can be acquired by multiplying their marginal distributions, i.e. $P_{X,Y} = P_X P_Y$. Specialized to our problem, we are assuming that $P_{D_{tr},D_{te}} = P_{D_{tr}} P_{D_{te}}$. It is a common assumption in machine learning theory, e.g. one usually assumes that the training and test samples are i.i.d. This assumption has also been adopted in DG analysis, e.g. (Eastwood et al., 2022), where the domains are assumed to be i.i.d sampled.
>
> Cian Eastwood, Alexander Robey, Shashank Singh, Julius Von Kugelgen, Hamed Hassani, George J Pappas, and Bernhard Scholkopf. Probable domain generalization via quantile risk minimization.
>
> **The operation of diving data points into separate dimensions assume the independence between these dimensions. It is not obvious whether this assumption is realistic or not.**
>
> In fact, the independence between different dimensions is **not necessary** to apply our algorithm. By dividing and aligning the data points in each dimension, we can align each marginal distribution of the multivariate distribution. This simplification is also widely adopted in previous works, e.g. AND-mask, SAND-mask, IGA, Fishr. Although this may not be sufficient to align the entire distribution when these dimensions are dependent, it already meets our needs to minimize the training-domain population risk and achieve satisfactory performance. As seen in Table 10, PDM outperforms previous distribution matching methods based on moment matching (IGA and Fishr).
>
> **If the representation has already been aligned, why is the gradient alignment needed?**
>
> When the representations are aligned, it only guarantees that there are no covariate shifts, i.e. $I(R;D) = 0$. However, we aim to address the entire distribution shift, which further requires addressing the concept shift $I(Y;D|R)$. This indicates that even though the representations are aligned, the correlations between the representation $R$ and the label $Y$, i.e. $P_{Y|R}$, may still vary across different domains. As we discussed after Theorem 2, minimizing $I(W;D_i)$ (i.e. gradient matching) encourages the model to discard such domain-specific correlations and achieve domain generalization.
>
> **Fishr [1] also aligns the gradients across domains, reducing the algorithm's novelty.**
>
> We agree that the ideas of gradient or representation matching have already been explored in previous works. However, we highlight that this is the first work to give a **clear explanation** of why gradient or representation matching helps domain generalization from an information-theoretic perspective, without any **stringent assumptions** on the loss landscape, the features, or the gradients required in previous analyses (e.g. Fishr, IGA, AND-mask). Our theoretical analysis proves that gradient and representation alignment together is a **sufficient condition** to solve the domain generalization problem, which has never been explored in previous works. We also point out the key defect of previous distribution matching methods (for both representations and gradients) built upon moment matching, which is then solved by the proposed PDM method.
>
> **An important baseline SWAD [2] is missing in Table 2.**
>
> It is important to point out that SWAD is more of a **model selection** method than a DG algorithm. In this paper, we follow the settings of DomainBed and use test-domain validation for model selection. Since SWAD requires accessing historical validation accuracies in each epoch, which is not available in DomainBed using test-domain validation, the performance is not generally comparable. Actually, as a model selection strategy, SWAD can work seamlessly with any existing DG algorithms to achieve better performance, including IDM. This is also of great interest to our research but is beyond the scope of this paper.

---

> ### Author Response · Authors · 2023-11-17
> **Response to Reviewer pEKy (2/2)**
>
> **In the ablation study of Table 6, removing GA (gradient alignment) brings increase to the performance. Is gradient alignment really needed?**
>
> As seen in Table 5, gradient matching greatly enhances the performance on ColoredMNIST (57.7\% to 72.0\%). Additionally, we conduct ablation studies to examine the effect of gradient matching on other realistic datasets, showing **significant performance gain** on VLCS (Table 7, 77.4\% to 78.1\%) and PACS (Table 8, 86.8\% to 87.6\%). Meanwhile, the accuracy drop on OfficeHome is also mild (68.4\% to 68.3\%) compared to performance gain on other datasets. We believe the current results are sufficient to prove the effectiveness of gradient matching.
>
> **In Table 2, IDM brings the biggest increase on CMNIST compared with Fishr, while CMNIST is a half-synthetic and simple dataset. This weakens the demonstrations of the algorithm's effectiveness.**
>
> Although ColoredMNIST is synthetic in nature, it is the most ideal benchmark to evaluate the ability of DG algorithms to tackle the concept shift, as seen in various works in the literature, e.g. IRM, V-Rex, Fishr.
>
> Meanwhile, improving accuracy on Colored MNIST is not easy, given that no algorithm actually demonstrated significant improvements over IRM in the past four years, with most algorithms even failing to achieve substantial improvement over ERM. To the best of our knowledge, Fishr is the only method that exhibits superior performance to IRM, yet the improvement does not surpass the standard deviation. In contrast, IDM significantly outperforms IRM on Colored MNIST.
>
> Moreover, IDM not only achieves the highest accuracy on CMNIST, but also the highest average accuracy over $7$ datasets, and the best rankings (mean, median, and worst). Most importantly, IDM consistently achieves top accuracies on DomainBed (the worst ranking is 6), while all other algorithms fail to outperform 2/3 of the competitors on at least 1 dataset (the second-best worst ranking is 14 among 22 algorithms). This indicates that IDM is the only algorithm that is consistently among the best methods under various distribution shifts.

---

> ### Comment · Reviewer_pEKy · 2023-11-20
> **Reply to authors' response**
>
> > The target domains $D_{te}$ are independent of source domains $D_{tr}$. Specialized to our problem, we are assuming that $P_{D_{tr},D_{te}}=P_{D_{tr}}P_{D_{te}}$.
>
> What I am confused about is not the basic concept of independence, but the independence between "domains". For samples in a specific domain, the independence is obvious, but how do you define the independence of domains? If you define them as $P_{D_{tr},D_{te}}=P_{D_{tr}}P_{D_{te}}$, based on the notations in your paper, $D_{tr}$ and $D_{te}$ seem like two distributions instead of samples (whereas $S_{tr}$ seems to be the training samples), how do you define independence of distributions? This requires clearer explanations.
>
> > It is a common assumption in machine learning theory, e.g. one usually assumes that the training and test samples are i.i.d.
> >
> > This assumption has also been adopted in DG analysis, e.g. (Eastwood et al., 2022), where the domains are assumed to be i.i.d sampled.
>
> The i.i.d assumption in the scope of DG, also the common assumption in traditional machine learning, mainly refers to that training data and test data are identically and independently distributed [1,2]. Note that "i.i.d." include not only "independently" but also "identically", here indicating that test data follows the identical distribution as training data [1,2], which definitely cannot be satisfied in DG/OOD since this is exactly the problem DG/OOD tries to address.
>
> I suppose you mean domains are i.i.d. sampled, which is adopted by Eastwood et al. [3], but this is not as easy to be satisfied as claimed in your paper "we propose the following high-probability objective by leveraging the mild Assumption 1, which is trivially satisfied in practice". You may refer to Section 7 of [3] for the discussion on "The assumption of i.i.d. domains".
>
> > It is important to point out that SWAD is more of a **model selection** method than a DG algorithm.
>
> SWAD is an algorithm seeking flat minima through averaging networks weights from different iterations [4]. It is less sensitive to model selection than ERM, but it is not a model selection method since the model it obtains is different from any models in the pool to be selected in the learning process of ERM (i.e. checkpoints in learning the process of ERM), but a model averaged from them.
>
> > Since SWAD requires accessing historical validation accuracies in each epoch, which is not available in DomainBed using test-domain validation.
>
> This can be easily achieved since you can record historical test-domain validation accuracies. To say the least, you can add SWAD to table 10 where training-domain validation is adopted, and by no means should SWAD be directly ignored without even citing.
>
> > SWAD can work seamlessly with any existing DG algorithms to achieve better performance.
>
> It is fine to not compare IDM with SWAD directly, but considering that SWAD is an important baseline and one of the most cited works in DG in recent two years, it will be persuasive if you compare SWAD+IDM with SWAD and show the improvement.
>
> > Additionally, we conduct ablation studies to examine the effect of gradient matching on other realistic datasets, showing **significant performance gain** on VLCS (Table 7, 77.4% to 78.1%) and PACS (Table 8, 86.8% to 87.6%).
>
> I appreciate that the improvement can be seen on VLCS and PACS for gradient matching. However, there is no need to emphasize that improvements lower than 1 pp are significant.
>
> > Although ColoredMNIST is synthetic in nature, it is the most ideal benchmark to evaluate the ability of DG algorithms to tackle the concept shift, as seen in various works in the literature, e.g. IRM, V-Rex, Fishr.
>
> It is true that ColoredMNIST can better evaluate OOD generalization ability under concept shift. However, as widely recognized, in visual data, covariate shift generally prevails instead of concept shift, which is also claimed in your paper". While CMNIST manually induces high concept shift, covariate shift is instead dominant in other datasets". It is far from persuasive when most improvements are made in terms of concept shift in the are of DG.
>
> > Meanwhile, improving accuracy on Colored MNIST is not easy, given that no algorithm actually demonstrated significant improvements over IRM in the past four years
>
> This could be true for DG since DG still mainly focuses on tackling covariate shift. However, in the area of invariant learning, many algorithms demonstrate significant improvements over IRM on Colored MNIST. To name a few: SparseIRM [5], EIIL [6], MAPLE [7], MRM [8]. If you really emphasize the improvement under strong concept shifts, which is beyond the main scope of DG, these methods should also be compared.

---

> ### Comment · Reviewer_pEKy · 2023-11-20
> **Additional questions**
>
> After going through the reference [3] that authors mentioned in their rebuttal, I come up with additional comments and questions:
>
> - "In this paper, we formulate DG from a novel probabilistic perspective". In fact, Eastwood et al. also propose a new probabilistic perspective and objective for DG [3], decreasing the novelty of authors' problem formulation.
> - "we provide key insights into high-probability DG by showing that the input-output mutual information of the learning algorithm and the extent of distribution shift together control the gap between training and test-domain population risks." To my understanding, the input-output mutual information can be considered as the predictive ability of the models on training data. From this perspective, this insight shares a similar conclusion with the bound of DA [9] that includes a term of source domain risk and the distribution divergence, which is not a novel or surprising insight.
> - "We start by demonstrating that the achievable level of average-case risk $L(w)$ is constrained by the degree of concept shift". This is also not a new insight. Proposition 1 in [11] shows a similar implication.
> - Given the presence of [11], theoretical analyses in this paper, although newly made, are mainly based on the theoretical framework of [11], which further decreases the theoretical contributions.
> - After going through the paper more detailedly and checking the comments of other reviewers and authors' corresponding responses, another major weakness is that most improvements can only be demonstrated under the usage of test-domain validation. Referring to Table 10 using training-domain validation, there is only marginal improvements even compared with simple ERM. As suggested by DomainBed [10], this violates the basic condition that no test data is accessible during the training and validation stage. If test data is available, why not use DA algorithms directly?
>
> Some minor issues:
>
> - "Our formulation leverages the mild identical distribution assumption of the environments and enables direct optimization. " What does the identical distribution assumption here refer to?
> - "IDM jointly working with PDM achieves superior performance on the Colored MNIST dataset (Arjovsky et al., 2019) and the DomainBed benchmark (Gulrajani & Lopez-Paz, 2020).". Colored MNIST itself is already included in DomainBed benchmark [10], thus they do not be mentioned separately.
> - "The source domains $D_{tr}=\\{ D_i \\}_{i=1}^m$
>
> and target domains $D_{te}=\\{ D_k \\}_{k=1}^{m'}$
>
> are both randomly sampled from $ν$. "
>
> Here domains in train and test should be distinguished in terms of notations, e.g.  $D_{tr}=\\{D_i^{tr}\\}_{i=1}^m$.
>
> - "With each subset $S_i=\\{Z_j^i\\}_{j=1}^n$"
>
> Here the upscript $n$ should be able to indicate the domain index, e.g. $S_i=\\{Z_j^i\\}_{j=1}^{n_i}$.
> - In problem 1 "$L_{tr}(W)$" here I suppose $W$ is also the parameter of a specific predictor $f_w$ where $w\in\mathcal{W}$ as introduced in Sec 2. Generally, upper case letters and lower case letters may have different meanings, thus they should not be used interchangeably.
> - "Theorem 1. For any predictor $Q_{Y|X}$" The notation $Q_{Y|X}$​ is not introduced in notations of Sec 2, seeming like a probability distribution, while in Sec 2 the concept of "predictor" is a determinant function "each $w\in \mathcal{W}$ characterizes a predictor $f_w$ mapping from $\mathcal{X}$ to $\mathcal{Y}$.
> - "Specialized to our problem, we quantify the changes in the hypothesis" From the Theorem 2, I suppose here should be changes in the risk or loss instead of the hypothesis?

---

> ### Comment · Reviewer_pEKy · 2023-11-20
> **Non-technical issues**
>
> - It is confusing that why the revision of pdf is not mentioned in your reply at all. For your reply to "Is gradient alignment really needed", the ablation studies of VLCS and PACS (Table 7 and 8) cannot be found in the original pdf, confusing me at at first, and I find that a new version of paper has been uploaded without informing the reviewers in a common response. If not carefully checking, it seems like as if these ablation studies were present in the submitted version. Of course it is beneficial and encouraged to provide a revision with additional experiments for rebuttal, but it is questionable to provide a revision without informing the reviewers clearly.
> - There is no need to emphasize "Independence is an essential concept in probability theory" as if any of us does not know such a basic concept. Obviously all reviewers and chairs are quite familiar with that.
> - The part of writing and presentation issues in my original review seems to be ignored without being mentioned in authors' reply.

---

> ### Comment · Reviewer_pEKy · 2023-11-20
> **Summary of above comments**
>
> After going through authors' response to me and other reviewers' feedback along with authors' corresponding responses, I find new major concerns:
>
> - Empirical contributions are significantly decreased since experimental improvements are exhibited mainly under test-domain validation, which is not realistic in the setting of DG, while improvements are much lower in terms of training-domain validation. (Also mentioned by Reviewer pzid)
> - Contribution of the probabilistic formulation of DG is decreased after taking [3] into account.
> - Theoretical contributions are decreased after taking [11] into account. (Also mentioned by Reviewer rTAB)
>
> Other major concerns not solved by the current response from authors:
>
> - Minor improvement on datasets except ColoredMNIST. (Also mentioned by Reviewer G8me)
> - Missing comparison with the important baseline SWAD [4], even with no citation at all.
> - Too many unclarities in notations and concepts as stated above.
>
> Considering the new concerns along with the non-technical issues, I find that I still overestimated the contributions of this work in my initial review, so I decide to lower my score from 3 to 1 after a more careful review.
>
>
>
> ## References
>
> [1] Wang J, Lan C, Liu C, et al. Generalizing to unseen domains: A survey on domain generalization[J]. IEEE Transactions on Knowledge and Data Engineering, 2022.
>
> [2] Zhou K, Liu Z, Qiao Y, et al. Domain generalization: A survey[J]. IEEE Transactions on Pattern Analysis and Machine Intelligence, 2022.
>
> [3] Eastwood C, Robey A, Singh S, et al. Probable domain generalization via quantile risk minimization[J]. Advances in Neural Information Processing Systems, 2022, 35: 17340-17358.
>
> [4] Cha J, Chun S, Lee K, et al. Swad: Domain generalization by seeking flat minima[J]. Advances in Neural Information Processing Systems, 2021, 34: 22405-22418.
>
> [5] Zhou X, Lin Y, Zhang W, et al. Sparse invariant risk minimization[C]//International Conference on Machine Learning. PMLR, 2022: 27222-27244.
>
> [6] Creager E, Jacobsen J H, Zemel R. Environment inference for invariant learning[C]//International Conference on Machine Learning. PMLR, 2021: 2189-2200.
>
> [7] Zhou X, Lin Y, Pi R, et al. Model agnostic sample reweighting for out-of-distribution learning[C]//International Conference on Machine Learning. PMLR, 2022: 27203-27221.
>
> [8] Zhang D, Ahuja K, Xu Y, et al. Can subnetwork structure be the key to out-of-distribution generalization?[C]//International Conference on Machine Learning. PMLR, 2021: 12356-12367.
>
> [9] Ben-David S, Blitzer J, Crammer K, et al. A theory of learning from different domains[J]. Machine learning, 2010, 79: 151-175.
>
> [10] Gulrajani I, Lopez-Paz D. In Search of Lost Domain Generalization[C]//International Conference on Learning Representations. 2020.
>
> [11] Federici M, Tomioka R, Forré P. An information-theoretic approach to distribution shifts[J]. Advances in Neural Information Processing Systems, 2021, 34: 17628-17641.

---

> > ### Author Response · Authors · 2023-11-21
> > **Response to Reviewer pEKy**
> >
> > Dear Reviewer pEKy, we respect your effort in writing the review and making the decision. However, we would like to point out that the current evaluation is a severe misinterpretation of the contributions of this paper.
> >
> > **Regarding similarities between this paper and [3], [9], [11]**
> >
> > The main theoretical contributions in this paper are fundamentally different from these works. We highlight these differences as follows:
> >
> > - The quantile risk minimization problem in [3] aims to minimize a quantile of the risk distribution, and is fundamentally different from our Problem 1 which aims to minimize the distance between training and test risks. Meanwhile, [3] not only ignores the difference between training and test domains, but also adopts the stringent i.i.d assumption for all domains. In contrast, we separately consider the impact of training and test domains on the generalization error, and adopt a much milder assumption by allowing correlations between training (or test) domains.
> >
> > - The upper bounds provided in [9] are derived in terms of the $\mathcal{H}$-divergence, which cannot be used as our information-theoretic quantities to motivate gradient and representation alignment operations. Furthermore, the correlation between the hypothesis and source domains, which is not considered in [9], is crucial to deriving source-domain generalization bounds via input-output mutual information and explaining how gradient alignment avoids overfitting domain-specific correlations.
> >
> > - [11] only considers the distribution shift between training and test samples while ignoring the impact of multiple training/test domains, and does not provide any theoretical results concerning the generalization error. This prohibits its applications to domain generalization algorithms, which aim to learn invariance and generalize across multiple source domains. In contrast, we explicitly analyze the impact of multiple training/test domains, which is the core of our theoretical contribution when motivating the algorithm design.
> >
> > - Besides our generalization analysis, the main part of our contributions regarding theoretical explanations for the success of gradient and representation alignment under milder assumptions in Section 4, is completely ignored in the review. These novel insights have never been explored by previous works.
> >
> > **Regarding the choice of test-domain validation**
> >
> > As indicated in our response to Reviewer pzid and G8me, test-domain validation is widely adopted in various DG works, and is also applicable in practice. Meanwhile, training-domain validation has been criticized by multiple works in the literature. We also give a comprehensive discussion between training and test-domain validation strategies in Appendix F.3 to motivate our choice. Unfortunately, These arguments are completely ignored in the review.
> >
> > **Regarding the empirical performance of IDM**
> >
> > As seen in our response to Reviewer pzid, G8me and pEKy, IDM not only achieves the highest average accuracy on DomainBed, but also the best rankings (mean, median, and worst). It is shown that IDM is the only algorithm that consistently achieves top accuracies on DomainBed. These arguments are all ignored in the review.
> >
> > **Regarding the clarity of notions and concepts**
> >
> > Multiple concerns are raised regarding the definition of training domains $D_{tr}$ and test domains $D_{te}$. Actually, a clear explanation of these concepts is given in Section 2: "We denote random variables by capitalized letters", "The source domains $D_{tr}$ and target domains $D_{te}$ are both randomly sampled from $\nu$". These statements are unfortunately missed by the reviewer, and $D_{tr}$, $D_{te}$ are thought to be distributions instead of random variables.
> >
> > **Regarding the related work SWAD**
> >
> > We cite SWAD for completeness, but still emphasize that the performance is generally incomparable. This is because SWAD not only requires accessing validation accuracies in each update, but also uses completely different hyper-parameter settings in DomainBed. Since only the validation accuracy of the last epoch is available for test-domain selection, such comparisons would be extremely unfair to DG algorithms listed in Table 2.

---

> > > ### Comment · Reviewer_pEKy · 2023-11-21
> > >
> > > > The main theoretical contributions in this paper are fundamentally different from these works...
> > >
> > > Thanks for the clarification, but still the existence of [3] and [11] decreases the contributions.
> > >
> > > > test-domain validation is widely adopted in various DG works
> > >
> > > Test-domain validation is rare practice in DG.
> > >
> > > In Appendix F.3, "Test-domain validation is provided by the DomainBed benchmark as one of the default modelselection methods, and is also widely adopted in the literature in many significant works like IRM
> > > (Arjovsky et al., 2019), V-Rex (Krueger et al., 2021), and Fishr (Rame et al., 2022)". IRM belongs to invariant learning, which is  another important branch of OOD generalization more concentrated on concept shift whereas covariate prevails in DG, and the usage of test-domain validation is not justified in this paper, but can only be seen in the code. As for V-Rex, in their paper, "We evaluate V-REx on DomainBed using the most commonly used training-domain validation set method for model selection."
> > >
> > > To be persuasive, please list references other than Fishr.
> > >
> > > >  and is also applicable in practice...We also give a comprehensive discussion between training and test-domain validation strategies in Appendix F.3 to motivate our choice.
> > >
> > > As I said in my previous response "If test data is available, why not use DA algorithms directly?" The reasonableness can be verified only if you have a labeled dataset which cannot be used for training but can only be used for validation, but this seems impractical.
> > >
> > > > As seen in our response to Reviewer pzid, G8me and pEKy, IDM not only achieves the highest average accuracy on DomainBed, but also the best rankings (mean, median, and worst). It is shown that IDM is the only algorithm that consistently achieves top accuracies on DomainBed. These arguments are all ignored in the review.
> > >
> > > These arguments are not ignored in the review, but is far from convincing given that:
> > >
> > > - Most improvements are for ColoredMNIST which is a synthetic dataset for concept shift mainly. On all other real-world datasets, where covariate shift prevails as a common scenario in DG, the improvments are marginal. Considering thism directly looking at the average of all datasets is far from persuasive.
> > > - Test-domain validation is not a very practical setting in DG as I state in my responses, while improvements are even more marginal when applying the training-domain.
> > > - Missing comparison with important baseline SWAD.
> > >
> > > > Actually, a clear explanation of these concepts is given in Section 2: "We denote random variables by capitalized letters"
> > >
> > > It is not common practice to consider a domain as a random variable in the scope of DG. Still I suggest clearer statement can be added to Sec 2.
> > >
> > > > This is because SWAD not only requires accessing validation accuracies in each update, but also uses completely different hyper-parameter settings in DomainBed. Since only the validation accuracy of the last epoch is available for test-domain selection, such comparisons would be extremely unfair to DG algorithms listed in Table 2.
> > >
> > > As I suggest in my previous response, even it is unfair to DG algorithms listed in Table 2, you can compare SWAD+IDM with SWAD, without comparing with other algorithms. This does not relate to the issue of unfairness you imply. Beside, also as I mentioned in my previous response "To say the least, you can add SWAD to table 10 where training-domain validation is adopted".
> > >
> > > ## Non-technical issues
> > >
> > > I wonder why there is no response to the non-technical issues I listed in my previous response, at least and especially for the first point. It is just acceptable that you forgot to state clearly in your response for the revision of pdf and you may explain that, but I am getting even more confusing why this issue is completely ignored since this is a matter of academic morality.

---

> > > > ### Author Response · Authors · 2023-11-21
> > > > **Response to Reviewer pEKy**
> > > >
> > > > It seems that Reviewer pEKy consistently refuses to acknowledge our main contributions discussed in the rebuttal, but keeps on emphasizing certain similarities to related works which are already clarified in our response.
> > > >
> > > > **Regarding the choice of test-domain validation**
> > > >
> > > > Addressing the concept shift is also one of the main goals of domain generalization as we discussed in Section 3, and IRM is certainly a DG algorithm as it appears as a baseline method in DomainBed. Regarding the V-REx algorithm, the main performance gain of V-Rex is achieved on Colored MNIST, which definitely requires test-domain validation. Meanwhile, the advantage of V-Rex is completely lost using training-domain validation as seen in Table 3 in the V-Rex paper.
> > > >
> > > > Furthermore, various DG algorithms in the literature adopt test-domain validation as one of the model-selection strategies, as seen in SAND-mask, CausIRL, SelfReg, EQRM, etc. It is also reasonable to label a few target-domain samples, where the number of samples is insufficient for training models but enough for validation purposes. Such scenarios are not impractical since the cost of data annotation is expensive.
> > > >
> > > > **Regarding the empirical performance of IDM**
> > > >
> > > > It seems that the reviewer still refuses to acknowledge that IDM simultaneously achieves the best mean, median, and worst rankings among all baseline methods. This indicates that IDM is the only algorithm that consistently achieves top accuracies on DomainBed under various distribution shifts. We believe these scores serve as strong evidence for the effectiveness of IDM.
> > > >
> > > > **Regarding the related work SWAD**
> > > >
> > > > As indicated by our previous responses, evaluating the performance of IDM+SWAD is beyond the scope of this paper, as we are mainly interested in comparing IDM to conventional learning algorithms, instead of model selection or ensemble methods that rely on the trajectory of validation accuracies. Solely comparing SWAD with IDM+SWAD hardly demonstrates the effectiveness of the proposed method since the comparison against all other baseline methods is lacking. It is also unfair to compare with SWAD in Table 10, since the hyper-parameter settings adopted by SWAD are different from the original ones in DomainBed.
> > > >
> > > > **Regarding non-technical issues**
> > > >
> > > > We are sorry for forgetting to inform the reviewers that the manuscript has been updated. However, we are not ignoring these issues but just postponing relevant discussions until the major concerns raised by the reviewer are sufficiently addressed. Such postponement enables us to put our main effort into clarifying the misconceptions in the review and try our best to help the reviewer understand the main contributions of our work. Also, these issues should not influence the evaluation of our contributions in this paper.

---

> > > > > ### Comment · Reviewer_pEKy · 2023-11-22
> > > > >
> > > > > > Addressing the concept shift is also one of the main goals of domain generalization as we discussed in Section 3
> > > > >
> > > > > I am not sure whether this is practical since covariate shift is dominant in current DG real-world datasets while only Colored MNIST supports this. From my perspective, to demonstrate the practical significance of **addressing concept shift in DG**, you need to find off-the-shelf real-world datasets where concept shift dominates instead of covariate shift or collect a new realistic one instead of Colorer MNIST only. For example, concept shift prevails in tabular datasets, which I am not sure whether IDM is able to address well without further experiments.
> > > > >
> > > > > > IRM is certainly a DG algorithm as it appears as a baseline method in DomainBed.
> > > > >
> > > > > Appearing in DomainBed does not mean that it was designed for DG initially. And as stated, "the usage of test-domain validation is not justified in this paper, but can only be seen in the code."
> > > > >
> > > > > > Regarding the V-REx algorithm, the main performance gain of V-Rex is achieved on Colored MNIST, which definitely requires test-domain validation. Meanwhile, the advantage of V-Rex is completely lost using training-domain validation as seen in Table 3 in the V-Rex paper.
> > > > >
> > > > > The test-domain validation results of Color MNIST are either added with strikethrough in Table 2, implying that authors disagree with such practice. Meanwhile, the lost of advantage of V-Rex under training-domain validation does not serve as justification for test-domain validation.
> > > > >
> > > > > > various DG algorithms in the literature adopt test-domain validation as one of the model-selection strategies, as seen in SAND-mask, CausIRL, SelfReg, EQRM, etc.
> > > > >
> > > > > For SelfReg and EQRM, results in their main-body are based on train-domain validation while test-domain validation is only inlcuded in appendix. For SAND-mask and CausIRL, they are not peer-reviewed papers, not qualified enough to justify the usage of test-domain validation.
> > > > >
> > > > > >  It is also reasonable to label a few target-domain samples, where the number of samples is insufficient for training models but enough for validation purposes. Such scenarios are not impractical since the cost of data annotation is expensive.
> > > > >
> > > > > To demonstrate the feasibility of labeling a few target-domain samples that are enough for validation purposes, experiments are required that validation is conducted on only a few target-domain samples. In current experiments, at least 20% of target-domain data is used for validation, which is sufficient for DA algorithms or directly training together.
> > > > >
> > > > > > It seems that the reviewer still refuses to acknowledge that IDM simultaneously achieves the best mean, median, and worst rankings among all baseline methods.
> > > > >
> > > > > I checked my statements carefully, and I am certain that I did not refuse to acknowledge that. In my previous responses, I consider this as "far from persuasive" since it is conducted with test domain validation and most improvements are brought on ColoredMNIST. This has a large different from "refuse to acknowledge". Please be careful with your expressions, and avoid factual mistakes and false accusation.
> > > > >
> > > > > > Solely comparing SWAD with IDM+SWAD hardly demonstrates the effectiveness of the proposed method since the comparison against all other baseline methods is lacking.
> > > > >
> > > > > No comment. I think we should let other reviewers and AC to consider whether such a demonstration will be effective or not.
> > > > >
> > > > > > It is also unfair to compare with SWAD in Table 10, since the hyper-parameter settings adopted by SWAD are different from the original ones in DomainBed.
> > > > >
> > > > > Although SWAD has a smaller hyperparameter search space, you can compare with SWAD under the original HP search space or their search space. If SWAD+IDM outperforms SWAD in Table 10 on either of the HP setting, it would be more persuasive than now.
> > > > >
> > > > > > However, we are not ignoring these issues but just postponing relevant discussions until the major concerns raised by the reviewer are sufficiently addressed. Such postponement enables us to put our main effort into clarifying the misconceptions in the review and try our best to help the reviewer understand the main contributions of our work.
> > > > >
> > > > > It is the first time I see that authors do not address reviews' concerns all at once, and postpone some of the concerns until others are resolved. I think we should let other reviewers and AC to consider this situation.
> > > > >
> > > > > > Also, these issues should not influence the evaluation of our contributions in this paper.
> > > > >
> > > > > Even though this does not affect the technical contributions much, this certainly affects the entire evaluation of this paper for deciding acceptance or not since this could be a matter of academic morality.

---

> > > > > ### Comment · Reviewer_pEKy · 2023-11-22
> > > > > **Please clearly state the existence of the pdf revision and modifications in this version in a common response.**
> > > > >
> > > > > Since other reviewers have not replied to you, I suggest you clearly state the existence of the pdf revision and modifications in this version in a common response in order to avoid their puzzlement.

---

### Official Review · Reviewer_G8me · 2023-11-02

**Soundness:** 3 good
**Presentation:** 3 good
**Contribution:** 3 good
**Rating:** 8
**Confidence:** 4

**Summary:**

This paper addresses the domain generalization problem through the lens of information theory. Under some mild conditions, the authors provide useful statements to understand the domain generalization problem: (1) the achievable level of average-case risk is constrained by the degree of concept shift; (2) test-domain population risk is an unbiased estimate of unconditional population risk; (3) generalization bounds for source and target domains. Based on the theoretical analysis, the inter-domain distribution matching to solve the high-probability domain generalization problem is introduced. Finally, the authors applied the proposed method by comparing it with several representative domain generalization algorithms.

**Strengths:**

- Sound theoretical analysis of domain generalization problem, which aligns with previous (empirical) findings.
- Most previous work focused on worst-case optimization, which is more appropriate for subpopulation shift. However, this paper considers a random testing domain, which covers more general cases.
- Proposes a novel algorithm based on the theoretical analysis. Provides extensive experimental results and thorough analysis.

**Weaknesses:**

- It might have been better if the authors included some other DG benchmark dataset other than the domainbed, such as WILDS, which is often regarded as more realistic. Also, it could have been more interesting if the authors considered some tasks other than computer vision classification.
- Even if the authors included extensive sets of datasets and algorithms in Table 2, it doesn't seem like the methods show substantially different average accuracies, except for CMNIST. It seems like in most cases, the IDM doesn't completely fail, however, at the same time, it does clearly outperform the others in most cases.
- Comparing the results presented in Table 2 (model selection using test domain) and Table 10 (model selection using training domain) seems to suggest that the practical utility of the IDM might be a bit limited; however, still, it outperforms the ERM, implying that the IDM provides better generalizability.

**Questions:**

- Just a follow-up question about "it doesn't seem like the methods show substantially different average accuracies." It might be because, as many previous studies reported, the DG methods do not outperform the ERM by a large margin. However, on the other hand, I think it probably is because the authors used the average of accuracies from multiple cases with different combinations of training/testing domains. If we look into more details, e.g., relative improvement compared to the ERM, or the worst case, we might be able to observe more apparent differences. For example, for the PACS dataset, P and A as testing domains are easier (higher accuracies), while C and S as testing domains are relatively more difficult (lower accuracies). Improvement of e.g., 2%p from P as testing and S as testing should be treated differently. But by averaging, one is ignoring such differences.
- For a similar reason, not sure if the average across different datasets makes sense.
- It seems like finding a decent model selection strategy for the IDM is left as a future study. Do the authors have any rough ideas on that?
- Plans to share the codes to reproduce all the results?

---

> ### Author Response · Authors · 2023-11-17
> **Response to Reviewer G8me (1/2)**
>
> Dear Reviewer G8me, thank you for the constructive suggestions! We address the raised questions as follows:
>
> **It might have been better if the authors included some other DG benchmark dataset other than the domainbed, such as WILDS, which is often regarded as more realistic. Also, it could have been more interesting if the authors considered some tasks other than computer vision classification.**
>
> We agree that conducting more experiments on datasets beyond DomainBed would yield more convincing results. However, our primary goal is to verify the performance gain of IDM compared to previous distribution matching-based competitors (e.g. CORAL, DANN, AND-mask, SAND-mask, IGA) or algorithms able to tackle the concept shift (e.g. IRM, V-Rex, Fishr). Considering that the performances of these algorithms are hardly available for datasets beyond DomainBed, it will be difficult to demonstrate the advantage of IDM over these competitors. Meanwhile, most datasets in DomainBed (i.e. PACS, VLCS, OfficeHome, TerraIncognita, and DomainNet) are also realistic. We believe the current results are sufficient to demonstrate the effectiveness of IDM.
>
> **Even if the authors included extensive sets of datasets and algorithms in Table 2, it doesn't seem like the methods show substantially different average accuracies, except for CMNIST. It seems like in most cases, the IDM doesn't completely fail, however, at the same time, it does clearly outperform the others in most cases.**
>
> It is important to note that achieving top-1 accuracies is extremely hard in DomainBed. As seen in Table 2, no algorithm can outperform all other competitors on more than $1$ dataset.
>
> We highlight that IDM not only achieves the highest average accuracy, but also the best rankings (mean, median, and worst). Most importantly, IDM consistently achieves top accuracies on DomainBed (the worst ranking is 6), while all other algorithms fail to outperform 2/3 of the competitors on at least 1 dataset (the second-best worst ranking is 14 among 22 algorithms). This indicates that IDM is the only algorithm that is consistently among the best methods under various distribution shifts.
>
> **Comparing the results presented in Table 2 (model selection using test domain) and Table 10 (model selection using training domain) seems to suggest that the practical utility of the IDM might be a bit limited; however, still, it outperforms the ERM, implying that the IDM provides better generalizability.**
>
> In fact, the test-domain validation is indispensable for DG algorithms aiming to address concept shifts, e.g. IRM, V-Rex, and Fishr. As seen in Table 10, no algorithm can significantly outperform random-guessing (50\% accuracy) on ColoredMNIST using training-domain validation (ARM is an exception, as we discussed in Section F.3). The choice of test-domain validation is proven to be superior compared to training-domain validation, and is also practical for real applications, as we comprehensively discussed in Section F.3.
>
> **Just a follow-up question about "it doesn't seem like the methods show substantially different average accuracies." It might be because, as many previous studies reported, the DG methods do not outperform the ERM by a large margin. However, on the other hand, I think it probably is because the authors used the average of accuracies from multiple cases with different combinations of training/testing domains. If we look into more details, e.g., relative improvement compared to the ERM, or the worst case, we might be able to observe more apparent differences. For example, for the PACS dataset, P and A as testing domains are easier (higher accuracies), while C and S as testing domains are relatively more difficult (lower accuracies). Improvement of e.g., 2\%p from P as testing and S as testing should be treated differently. But by averaging, one is ignoring such differences.**
>
> As suggested by the reviewer, we report detailed accuracies for each domain in Table 13 - 19 for a comprehensive comparison.
>
> **For a similar reason, not sure if the average across different datasets makes sense.**
>
> The average accuracy is widely adopted as an essential criterion for evaluating the overall performance of DG algorithms in the literature. Besides the average accuracy, we also report different rankings scores (mean, median, and worst), with IDM consistently demonstrating superior scores compared to other algorithms. We believe that these rankings are strong evidence to prove the superiority of the proposed method.

---

> ### Author Response · Authors · 2023-11-17
> **Response to Reviewer G8me (2/2)**
>
> **It seems like finding a decent model selection strategy for the IDM is left as a future study. Do the authors have any rough ideas on that?**
>
> The test-domain validation strategy adopted in this paper can serve as a candidate, as deploying models in target environments without any form of verification is unrealistic in practice, making the effort of labeling a few test-domain samples for validation purposes indispensable.
>
> Numerous efforts have been made toward improving model selection in the literature, e.g. Model Zoo (Chen et al., 2023), and the one mentioned by Reviewer pEKy (SWAD). Designing such strategies is also of great interest to the community, but is beyond the scope of this paper. We will leave this for future research.
>
> Chen Y, Hu T, Zhou F, Li Z, Ma ZM. Explore and exploit the diverse knowledge in model zoo for domain generalization.
>
> **Plans to share the codes to reproduce all the results?**
>
> Our source code is available in the supplementary materials.

---

### Official Review · Reviewer_pzid · 2023-11-03

**Soundness:** 3 good
**Presentation:** 3 good
**Contribution:** 2 fair
**Rating:** 5
**Confidence:** 4

**Summary:**

This paper assumes that test domains are sampled iid from the training domains in a domain generalization problem. First, the authors prove information-theoretic bounds on the generalization error using the mutual information between domain index and inputs (Theorem 5). They also give bounds in terms of the mutual information between domain index and the classifier learned by the algorithm (Theorem 3). Second, an algorithm is proposed as a new method for domain generalization. The algorithm simultaneously matches the distributions of conditional representations and the distributions of gradients (with respect to the classifier head only). In contrast to previous works that does domain matching, the new algorithm matches at each dimension, e.g. sorts the examples along each dimension and try to match the conditional distribution. Third, experiments are performed on ColoredMNIST and DomainNet. This algorithm achieves good performance when selecting hyper-parameters based on the test domain validation set (using test domain labels), but average when selecting hyper-parameters based on training domain validation set.

**Strengths:**

1. The paper is very well written. I like how the assumptions are clearly listed for the theory part.
2. I like the soundness of the claims and conclusions in this paper. Most notably, the authors are very clear about hyperparameter selection criteria and gave clear justifications in the appendix. They also discussed why the algorithm isn't very effective on TerraIncognita and tried to reason about why.

**Weaknesses:**

1. The first major problem I find is that the theoretical part doesn't connect closely to the algorithms. The fact that I(W, D) and I(Z, D) appear in a generalization bound doesn't give direct justification for gradient space or representation space distribution matching. For example, Invariant Risk Minimization (IRM) would be minimizing I(W, D) more directly. The distinction of encoder vs classifier is quite arbitrary for deep neural networks, and I(Z,D) to me seems like a lower bound that says no domain generalization algorithm can do better than I(Z,D) than justifying a representation matching algorithm. The main reason that the authors picked the specific form of algorithms in this paper is perhaps they found superior empirical performance, which leads to the next problem.
2. On the empirical results, the proposed algorithm is a combination of existing gradient-matching algorithm and a somewhat new representation matching algorithm. There's some innovation in the latter part, where the per-dimension sorting and matching is more fine-grained, but the general idea is not new. When we look at empirical performance, the paper shows that this new combination achieves better average OOD accuracy only when tuning hyperparameters on test domains, which many previous papers have argued against [Gulrajani and Lopez-Paz, In Search of Lost Domain generalization]. When selecting on training domains, the method doesn't outperform baselines. The authors made a strong argument in the appendix for their model selection method. I don't have a fundamental objection to the argument, but I think the fact that the proposed algorithm has more hyperparameters than many baselines mean that it has an unfair disadvantage when tuned on test domains.
3. Overall I think the novelty in algorithm and theory is lacking, and the empirical performance are not much better.

**Questions:**

1. Are your generalization bounds vacuous for deep neural networks?
2. Table 6 shows that not doing gradient matching is actually better for OfficeHome. How do you do warm-up when you don't do gradient matching? Does this ablation also hold for other realistic datasets? I think ColoredMNIST is quite contrived and we should show more ablations on realistic datasets.

---

> ### Author Response · Authors · 2023-11-17
> **Response to Reviewer pzid (1/2)**
>
> Dear Reviewer pzid, thanks for your valuable comments! We would like to clarify certain misconceptions about our theoretical contributions:
>
> **The first major problem I find is that the theoretical part doesn't connect closely to the algorithms. The fact that I(W, D) and I(Z, D) appear in a generalization bound doesn't give direct justification for gradient space or representation space distribution matching. For example, Invariant Risk Minimization (IRM) would be minimizing I(W, D) more directly. The distinction of encoder vs classifier is quite arbitrary for deep neural networks, and I(Z, D) to me seems like a lower bound that says no domain generalization algorithm can do better than I(Z, D) than justifying a representation matching algorithm. The main reason that the authors picked the specific form of algorithms in this paper is perhaps they found superior empirical performance, which leads to the next problem.**
>
> We would like to highlight that the proposed algorithms in this paper are well-motivated from our theoretical findings:
>
> - In Section 4 "Gradient Space Distribution Matching", Theorem 6 shows that $I(W;D_i)$ could be minimized by minimizing $I(G_t;D_i|W_{t-1})$ in each step $t$, where $G_t$ is the gradient. By noticing that $I(G_t;D_i|W_{t-1})$ is equivalent to the KL divergence between $P_{G_t|W_{t-1},D_i}$ and $P_{G_t|W_{t-1}}$, this directly motivates us to match these two distributions, i.e. match the inter-domain gradients.
>
> - In Section 4 "Representation Space Distribution Matching", it is shown that the representation space distribution shift $I(R,Y;D)$ controls the test-domain population risk. Combining with Theorem 5 which suggests that minimizing the covariate shift only is sufficient to solve the problem, this directly motivates us to minimize the representation space covariate shift $I(R;D)$, i.e. match the inter-domain representations.
>
> - Combining the theoretical findings above directly suggests that matching the inter-domain gradients and representations simultaneously is a **sufficient condition** to solve Problem 1. In contrast, there are no theoretical guarantees whether invariant risk minimization indeed minimizes $I(W;D_i)$.
>
> **On the empirical results, the proposed algorithm is a combination of existing gradient-matching algorithm and a somewhat new representation matching algorithm. There's some innovation in the latter part, where the per-dimension sorting and matching is more fine-grained, but the general idea is not new. When we look at empirical performance, the paper shows that this new combination achieves better average OOD accuracy only when tuning hyperparameters on test domains, which many previous papers have argued against [Gulrajani and Lopez-Paz, In Search of Lost Domain generalization]. When selecting on training domains, the method doesn't outperform baselines. The authors made a strong argument in the appendix for their model selection method. I don't have a fundamental objection to the argument, but I think the fact that the proposed algorithm has more hyperparameters than many baselines mean that it has an unfair disadvantage when tuned on test domains.**
>
> We agree that the ideas of gradient or representation matching have already been explored in previous works. However, we highlight that this is the first work to give a **clear explanation** of why gradient or representation matching helps domain generalization from an information-theoretic perspective (see our response above), without any **stringent assumptions** on the loss landscape, the features, or the gradients required in previous analyses (e.g. Fishr, IGA, AND-mask). Our theoretical analysis proves that gradient and representation alignment together is a sufficient condition to solve the domain generalization problem, which has never been explored in previous works. We also point out the key defect of previous distribution matching methods (for both representations and gradients) built upon moment matching, which is then solved by the proposed PDM method.
>
> We emphasize that test-domain validation is indispensable for DG algorithms aiming to address concept shifts, e.g. IRM, V-Rex, and Fishr. The choice of test-domain validation is highly motivated by various reasons as we discussed in Section F.3. Moreover, as the total number of hyperparameter tuning attempts is limited to $20$ in DomainBed, this indicates that our algorithm requires no more computational resources for hyperparameter tuning to achieve better performance, despite that we have more hyperparameters than other baselines.

---

> ### Author Response · Authors · 2023-11-17
> **Response to Reviewer pzid (2/2)**
>
> **Overall I think the novelty in algorithm and theory is lacking, and the empirical performance are not much better.**
>
> Please refer to our foregoing response detailing the novelty of our theories and algorithms. We highlight that IDM not only achieves the highest average accuracy, but also the best rankings (mean, median, and worst). Most importantly, IDM consistently achieves top accuracies on DomainBed (the worst ranking is 6), while all other algorithms fail to outperform 2/3 of the competitors on at least 1 dataset (the second-best worst ranking is 14 among 22 algorithms). This indicates that IDM is the only algorithm that is consistently among the best methods under various distribution shifts.
>
> **Are your generalization bounds vacuous for deep neural networks?**
>
> It is not easy to check directly if these bounds are vacuous or not, as the key mutual information measures are computationally intractable. However, the main purpose of these bounds is not to provide upper-bound predictions for the generalization error, but to guide the design of our DG algorithm as discussed above. We comprehensively illustrate how these theoretical findings motivate the design of the IDM algorithm in Section 4.
>
> **Table 6 shows that not doing gradient matching is actually better for OfficeHome. How do you do warm-up when you don't do gradient matching? Does this ablation also hold for other realistic datasets? I think ColoredMNIST is quite contrived and we should show more ablations on realistic datasets.**
>
> Warmup itself does not have any effect without gradient matching. Although ColoredMNIST is synthetic in nature, it is an ideal benchmark to evaluate the ability of DG algorithms to tackle the concept shift, as seen in various works in the literature, e.g. IRM, V-Rex, Fishr.
>
> As seen in Table 5, gradient matching greatly enhances the performance on ColoredMNIST (57.7\% to 72.0\%). Additionally, we conduct ablation studies to examine the effect of gradient matching on other realistic datasets, showing **significant performance gain** on VLCS (Table 7, 77.4\% to 78.1\%) and PACS (Table 8, 86.8\% to 87.6\%). Meanwhile, the accuracy drop on OfficeHome is also mild (68.4\% to 68.3\%) compared to performance gain on other datasets. We believe the current results are sufficient to prove the effectiveness of gradient matching.

---

### Author Response · Authors · 2023-11-22
**General Response to Reviewers**

We thank all anonymous reviewers for your valuable and insightful comments. These suggestions are all helpful for us to improve the manuscript. We have revised the paper to incorporate suggestions raised by the reviewers and also provided individual responses to each reviewer to address your concerns. Below, we address several common issues that have been raised by multiple reviewers:

**The main contribution of this paper**

In this paper, we formulate the DG problem from a novel probabilistic perspective. Under this formulation, we then provide information-theoretic generalization bounds for both source and target-domain generalization risks, which directly motivate our algorithm design by simultaneously matching inter-domain gradients and representations. We highlight that this is the first work to give a clear explanation of why gradient or representation matching helps domain generalization, without any stringent assumptions on the loss landscape, the features, or the gradients required in previous analyses (e.g. Fishr, IGA, AND-mask). Our theoretical analysis suggests that gradient and representation alignment together is a sufficient condition to address our high-probability DG formulation (Problem 1). We also point out that previous distribution matching methods built upon moment matching are ineffective towards more complex distributions. Such a problem is then solved by the proposed PDM method.

**The empirical performance of IDM**

It is important to note that IDM not only achieves significant performance gain on Colored MNIST, but also the highest average accuracy and best rankings (mean, median, and worst). Most importantly, IDM consistently achieves top accuracies on DomainBed (the worst ranking is 6), while all other algorithms fail to outperform 2/3 of the competitors on at least 1 dataset (the second-best worst ranking is 14 among 22 algorithms). This indicates that IDM is the only algorithm that is consistently among the best methods under various distribution shifts. We also provide clear motivations for the choice of test-domain validation, as we comprehensively discussed in Section F.3.

**Performance gain of gradient alignment**

To further demonstrate the effect of gradient alignment on realistic datasets, we additionally replenish ablation studies in the revised manuscript to examine the effect of gradient matching on other realistic datasets, as shown in Tables 7 and 8. It can be seen that without gradient matching, IDM cannot even significantly outperform ERM (77.6\% v.s. 77.4\% on VLCS, 86.7\% to 86.8\% on PACS), while IDM with gradient matching achieves competitive performances to the previous SOTA SelfReg (78.1\% v.s. 78.2\% on VLCS, 87.6\% to 87.7\% on PACS). Meanwhile, the accuracy drop on OfficeHome is also mild (68.4\% to 68.3\%) compared to performance gain on other datasets.

---

### Meta-Review · Area_Chair_FXm6 · 2023-12-08

**Metareview:**

The manuscript has received mixed ratings and extensive discussion during the rebuttal phase. The main concerns of the reviewers are two fold: (1). Algorithmically, both the gradient matching and the feature matching are not new. The specific variant for feature matching proposed in this paper is new, but conceptually it still belongs to the general line of works focusing on feature matching. Gradient matching has been proposed in Fishr, as pointed out by multiple reviewers. (2). Empirically, the generalization results are mixed and the paper could benefit from more extensive experiments under other DG benchmarks, such as the WILDS dataset as pointed out by R2.

I also read the paper myself, and I do share the concerns of the reviewers. Furthermore, I'd also like to point out certain technical concerns:

-   Is Assumption 3 really realistic? It is true that the subgaussian assumption is convenient in analysis for its nice concentration properties. However, the paper focuses on using neural networks for feature learning it is not clear at all whether the loss function will satisfy the subgaussian assumption, especially under highly non-linear neural networks.
-   The reference to Eq. (1) seems off -- it is folklore in the DA/DG literature that the joint shift could be explained by two parts, i.e., the marginal shift and the conditional shift, and this has been formally discussed and pointed out in the literature much earlier than Federici et al., 2021. For example, see [1] (Theorem 3.1) and [2] (Theorem 3.1), which provide two different ways of such decomposition.
-    Theorem 1 is quite obvious -- I'd suggest toning it down a bit to make it a proposition instead so that readers can focus on the more important main theoretical results.
-    As it's currently stated, Problem 1 is not always achievable -- the authors need to make further assumptions on the underlying distributions to make it well-defined, e.g., realizability or noiseless assumption. To see this, what if the Bayes errors of the two domains are very different?

[1].    https://arxiv.org/pdf/2003.04475.pdf
[2].    https://arxiv.org/pdf/2010.04647.pdf

Overall the paper is well-written and I do like the rigor the authors put into the analysis. However, I do not think the paper is ready for publication at this point. I'd suggest the authors address the concerns of the reviewers.

**Justification For Why Not Higher Score:**

Significant technical issues remain even after the rebuttal period.

**Justification For Why Not Lower Score:**

N/A

---

### Decision · Program_Chairs · 2024-01-16

Reject